# $\mu$pscaling Small Models:
# Principled Warm Starts and Hyperparameter Transfer

Yuxin Ma [1]   Nan Chen [1]   Mateo Díaz [1]   Soufiane Hayou [1]   Dmitriy Kunisky [1]   Soledad Villar [1]

## Abstract

Modern large-scale neural networks are often trained and released in multiple sizes to accommodate diverse inference budgets. To improve efficiency, recent work has explored *model upscaling*: initializing larger models from trained smaller ones to accelerate convergence. However, this method can be sensitive to hyperparameters that need to be tuned at the target upscaled model size, which is prohibitively costly to do directly. It remains unclear whether tuning hyperparameters on smaller models and extrapolating via scaling laws is sound in this setting. We address this with principled approaches to width-based upscaling and efficient hyperparameter tuning in this setting. Motivated by $\mu$P and any-dimensional architectures, we introduce a general upscaling method that, like Net2Net, copies and perturbs weights, but uses theoretically grounded, width-dependent scalings for the perturbation noise and optimizer hyperparameters. First, we prove that under zero perturbation, the upscaled model is functionally equivalent to the base model throughout training. Second, we extend the $\mu$P theory to enable infinite-width limit analysis and establish hyperparameter transfer for upscaled models, greatly reducing the tuning cost. We empirically demonstrate that this method is effective on realistic datasets and architectures.

## 1. Introduction

Modern neural network workflows typically involve training families of models at multiple scales. In the early, exploratory stage, smaller models enable rapid prototyping and scaling laws analysis, helping guide architecture choices and forecast the performance of larger variants. Later, for a final release, researchers devote substantially more compute to train much larger models of varying sizes, often shipped as suites to meet diverse downstream hardware, latency, and cost constraints (as, e.g., for LLaMA or GPT). As models of varying scale are typically trained from scratch, this workflow involves a wasteful cycle of disposal and re-training. A natural question is whether smaller trained models can be *upscaled* to warm-start the training of larger ones, transferring information across scales to reduce the total compute required.

Several strategies have been proposed (e.g. Chen et al. (2016); Gong et al. (2019); Chen et al. (2022); Kim et al. (2024)), but they are largely empirical or heuristic in nature. The main theoretical ingredient is *function-preserving weight transformations*: expanding a pre-trained smaller model into a larger one that produces identical outputs for any input, thereby immediately inheriting the smaller model's performance before further optimization. However, the training dynamics of upscaled models could be quite different from those of large models trained from scratch. As a result, to use upscaling effectively requires finding good choices of several hyperparameters that are highly dependent on the model architecture and optimizer. In practice, hyperparameter tuning at the scale of modern large models is infeasible, so hyperparameters are typically chosen by extrapolating from smaller models using scaling laws. Unfortunately, upscaling appears *a priori* to be incompatible with this approach, since training a single model with upscaling involves training *both* at small and then at large scales. Previous work has relied only on informal heuristics for hyperparameter tuning in the presence of upscaling.

> In this work, we develop a theoretical framework for model upscaling, applicable across architectures and optimizers, that enables efficient tuning via *hyperparameter transfer* across scales.

As our **first contribution**, presented in Section 2, we introduce the theory of *dynamic equivalence*. We show that, by combining a function-preserving weight transformation (duplicating and scaling weights) with a coordinated rescaling

---

[1]Department of Applied Mathematics and Statistics, Johns Hopkins University, Baltimore, MD, USA. Correspondence to: Yuxin Ma <yma93@jh.edu>.

*Proceedings of the 43$^{rd}$ International Conference on Machine Learning*, Seoul, South Korea. PMLR 306, 2026. Copyright 2026 by the author(s).

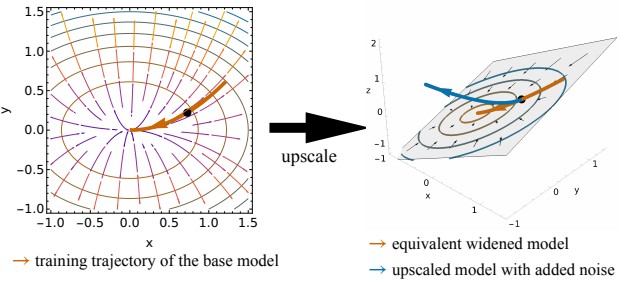

*(a)* Schematic of the upscaling method: without noise, the equivalent widened model remains confined to the low-dimensional weight subspace and follows the same trajectory; adding a small noise allows it to escape and exploit the additional capacity of the higher-dimensional weight space.

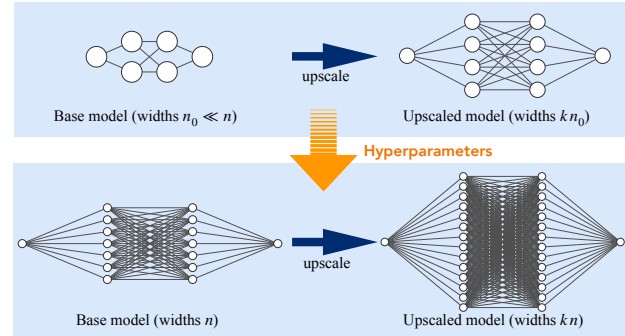

*(b)* Schematic of the hyperparameter transfer method: tuning hyperparameters on a smaller system of upscaled models, and transferring them to a larger one.

*Figure 1.* Illustration of upscaling method and hyperparameter transfer method.

of the learning rate and other optimizer hyperparameters, the wider model computes the same function not only at initialization but also throughout the training process. This extends the existing theoretical underpinning of model upscaling to encompass the training trajectory of the wider model. We further show that the required hyperparameter rescalings are closely connected to the Maximal Update Parametrization ($\mu$P) of Yang & Hu (2021); Littwin & Yang (2023); see Section 2.3.

As our **second contribution**, presented in Section 3, we propose a principled upscaling algorithm grounded in the above theory. Intuitively, dynamic equivalence implies that the training dynamics of the narrower model evolve within a lower-dimensional subspace of the wider model's ambient parameter space. Accordingly, after constructing a dynamically equivalent wider model, we inject a small symmetry-breaking noise into the parameters so that the upscaled dynamics can escape this lower-dimensional subspace and exploit the additional model capacity; see Figure 1a. While the resulting "copy-and-perturb" procedure is mechanically similar to Net2WiderNet (Chen et al., 2016), our theory-guided choices of noise and hyperparameter scaling go substantially beyond it (see Appendix B for a detailed comparison) and yield the following guarantees:

(1) With zero injected noise, the wider model exactly behaves as the base model would, as if upscaling had not been applied (a direct consequence of the dynamic equivalence theory in Section 2).

(2) In the infinite-width limit, the injected noise is provably "safe" and "useful": the base-model signal is preserved, and the injected noise contributes meaningfully without vanishing or exploding.

(3) The procedure enables $\mu$P-grounded zero-shot hyperparameter transfer: one can tune hyperparameters on narrow upscaled models, and the optimal values transfer directly to large-scale upscaling (see Figure 1b). To

our knowledge, this is the first rigorous framework for hyperparameter transfer in the upscaling setting.

As our **third contribution**, presented in Section 3.2 (with details in Appendix E), we use the Tensor Programs machinery (Littwin & Yang, 2023) to rigorously characterize the infinite-width limit of upscaled training for common architectures. This characterization provides the formal basis for guarantees (2) and (3) above and enables further theoretical analysis of training dynamics involving upscaling.

To facilitate adoption, we extend the $\mu$P software package (Yang et al., 2021) so that upscaling can be easily applied to any standard architecture, including multi-layer perceptrons (MLPs), ResNets, and transformers. In Section 4, we present experiments demonstrating the performance of our upscaling algorithm across various architectures and datasets, and illustrate hyperparameter transfer numerically. Our implementation and experiments are publicly available at `https://github.com/yuxinma98/mupscaling`. The related work is summarized in Appendix B; conclusions and limitations are given in Appendix G.

While we focus here on width-wise upscaling, extending to depth is a natural next step that we leave to future work (see, e.g., Yang et al. 2023b; Bordelon et al. 2024b for hyperparameter transfer across depths).

**Notation.** We use the symbol $\odot$ to denote the Hadamard (entry-wise) product and $\otimes$ to denote the Kronecker product. In particular, if we take a matrix $A \in \mathbb{R}^{m \times n}$ and the matrix of all ones $\mathbf{1}_k \mathbf{1}_\ell^\top \in \mathbb{R}^{k \times \ell}$, then, $A \otimes (\mathbf{1}_k \mathbf{1}_\ell^\top)$ is a matrix of size $(mk) \times (n\ell)$, that replaces each entry $A_{i,j}$ of $A$ with a $k \times \ell$ block filled with $A_{i,j}$. In other words, $A \otimes (\mathbf{1}_k \mathbf{1}_\ell^\top)$ corresponds to `A.repeat_interleave(k, dim=0).repeat_interleave(l, dim=1)` in PyTorch. See Appendix A for more details.

## 2. Equivalent Models of Different Widths

Our goal is to use pre-trained small models as a warm start for training larger models. Prior work has approached this via function-preserving weight transformations (Chen et al., 2016), which establish a *static* equivalence between models of different widths: the narrow and wide models parametrize exactly the same function. In this section, we strengthen this to a *dynamic* equivalence, which concerns an architecture–optimizer pair and means that two models of different widths follow identical training trajectories in function space—that is, they parametrize the same function at every training step.

These two perspectives are unified by the framework of Tensor Programs, which we leverage to obtain general results in Section 2.3. As a warm-up, in Section 2.1 we first illustrate both notions separately using a bias-free MLP trained with vanilla stochastic gradient descent (SGD). The subsequent subsections generalize this analysis, first to other optimization methods and then to general architectures.

### 2.1. Warm-up: MLP Trained with SGD

Consider an $L$-layer MLP parametrizing a function from $\mathbb{R}^{d_{\text{in}}}$ to $\mathbb{R}^{d_{\text{out}}}$ that maps $x^{(0)} \mapsto h^{(L)}$ by recursing

$$
\begin{aligned}
h^{(\ell)} &= W^{(\ell)} x^{(\ell-1)} \in \mathbb{R}^{n_\ell}, \\
x^{(\ell)} &= \phi(h^{(\ell)}) \in \mathbb{R}^{n_\ell}, \quad \text{for } \ell = 1, 2, \dots, L,
\end{aligned} \tag{1}
$$

where the input and output dimensions are $n_0 = d_{\text{in}}$ and $n_L = d_{\text{out}}$, respectively, $\big(W^{(\ell)} \in \mathbb{R}^{n_\ell \times n_{\ell-1}}\big)_{\ell=1}^{L}$ are the layer-wise weight matrices, and $\phi$ is the activation function applied elementwise at each layer. Fixing $n_0$ and $n_L$ and increasing the hidden widths $n_1, \dots, n_{L-1}$ yields wider, more expressive MLP models.

We first show that duplicating and appropriately rescaling the weight matrices of any base MLP produces a widened MLP that parametrizes the exact same function. This construction is a special case of the Net2WiderNet operation of Chen et al. (2016), adapted to our notation.

**Proposition 2.1** (Static equivalence of MLPs). *Consider a base MLP with weight matrices $\big(W^{(\ell)} \in \mathbb{R}^{n_\ell \times n_{\ell-1}}\big)_{\ell=1}^{L}$. Construct a widened MLP that uses the same activation function and preserves the input and output dimensions, with weights obtained by duplicating and rescaling those of the base MLP as*

$$
\big(W^{\uparrow(\ell)} := k_{\ell-1}^{-1} W^{(\ell)} \otimes \mathbf{1}_{k_\ell} \mathbf{1}_{k_{\ell-1}}^{\top} \in \mathbb{R}^{N_\ell \times N_{\ell-1}}\big)_{\ell=1}^{L}, \tag{2}
$$

*where $N_\ell = k_\ell n_\ell$ for all $\ell$ and the width multipliers $k_\ell \in \mathbb{N}$ satisfy $k_0 = k_L = 1$. Then, for any input $x^{(0)} \in \mathbb{R}^{d_{\text{in}}}$, both networks produce identical outputs $h^{(L)} \in \mathbb{R}^{d_{\text{out}}}$ and thus parametrize the same function.*

*Proof.* We check by induction over layers that the activations of the widened MLP, denoted by $x^{\uparrow(\ell)} \in \mathbb{R}^{N_\ell}$, are duplicated versions of those of the base MLP, denoted by $x^{(\ell)} \in \mathbb{R}^{n_\ell}$. Specifically, assume $x^{\uparrow(\ell-1)} = x^{(\ell-1)} \otimes \mathbf{1}_{k_{\ell-1}}$. Then

$$
\begin{aligned}
h^{\uparrow(\ell)} &= W^{\uparrow(\ell)} x^{\uparrow(\ell-1)} \\
&= k_{\ell-1}^{-1} \big(W^{(\ell)} \otimes \mathbf{1}_{k_\ell} \mathbf{1}_{k_{\ell-1}}^{\top}\big) \big(x^{(\ell-1)} \otimes \mathbf{1}_{k_{\ell-1}}\big) \\
&= h^{(\ell)} \otimes \mathbf{1}_{k_\ell},
\end{aligned}
$$

and hence $x^{\uparrow(\ell)} = x^{(\ell)} \otimes \mathbf{1}_{k_\ell}$ as well. $\qquad\square$

Following the forward pass, backpropagation for the MLP defined in (1) proceeds as follows:

$$
\begin{aligned}
dh^{(L)} &= \nabla_{h^{(L)}} \mathcal{L} \in \mathbb{R}^{d_{\text{out}}}, \\
dx^{(\ell-1)} &= (W^{(\ell)})^{\top} dh^{(\ell)} \in \mathbb{R}^{n_{\ell-1}}, \\
dh^{(\ell-1)} &= dx^{(\ell-1)} \odot \phi'(h^{(\ell-1)}) \in \mathbb{R}^{n_{\ell-1}}, \qquad (3) \\
dW^{(\ell)} &= dh^{(\ell)} (x^{(\ell-1)})^{\top} \in \mathbb{R}^{n_\ell \times n_{\ell-1}}, \\
&\quad \text{for } \ell = L, L-1, \dots, 1,
\end{aligned}
$$

where $\mathcal{L}$ denotes the loss, and each gradient $d\bullet$ equals $\frac{\partial \mathcal{L}}{\partial \bullet}$. Applying SGD with layer-wise learning rate $\gamma^{(\ell)}$ to $W^{(\ell)}$, the weights are updated at each training step $t$ by

$$
W_{t+1}^{(\ell)} := W_t^{(\ell)} - \gamma^{(\ell)} dW_t^{(\ell)},
$$

where $W_t^{(\ell)}, dW_t^{(\ell)}$ denote the weight and the gradient at $t$-th training step respectively.

We further show that if the per-layer learning rates of the widened MLP are chosen as a specific rescaling of those in the base MLP, then the two equivalent models undergo equivalent SGD updates and hence follow identical training trajectories in function space.

**Proposition 2.2** (Dynamic equivalence of MLPs trained with SGD). *Suppose we have a base MLP with weights $\big(W^{(\ell)} \in \mathbb{R}^{n_\ell \times n_{\ell-1}}\big)_{\ell=1}^{L}$ trained by SGD with per-layer learning rates $\gamma^{(\ell)}$. Construct a widened MLP with the same activation function and with weights $\big(W^{\uparrow(\ell)} := k_{\ell-1}^{-1} W^{(\ell)} \otimes \mathbf{1}_{k_\ell} \mathbf{1}_{k_{\ell-1}}^{\top} \in \mathbb{R}^{N_\ell \times N_{\ell-1}}\big)_{\ell=1}^{L}$, where $N_\ell = k_\ell n_\ell$ for all $\ell$ and $k_0 = k_L = 1$, and train it by SGD using per-layer learning rates $\gamma^{\uparrow(\ell)} := k_\ell k_{\ell-1}^{-1} \gamma^{(\ell)}$. Then, for all training steps $t \geq 0$, the weights satisfy*

$$
W_t^{\uparrow(\ell)} = k_{\ell-1}^{-1} W_t^{(\ell)} \otimes \mathbf{1}_{k_\ell} \mathbf{1}_{k_{\ell-1}}^{\top} \tag{4}
$$

*for all $\ell = 1, \dots, L$, and therefore both networks parametrize the same function at every step (assuming they access the same data and randomness).*

Note that if the width multipliers are equal across dimensions ($k_1 = \dots = k_{L-1}$), then this procedure leaves the

learning rates of all hidden weights unchanged ($\gamma^{\uparrow(\ell)} := \gamma^{(\ell)}$ for all $\ell = 2, \ldots, L-1$), and only modifies the learning rates for $W^{(1)}$ and $W^{(L)}$.

*Proof.* By Proposition 2.1, prior to training (at $t = 0$) the two MLPs parametrize the same function. Consequently, the gradient at the output layer matches, $dh^{\uparrow(L)} = dh^{(L)}$, when computed on the same loss and data. One can check by induction that for $\ell = L-1, \ldots, 1$ the backpropagated signals satisfy

$$dx^{\uparrow(\ell)} = k_\ell^{-1} dx^{(\ell)} \otimes \mathbf{1}_{k_\ell}, \qquad dh^{\uparrow(\ell)} = k_\ell^{-1} dh^{(\ell)} \otimes \mathbf{1}_{k_\ell},$$

$$dW^{\uparrow(\ell)} = k_\ell^{-1} dW^{(\ell)} \otimes \mathbf{1}_{k_\ell} \mathbf{1}_{k_{\ell-1}}^\top.$$

With these identities, one SGD step on the widened weights yields

$$\begin{aligned}
& W^{\uparrow(\ell)} - \gamma^{\uparrow(\ell)} dW^{\uparrow(\ell)} \\
&= k_{\ell-1}^{-1} W^{(\ell)} \otimes \mathbf{1}_{k_\ell} \mathbf{1}_{k_{\ell-1}}^\top \\
&\quad - \left( k_\ell k_{\ell-1}^{-1} \gamma^{(\ell)} \right) \left( k_\ell^{-1} dW^{(\ell)} \otimes \mathbf{1}_{k_\ell} \mathbf{1}_{k_{\ell-1}}^\top \right) \\
&= k_{\ell-1}^{-1} (W^{(\ell)} - \gamma^{(\ell)} dW^{(\ell)}) \otimes \mathbf{1}_{k_\ell} \mathbf{1}_{k_{\ell-1}}^\top,
\end{aligned}$$

which preserves the widening relation (4). By induction over $t$, (4) holds for all steps. $\square$

## 2.2. Extension to General Optimizers

We proceed to extend the previous observation from vanilla SGD to general entrywise optimizers considered in Littwin & Yang (2023), where parameter updates depend on the current and past gradients. This framework encompasses many commonly used optimizers, including Adam (Kingma & Ba, 2015) and AdamW (Loshchilov & Hutter, 2019).

**Definition 2.3** (Entrywise optimizer with weight decay). For a weight matrix $W \in \mathbb{R}^{n \times m}$, an *entrywise optimizer* (with learning rate $\gamma$) updates $W$ at training step $t$ according to the following rules for $\alpha \in [n]$ and $\beta \in [m]$.

- Under weight decay with constant $\lambda$,

$$\begin{aligned}
(W_{t+1})_{\alpha,\beta} &= (W_t)_{\alpha,\beta} \\
&- \gamma Q_t \left( (dW_0 + \lambda W_0)_{\alpha,\beta}, \ldots, (dW_t + \lambda W_t)_{\alpha,\beta}; \varepsilon \right).
\end{aligned} \tag{5}$$

- Under decoupled weight decay with constant $\lambda$,

$$\begin{aligned}
(W_{t+1})_{\alpha,\beta} &= (1 - \lambda\gamma)(W_t)_{\alpha,\beta} \\
&- \gamma Q_t \left( (dW_0)_{\alpha,\beta}, \ldots, (dW_t)_{\alpha,\beta}; \varepsilon \right).
\end{aligned} \tag{6}$$

Here, $\varepsilon \in \mathbb{R}^s$ refers to additional hyperparameters that may also be scaled, e.g., `eps` in the PyTorch implementation of Adam (Paszke et al., 2017), and $Q_t : \mathbb{R}^{t+1+s} \to \mathbb{R}$ is an *update function*, acting as a temporal filter of the gradient history, which can encode momentum and adaptivity.

Next, we show that for any such optimizer with a *homogeneous* update function, it is possible to choose the learning rate, weight decay coefficient, and additional hyperparameters so that the widened MLP is dynamically equivalent.

**Proposition 2.4** (Dynamic equivalence of MLPs trained with general optimizers). *Consider an entrywise optimizer whose update function $Q_t$ is homogeneous of degree $m$ for all $t$, i.e, for any $t \in \mathbb{N}$, with $x_0, \ldots, x_t \in \mathbb{R}$ and $\varepsilon \in \mathbb{R}^s$, we have that for all $a \in \mathbb{R}$,*

$$Q_t(ax_0, \ldots, ax_t; a\varepsilon) = a^m Q_t(x_0, \ldots, x_t; \varepsilon).$$

*Suppose we have a base MLP with weights $\left( W^{(\ell)} \in \mathbb{R}^{n_\ell \times n_{\ell-1}} \right)_{\ell=1}^L$ trained by the above optimizer with per-layer learning rate $\gamma^{(\ell)}$, weight decay coefficient $\lambda^{(\ell)}$, and additional hyperparameter $\varepsilon^{(\ell)}$. Construct a widened MLP with the same activation function and with weights $\left( W^{\uparrow(\ell)} := k_{\ell-1}^{-1} W^{(\ell)} \otimes \mathbf{1}_{k_\ell} \mathbf{1}_{k_{\ell-1}}^\top \in \mathbb{R}^{N_\ell \times N_{\ell-1}} \right)_{\ell=1}^L$, where $N_\ell = k_\ell n_\ell$ for all $\ell$ and $k_0 = k_L = 1$, and train it with the same optimizer using the following hyperparameters:*

$$\gamma^{\uparrow(\ell)} := k_\ell^m k_{\ell-1}^{-1} \gamma^{(\ell)}, \qquad \varepsilon^{\uparrow(\ell)} := k_\ell^{-1} \varepsilon^{(\ell)},$$

$$\lambda^{\uparrow(\ell)} := \begin{cases} k_{\ell-1} k_\ell^{-1} \lambda^{(\ell)} & \text{for vanilla weight decay,} \\ k_{\ell-1} k_\ell^{-m} \lambda^{(\ell)} & \text{for decoupled weight decay.} \end{cases}$$

*Then, for all training steps $t \geq 0$, the weights satisfy*

$$W_t^{\uparrow(\ell)} = k_{\ell-1}^{-1} W_t^{(\ell)} \otimes \mathbf{1}_{k_\ell} \mathbf{1}_{k_{\ell-1}}^\top$$

*for all $\ell = 1, \ldots, L$, and therefore both networks parametrize the same function at every step.*

We prove the Proposition in Appendix C.1, and describe explicitly how to instantiate it for the SGD (including variants with momentum), Adam, and AdamW optimizers.

## 2.3. Extension to General Network Architectures

Finally, we present our general result, showing that the observations made above for MLPs extend to virtually all "standard" neural network architectures. Here, we state an informal result since the formal version requires additional technical background. A detailed version of this result is deferred to Theorem C.5 in Appendix C.2.

**Theorem 2.5** (Informal). *Consider a "standard" neural network architecture where the final readout step averages rather than sums along the width axis.[1] Assume an entrywise optimizer whose update functions are homogeneous of degree $m$. Suppose that we train a base model with*

---

[1] Standard architectures typically sum along the width axis in the final readout. We instead average. For example, in an MLP where the final readout is $W^{(L)} x^{(L-1)}$ with $W^{(L)} \in \mathbb{R}^{d_{\text{out}} \times n_{L-1}}$, we replace it with $n_{L-1}^{-1} W^{(L)} x^{(L-1)}$.

*Table 1.* Construction of weights and optimizer hyperparameters to obtain a widened model with equivalent training trajectories.

| Type of weights | Weights widening operation | Hyperparameter rescaling |
|---|---|---|
| Scalar-like | $W^{\uparrow} := W$ | $\gamma^{\uparrow} := \gamma, \quad \lambda^{\uparrow} := \lambda, \quad \varepsilon^{\uparrow} := \varepsilon$ |
| Vector-like $\mathbb{R}^{n \times d} \to \mathbb{R}^{N \times d}$ *width* $n \to$ *width* $N := nk$ | $W^{\uparrow} := W \otimes \mathbf{1}_k$ `W.repeat_interleave(k, dim=0)` | $\gamma^{\uparrow} := k^m \gamma, \quad \varepsilon^{\uparrow} := k^{-1}\varepsilon$ $\lambda^{\uparrow} := \begin{cases} k^{-1}\lambda & \text{(vanilla)} \\ k^{-m}\lambda & \text{(decoupled)} \end{cases}$ |
| Matrix-like $\mathbb{R}^{n_{\text{out}} \times n_{\text{in}}} \to \mathbb{R}^{N_{\text{out}} \times N_{\text{in}}}$ *width* $(n_{\text{out}}, n_{\text{in}}) \to$ *width* $(N_{\text{out}}, N_{\text{in}})$ $N_{\text{out}} := n_{\text{out}}k_{\text{out}}, \quad N_{\text{in}} := n_{\text{in}}k_{\text{in}}$ | $W^{\uparrow} := k_{\text{in}}^{-1} W \otimes \left(\mathbf{1}_{k_{\text{out}}} \mathbf{1}_{k_{\text{in}}}^{\top}\right)$ `W.repeat_interleave(k_out, dim=0)` `.repeat_interleave(k_in, dim=1) / k_in` | $\gamma^{\uparrow} := k_{\text{out}}^m k_{\text{in}}^{-1}\gamma, \quad \varepsilon^{\uparrow} := k_{\text{out}}^{-1}\varepsilon,$ $\lambda^{\uparrow} := \begin{cases} k_{\text{in}} k_{\text{out}}^{-1}\lambda & \text{(vanilla)} \\ k_{\text{in}} k_{\text{out}}^{-m}\lambda & \text{(decoupled)} \end{cases}$ |

*learning rate $\gamma$, weight decay coefficient $\lambda$, and additional hyperparameter $\varepsilon$. Consider a widened model with the same architecture and depth but larger width. Assume that the widened model's weights are obtained by duplicating units along the designated width axes and rescaling appropriately, and that all hyperparameters are rescaled in tandem according to Table 1. Then, at every training step, the base and widened models parametrize the same function.*

The term "standard" architecture refers to any neural network representable in the $\text{NE}{\otimes}\text{OR}{\top}$ program developed in the Tensor Program literature (Yang, 2019; 2020a; Littwin & Yang, 2023). Intuitively, $\text{NE}{\otimes}\text{OR}{\top}$ is a formal programming language in which each variable is typed as matrix-like, vector-like, or scalar-like, and new variables are generated by: (i) multiplication of a vector-like variable by a matrix-like one; (ii) elementwise nonlinear transformations of vector-like variables; and (iii) averaging the entries of a vector-like variable. See Appendix C.2.1 for the formal definition. This framework encompasses many widely used architectures and components, including MLPs, RNNs, convolution, attention, pooling layers, skip connections, and batch/layer normalization, as established by Yang (2019). Furthermore, Littwin & Yang (2023) demonstrates that any network specified in the $\text{NE}{\otimes}\text{OR}{\top}$ program admits a corresponding backpropagation program also expressible in $\text{NE}{\otimes}\text{OR}{\top}$.

In $\text{NE}{\otimes}\text{OR}{\top}$, matrix-like parameters have two dimensions that scale with width, vector-like parameters have one, and scalar-like parameters have none. In Table 1, we treat each of these categories separately in describing how to construct a dynamically equivalent widened model. For example, in MLPs, input/output dimensions are constant while hidden dimensions scale with width. Consequently, input and output weights and hidden biases are vector-like, hidden weights are matrix-like, and the output bias is scalar-like. Similarly, in transformers, the context length is fixed while `d_model`, `n_head`, and `dim_feedforward` could scale with width,[2] so the query, key, value, and output projec-

tion matrices $W^Q, W^K, W^V, W^O \in \mathbb{R}^{\text{d\_model} \times \text{d\_model}}$ are matrix-like. Since nearly any relevant neural computation—including forward and backward passes—can be expressed as a $\text{NE}{\otimes}\text{OR}{\top}$ program, we use this framework to formally extend Propositions 2.1 and 2.4 from MLPs to a much broader class of architectures via similar inductive reasoning.

**Transfer of optimizer internal state.** In practice, the construction of a dynamically equivalent wider model is applied mid-training: the base model has already been trained for some number of steps, and the goal is to construct a wider model that behaves exactly as if training of the base model were continued. For optimizers that maintain internal state (e.g., momentum in SGD, or the first- and second-moment estimates in Adam/AdamW), achieving this requires transferring the accumulated optimizer state alongside the weights. The state is duplicated and rescaled following a similar principle as the weight transfer in Table 1. The specific rules are given in Appendix D.

**Connection to $\mu$P.** Notably, the widening rules in Table 1 are compatible with the $\mu$P scaling of Yang & Hu (2021); Littwin & Yang (2023). For reference, Table 2 summarizes $\mu$P, which prescribes width-dependent choices of initialization and optimization hyperparameters so that training dynamics exhibit optimal feature learning behavior in the infinite-width limit. Now, suppose we have a trained base model and instantiate a widened model by applying the "Weights widening operation" rules in Table 1. We then continue training the widened model under $\mu$P, using base constants $\overline{\gamma}$, $\overline{\lambda}$, and $\overline{\varepsilon}$ chosen to match the base model's hyperparameters at width $n$. (Since $\mu$P only prescribes how hyperparameters scale with width—not their absolute values at any fixed width—such a choice is always possible regardless of how the base model was trained.) The resulting hyperparameters at width $N$ match exactly the "Hyperparameters rescaling" column of Table 1. For exam-

---

[2]We borrow the notation `d_model`, `n_head`, and `dim_feedforward` from the PyTorch implementation of transformer (Paszke et al., 2017).

*Table 2.* $\mu$P width scalings. The output multiplier in the last row should be interpreted as in Theorem 2.5. Scalar-like weights have all factors equal to 1 (no width dependence) and are therefore omitted here. Entries are the width-dependent multiplicative factors $(B, C, D, \tilde{D}, E)$ applied to width-independent base constants (denoted with bars). The actual hyperparameters equal the base constants times the listed powers of the widths. See Appendix C.3 for a detailed comparison with the versions presented in Yang et al. (2021); Littwin & Yang (2023).

| Type of weights | Init variance $\mathcal{N}(0, B \cdot \overline{\sigma}^2)$ | Learning rate $\gamma = C \cdot \overline{\gamma}$ | Weight decay (vanilla) $\lambda = D \cdot \overline{\lambda}$ | Weight decay (decoupled) $\lambda = \tilde{D} \cdot \overline{\lambda}$ | Additional hyperparameters $\varepsilon = E \cdot \overline{\varepsilon}$ | Output multiplier |
|---|---|---|---|---|---|---|
| Vector-like | 1 | $n^m$ | $n^{-1}$ | $n^{-m}$ | $n^{-1}$ | $n^{-1}$ |
| Matrix-like | $n_{\text{in}}^{-1}$ | $n_{\text{out}}^m n_{\text{in}}^{-1}$ | $n_{\text{in}} n_{\text{out}}^{-1}$ | $n_{\text{in}} n_{\text{out}}^{-m}$ | $n_{\text{out}}^{-1}$ | $--$ |

ple, for a matrix-like weight, $\mu$P sets the widened-model learning rate to $\gamma^{\uparrow} = N_{\text{out}}^m N_{\text{in}}^{-1} \overline{\gamma}$, whereas the base model uses $\gamma = n_{\text{out}}^m n_{\text{in}}^{-1} \overline{\gamma}$; therefore $\gamma^{\uparrow} = k_{\text{out}}^m k_{\text{in}}^{-1} \gamma$, exactly as required by Table 1. The same reasoning applies to the remaining hyperparameters and to the other weight types. In other words, while the explicit construction of an equivalent widened model may appear involved, under $\mu$P the procedure becomes essentially mechanical: one transfers the learned weights according to Table 1, and the associated hyperparameters adjust automatically. Consequently, from that point onward, the widened model receives exactly the same parameter updates as the base model, i.e., training proceeds as if no widening had occurred.

In the next section, we leverage this observation and adopt the $\mu$P framework for the upscaled model throughout the rest of the paper. We show that it yields a simple, easily implementable upscaling algorithm and, moreover, leads to desirable properties: it maintains optimal training dynamics, enables hyperparameter transfer, and facilitates infinite-width analysis.

## 3. Training from Upscaled Initialization

The previous sections focused on constructing an equivalent wider model that retains the knowledge learned by the narrower base model, which we call *widening*. To *upscale*—i.e., to initialize training of a wider model from an existing narrow model—we apply such a widening procedure and then inject a small symmetry-breaking noise into the parameters. Without noise, the duplicated weights remain identical throughout training and the widened model cannot exploit its extra capacity; injecting noise perturbs the parameters away from this lower-dimensional subspace, potentially increasing the initialization loss but unlocking the additional capacity of the wider model. This raises two practical questions: how much noise to add, and which hyperparameters (particularly the learning rate) to use when training the upscaled model.

### 3.1. Upscaling Algorithm

We propose an upscaling method that addresses both questions within a unified procedure, applicable to the general architectures and optimizers discussed in Section 2. Meta-algorithm 1 provides the pseudocode, with additional details deferred to Appendix D.

---

**Meta-algorithm 1** Upscaling procedure

---

**Input:** Base model checkpoint at width $n$ and the corresponding optimizer checkpoint; expansion multiplier $k$.

**Output:** Trained upscaled model of width $N = nk$.

**Step 0 (Hyperparameter tuning).** Run Steps 1–4 below on a small system (width $n_0 \to k n_0$ with $n_0 \ll n$) many times with different choices of $\overline{\sigma_\Delta}$ and $\overline{\gamma^{\uparrow}}$, selecting the values that minimize the terminal training loss.

**Step 1.** Widen the base model's checkpoint to width $N$ using the "Weights widening operation" in Table 1.

**Step 2.** Add noise to the widened model, with standard deviation (std) following the same scaling that $\mu$P prescribes for random initialization, controlled by the base constant $\overline{\sigma_\Delta}$.

**Step 3.** Create an optimizer for the upscaled model. For optimizers with internal state, transfer this state by applying a similar widening procedure as used for the weights.

**Step 4.** Train the upscaled model constructed in Step 2 under $\mu$P using the optimizer in Step 3, with learning rate base constant $\overline{\gamma^{\uparrow}}$.

---

For simplicity, this algorithm assumes that all hidden-layer dimensions are equal (i.e., upscaling from width $n$ to $kn$), but it extends straightforwardly to heterogeneous hidden-layer widths. For concreteness, Algorithm 2 in Appendix D instantiates the method for an MLP trained with SGD, where the specific distributions of the injected noise are specified. We also provide a *signal-normalized noise* variant of the algorithm (see Appendix D) that sets the noise magnitude per layer as a fraction $t \in [0, 1]$ of the widened weight's spectral norm; after tuning $t$ at small width, the resulting per-layer

noise base constants (rather than $t$ itself) are transferred to the target width.

This algorithm enjoys the following key properties. The theoretical basis for the first was established in Section 2; we develop that of the second and third in Section 3.2 below.

**Equivalence at zero noise.** By Theorem 2.5 and the $\mu$P connection in Section 2.3, injecting zero noise ($\overline{\sigma_\Delta} = 0$) yields a widened model whose training trajectory evolves exactly as if training continued on the base model (up to a potential change in learning rate base constant $\overline{\gamma^\uparrow}$). We empirically validate this equivalence across architectures and optimizers in Appendix C.4. Because dynamic equivalence governs the parametrized function across the full training trajectory—not just at initialization—it promotes well-behaved training dynamics; Section 3.2 justifies this more fully via infinite-width theory. Dynamic equivalence also provides a rigorous baseline: continuing base-model training is automatically included in any hyperparameter sweep over the noise level that contains zero. Consequently, if the sweep selects a nonzero $\overline{\sigma_\Delta}$, one can be confident that upscaling yields genuine benefits over simply continuing the base-model training.

**Optimal infinite-width dynamics.** The key design choice is the width-dependent scaling of noise and optimizer hyperparameters: the injected noise is scaled in the same way as the initialization in $\mu$P (see Step 2), and the upscaled model's optimizer hyperparameters follow $\mu$P scaling with the same base constants as the base model. In Section 3.2, we show that this ensures the entire training process—including mid-course upscaling and noise injection—exhibits optimal infinite-width dynamics in which the noise injection is both "safe" and "useful": the signal from the base model is preserved in the wider optimization landscape, while the injected noise contributes meaningfully without vanishing or exploding. This provides precise principled guidance for noise injection that prior heuristic approaches lack.

**Hyperparameter transfer.** As specified in Step 0, zero-shot hyperparameter transfer applies directly to upscaling. This significantly reduces the computational overhead of hyperparameter tuning, allowing one to choose the noise level and learning rate in a principled manner. As we show in Section 3.2, the success of this transfer inherits the same theoretical guarantees as $\mu$Transfer (Yang et al., 2021), owing to the infinite-width analysis of the entire training trajectory. We empirically validate this hyperparameter transfer in Section 4.

## 3.2. Infinite-Width Limit of Model Upscaling

Yang & Hu (2021); Littwin & Yang (2023) derive $\mu$P for any neural network architecture expressed in the NE$\otimes$OR$\top$ program (see Table 2), and rigorously show that it is the unique "optimal" parametrization in the infinite-width limit. Concretely, all hidden activations and their updates stay at $\Theta(1)$ scale throughout training, eliminating vanishing and exploding gradients and keeping the model in the "feature learning" regime. Moreover, Yang et al. (2021) empirically observe that training under $\mu$P yields dynamics that align across widths, enabling hyperparameters tuned on narrow models to transfer directly to wider ones.

At first glance, this result appears inapplicable to upscaling. As noted in Section 2.3, NE$\otimes$OR$\top$ program natively supports only matrix multiplication and elementwise nonlinearities of vectors, and, thus, does not explicitly permit the "duplication" operation $\otimes \mathbf{1}\mathbf{1}^\top$ that upscaling involves. Nevertheless, we provide a workaround showing that the upscaling procedure can be encoded within the NE$\otimes$OR$\top$ framework by introducing additional variables, so that the results in Yang et al. (2021) apply directly.

To illustrate the idea, take the MLP example from Section 2.1. Specifically, the first forward propagation for the upscaled network is defined recursively as:

$$h^{(\ell)} = (k_{\ell-1}^{-1} W^{(\ell)} \otimes \mathbf{1}_{k_\ell} \mathbf{1}_{k_{\ell-1}}^\top + \Delta^{(\ell)}) x^{(\ell-1)} \in \mathbb{R}^{N_\ell},$$
$$x^{(\ell)} = \phi(h^{(\ell)}) \in \mathbb{R}^{N_\ell},$$

where $\Delta^{(\ell)}$ denotes the Gaussian noise injected into the model after widening (Step 2 of Meta-algorithm 1).[3] Because of the $\otimes \mathbf{1}_{k_\ell} \mathbf{1}_{k_{\ell-1}}^\top$ operation, these equations do not give a NE$\otimes$OR$\top$ program. However, they can be rewritten in terms of variables of the original, not widened, dimension:

$$h_{(i)}^{(\ell)} = \sum_{j=1}^{k_{\ell-1}} \left( k_{\ell-1}^{-1} W^{(\ell)} + \Delta_{(i,j)}^{(\ell)} \right) x_{(j)}^{(\ell-1)} \in \mathbb{R}^{n_\ell},$$
$$x_{(i)}^{(\ell)} = \phi(h_{(i)}^{(\ell)}) \in \mathbb{R}^{n_\ell}, \quad \text{for } i = 1, \ldots, k_\ell,$$

where $h_{(i)}^{(\ell)} := (h_i^{(\ell)}, h_{i+k_\ell}^{(\ell)}, h_{i+2k_\ell}^{(\ell)}, \ldots) \in \mathbb{R}^{n_\ell}, i \in [k_\ell]$ partition the entries of the widened vector $h^{(\ell)} \in \mathbb{R}^{N_\ell}$, and similarly $(\Delta_{(i,j)}^{(\ell)} \in \mathbb{R}^{n_\ell \times n_{\ell-1}})_{i \in [k_\ell], j \in [k_{\ell-1}]}$ partition the entries of the matrix $\Delta^{(\ell)} \in \mathbb{R}^{N_\ell \times N_{\ell-1}}$. The two above sets of equations describe identical dynamics, but the latter is a combination only of matrix multiplications and elementwise nonlinearities, and therefore falls within the NE$\otimes$OR$\top$ framework.

We generalize this observation in Appendix E.2, showing that any architecture expressible in NE$\otimes$OR$\top$ admits

---

[3]See Algorithm 2 for the explicit distribution of $\Delta^{(\ell)}$ in the MLP case.

an upscaling procedure formulated within the NE⊗OR⊤ framework by introducing such partitioned variables. This allows us to directly apply the results of Yang et al. (2021) and conclude that the width-dependent scalings of noise and hyperparameters in Meta-Algorithm 1 are precisely those ensuring the entire training process—including mid-training upscaling—adheres consistently to μP. Three consequences follow.

First, following Yang & Hu (2021); Yang et al. (2023a), the upscaling method yields "optimal" training dynamics in the infinite-width limit (fixing the width multiplier $k$ and letting the base width $n \to \infty$). In particular, both the initialization and the injected noise make non-trivial contributions to the activations and their updates, maintaining signal from the base model while exploiting increased width.

Second, hyperparameters can be transferred reliably for training procedures that include upscaling, as illustrated in Figure 1b.

Third, using the Tensor Program framework of Littwin & Yang (2023), we can explicitly characterize the infinite-width limit of the entire training dynamics involving upscaling. To streamline future work, we introduce a modified Tensor Program tailored to upscaled training in Appendix E.1. Without upscaling, pre-activations converge to i.i.d. Gaussian random variables in the infinite-width limit. With upscaling by a factor $k$, pre-activations instead converge to i.i.d. blocks, each a $k$-dimensional Gaussian random vector with non-trivial covariance structure. Our modified framework tracks these vectors—through their evolving covariance structure—across the entire training trajectory. We illustrate this on two simple MLPs in Appendices E.3 and E.4, showing that the infinite-width dynamics after upscaling is in general qualitatively different from non-upscaled training. Consequently, optimal hyperparameters for non-upscaled training should not be expected to remain optimal after upscaling, justifying our approach of transferring hyperparameters across upscaling systems of different sizes.

## 4. Experiments

We evaluate the effectiveness of our upscaling algorithm across multiple architectures and optimizers, verifying the theory's predictions on hyperparameter transfer.

**Setup.** We consider three settings: an MLP with AdamW on Forest Cover Type tabular classification (Blackard & Dean, 1999), a ResNet (He et al., 2016) with SGD on CIFAR-100 image classification (Krizhevsky & Hinton, 2009), and GPT-2 (Radford et al., 2019) with AdamW on FineWeb (Penedo et al., 2024). For the MLP, we use a 4-layer ReLU network and upscale from width $n=500$

to $kn=2000$ ($k=4$), with hyperparameters tuned at width $n_0=100$ to $kn_0=400$. For ResNet-18, we upscale from the standard $2\times$ to a $4\times$ model ($k=2$), with hyperparameters tuned at the $0.5\times$ to $1\times$ scale; due to its heterogeneous channel widths, we use the alternative signal-normalized noise variant of the algorithm (Appendix D). For GPT-2 (12 layers, 12 heads), we upscale from 160 to 320 dimensions per head ($k=2$), with hyperparameters tuned at 16 to 32 dimensions per head; this yields a target-to-tuning scale ratio of $n/n_0 = 10$, demonstrating transfer across a large scale separation. Full experimental details, including architecture configurations and hyperparameter search grids, are provided in Appendix F. In all cases, we follow the three-step protocol below.

(1) Train a base model of width $n$ from scratch under μP, with hyperparameters either taken from previously reported best settings or obtained via the transfer procedure of Yang et al. (2021) at width $n_0 \ll n$.

(2) As a baseline, train a wider model of width $N = kn$ from scratch under μP, using the same hyperparameter base constants as in (1), which remain near-optimal by hyperparameter transfer (Yang et al., 2021).

(3) Train another model of width $N = kn$ using our upscaling algorithm (Meta-algorithm 1). The noise std and learning rate are tuned on a small upscaling system $n_0 \to kn_0$, selecting the configuration with lowest training loss after $T$ steps; the selected values are then applied to the target system ($n \to kn$) for the same $T$ steps. All other hyperparameters are kept identical to (1) and (2).

**Evaluation and results.** We treat the base model's training cost as sunk (e.g., because the base model was obtained from an open-source release), and the hyperparameter tuning cost as negligible relative to training the large model. Appendix F.4 relaxes both assumptions and shows that upscaling remains cost-effective even when these costs are fully charged. Under these assumptions, the comparison is direct: both the from-scratch baseline and the upscaled model share the same architecture and width, so we simply plot their loss curves as a function of training steps. Figure 2 summarizes the results. Across settings, the upscaled model offers two clear benefits: it converges faster *and* reaches lower final training loss than the model trained from scratch. In all experiments, the hyperparameter sweep selects a nonzero noise level (varying by dataset, task, and training horizon), which raises the initialization loss above the base model's terminal loss—a necessary trade-off for exploiting additional capacity—but the loss decreases sharply within a few steps. On validation, the upscaled MLP and GPT-2 mirror the same trends seen in training, whereas the upscaled ResNet shows worse generalization despite achieving lower training loss. Since our theory governs training

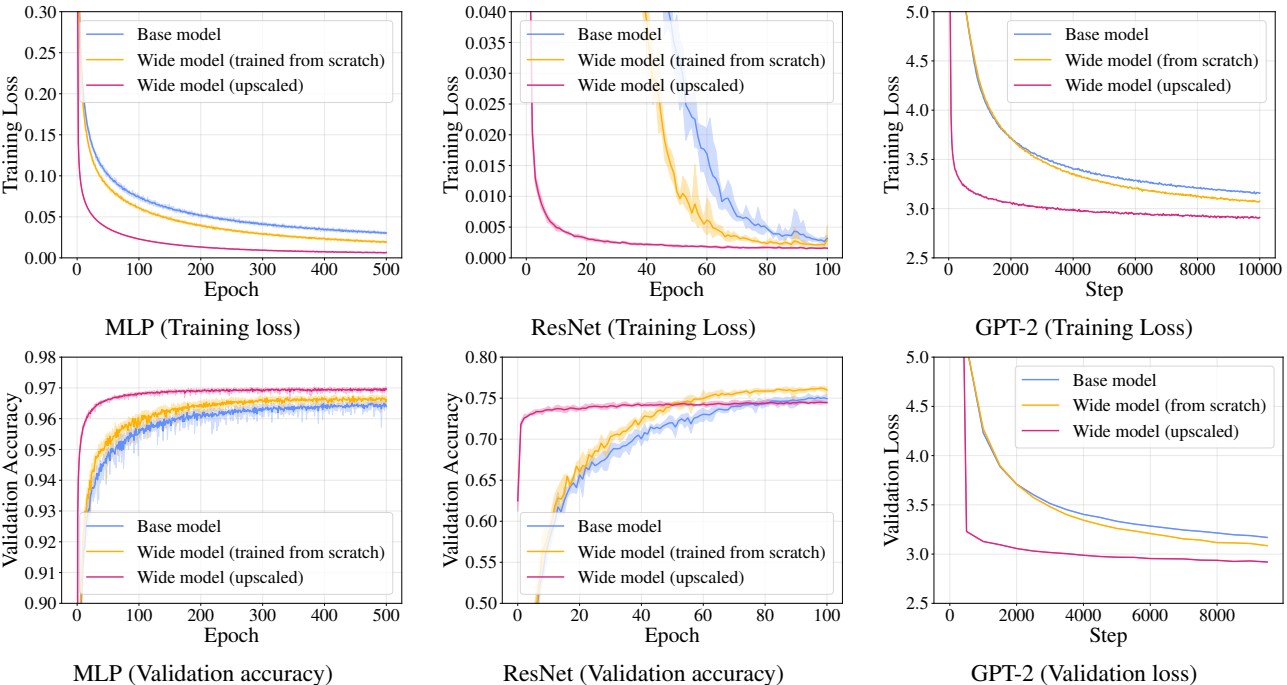

*Figure 2.* Training (top row) and validation (bottom row) performance for MLP, ResNet, and GPT-2. The y-axes are truncated to highlight differences between the curves in each panel. For the MLP and ResNet experiments which have training instability, plots show the mean over five random runs, with shaded min–max bands. More details and additional results are deferred to Appendix F.

dynamics rather than generalization, this is not a contradiction: large generalization gaps are a known property of CNN-based image classifiers, and here test performance is dominated by this gap. In practice, upscaling is most reliable when validation and training performance largely align; practitioners should exercise caution otherwise.

Our method can be directly compared with Net2Net (Chen et al., 2016), which uses a similar upscaling approach but tunes noise and learning rate directly at the expensive target width $kn$. By performing hyperparameter transfer at the small width $kn_0$, we obtain per-run FLOP speedups of $23.6\times$ for MLP ($n/n_0{=}5$), $16.0\times$ for ResNet ($n/n_0{=}4$), and $48.2\times$ for GPT-2 ($n/n_0{=}10$). These savings grow with $n/n_0$. Our experiments use moderate ratios due to limited compute, but hyperparameter transfer via $\mu$P works robustly for much larger ratios provided $n_0$ is sufficiently large (typically $\gtrsim 100$ units), so substantially greater savings could be attainable in large-scale settings.

**Verification of hyperparameter transfer.** We verify hyperparameter transfer empirically by repeating the upscaling hyperparameter sweep across a range of target widths. Figure 3 shows results for GPT-2 models upscaled by $k = 2$ from base widths $n \in \{64, 128, 256, 512\}$ to corresponding target widths $N \in \{128, 256, 512, 1024\}$. For each target width, we independently sweep the learning rate (left panel) or the injected noise magnitude (right panel) while holding

all other hyperparameters fixed, and plot the resulting terminal training loss. The optimal learning rate and noise level remain stable across target widths, confirming that values tuned on a small upscaling system transfer reliably to much larger ones, as illustrated in Figure 1b. Analogous results for MLPs are included in Appendix F.5.

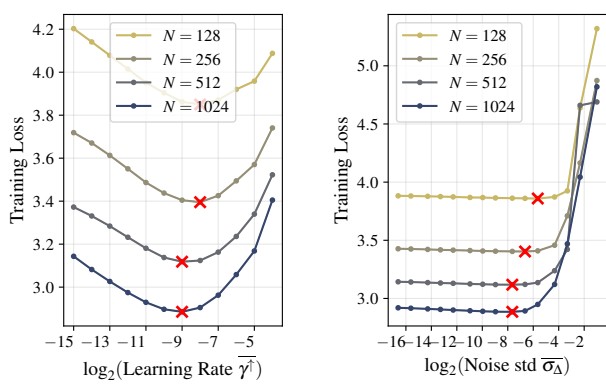

*(a)* Learning-rate sweep with fixed noise.

*(b)* Noise sweep with fixed learning rate.

*Figure 3.* Hyperparameter transfer for GPT-2 models upscaled by $k = 2$ to widths $N \in \{128, 256, 512, 1024\}$. Optimal values are stable across widths.

## Acknowledgements

YM was funded by NSF BSF 2430292 and Amazon AI fellowship. MD was partially supported by NSF awards CCF 2442615 and DMS 2502377. SV and NC were partially funded by the NSF–Simons Research Collaboration on the Mathematical and Scientific Foundations of Deep Learning (MoDL) (NSF DMS 2031985). SV was also funded by NSF CAREER 2339682, NSF CCF 2212457, and NSF BSF 2430292.

## Impact Statement

This paper presents work whose goal is to advance the field of Machine Learning. Because our contributions are primarily theoretical with preliminary empirical validation, we do not anticipate immediate broader societal impacts that must be specifically highlighted here.

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

## A. Notation

**Widening and upscaling.** Throughout the paper, $n$ denotes hidden widths. Specifically, $n_\ell$ denotes the width of the $\ell$-th layer of an MLP, while $n_{\text{in}}$ and $n_{\text{out}}$ denote the input and output dimensions of a weight matrix. We use $k$ to represent the expansion multiplier applied when "widening" or "upscaling" a base model. The hidden widths of the resulting widened or upscaled model are denoted by $N$, typically satisfying $N = kn$. Quantities associated with the widened or upscaled model (including weights and hyperparameters) are denoted with the superscript $\bullet^\uparrow$. We use $\overline{\bullet}$ to denote width-independent quantities, such as the hyperparameter base constants in $\mu$P or rescaled variables in "scaled" architectures.

**Matrices and vectors.** For a vector $x \in \mathbb{R}^{nk}$, we use $x_{(i)} = (x_i, x_{i+k}, \dots) \in \mathbb{R}^n$ for $i \in [k]$ to denote the vectors that partition $x$. Similarly, for a matrix $W \in \mathbb{R}^{nk \times nk}$, we use $W_{(i,j)} \in \mathbb{R}^{n \times n}$ for $i, j \in [k]$ to denote the blocks that partition $W$. Layer-specific quantities in an MLP are denoted by superscripts: $W^{(\ell)}$, $h^{(\ell)}$, and $x^{(\ell)}$ represent the weights, pre-activations, and post-activations of the $\ell$-th layer, respectively. Subscripts $\bullet_t$ denote quantities at a specific training step $t$.

**Kronecker product.** We use $\otimes$ to denote the Kronecker product. In particular, we use it with the all-ones matrix to express expansions that duplicate entries. For example, the diagram below shows $A \otimes \mathbf{1}_{k_{\text{out}}} \mathbf{1}_{k_{\text{in}}}^\top$ for $A \in \mathbb{R}^{n_{\text{out}} \otimes n_{\text{in}}}$:

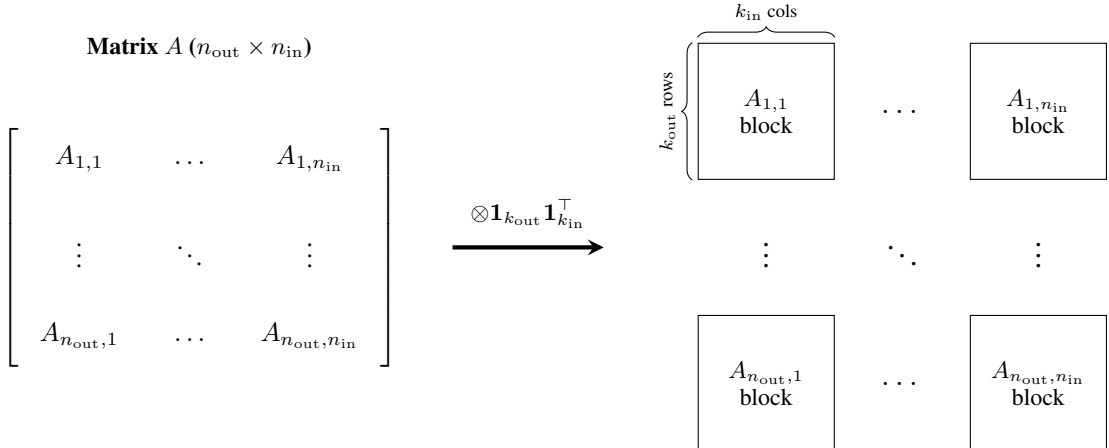

For a vector $x = (x_1, \dots, x_n)^\top \in \mathbb{R}^n$ and $k \geq 1$,

$$x \otimes \mathbf{1}_k = (\underbrace{x_1, \dots, x_1}_{k \text{ times}}, \ \dots, \ \underbrace{x_n, \dots, x_n}_{k \text{ times}})^\top \in \mathbb{R}^{nk}.$$

The Kronecker product is bilinear and associative. We will also frequently use the mixed-product property (for compatible dimensions), i.e.,

$$(A \otimes B)(C \otimes D) = (AC) \otimes (BD).$$

**Tensor Program.** Many of our theoretical results build on the Tensor Program literature, and we follow its notation. For a vector-like quantity $x$ in the Tensor Program, we write $|x\rangle$ for the random variable representing its infinite-width limit, and we decompose it as $|x\rangle = |\mathring{x}\rangle + |\grave{x}\rangle$, where $|\mathring{x}\rangle$ is the Gaussian part and $|\grave{x}\rangle$ is the correction term. For a scalar-valued quantity $c$, we denote by $\mathring{c}$ its deterministic limit as the width tends to infinity. Additional notation is introduced in Appendix E. For $x \in \mathbb{R}^n$, we sometimes denote the empirical average by

$$\langle x_\alpha \rangle_\alpha = \frac{1}{n} \sum_{\alpha \in [n]} x_\alpha.$$

For higher-order tensors $x \in (\mathbb{R}^n)^{\otimes k}$, we write

$$\langle x_{\alpha_1 \dots \alpha_k} \rangle_{\alpha_1 \dots \alpha_k} = \frac{1}{n^k} \sum_{\alpha_1, \dots, \alpha_k \in [n]} x_{\alpha_1 \dots \alpha_k}.$$

# B. Related Work

**Scaling limits of neural networks, Tensor Program, and hyperparameter transfer.** A large body of research seeks to derive tractable descriptions of neural networks in the infinite-width limit, both at initialization and during training. Early work establishes Gaussian-process descriptions at initialization (Neal, 1996; Lee et al., 2018; Matthews et al., 2018). The Neural Tangent Kernel (NTK) characterizes a lazy-training regime in which features remain nearly fixed (Jacot et al., 2018). Complementary analyses based on mean-field theory and PDEs capture aspects of feature learning beyond fixed-kernel approximations (Mei et al., 2018; Rotskoff & Vanden-Eijnden, 2022). More recently, the Tensor Program series (Yang, 2019; 2020a; Yang & Littwin, 2021; Yang, 2020b; Yang & Hu, 2021; Littwin & Yang, 2023; Yang et al., 2021; 2023b;a) provides a unified scaling-limit framework that applies across diverse architectures and optimizers and tracks dynamics in various regimes including the lazy-kernel regime and the feature learning regime. Our work builds on this framework.

The Tensor Program framework captures standard neural computations as compositions of matrix multiplications and elementwise nonlinearities, unified by the NE⊗OR⊤ program (see Definition C.3). For neural networks expressible in NE⊗OR⊤, and under appropriate scaling, training dynamics in the infinite-width limit can be characterized by deterministic state-evolution recursions that track the distributions of preactivations and model outputs over training. Additional background is provided at the beginning of Appendix E.1. Building on the NE⊗OR⊤ scaling-limit analysis, Yang and collaborators introduced the maximal-update parameterization ($\mu$P), which prescribes how weights, learning rates, initialization (and related hyperparameters) should scale with width. Within a broad class of width-dependent parameterizations, $\mu$P is, in a sense, optimal: it is the unique stable choice under which all parameters are initialized and trained with "maximal" (i.e., non-vanishing, non-exploding) updates in the infinite-width limit. Put differently, $\mu$P keeps the network in the optimal feature-learning regime, rather than drifting into a degenerate or lazy-training one. A practical advantage of $\mu$P—especially when contrasted with the standard parametrization typically paired with Kaiming initialization (He et al., 2015)—is hyperparameter transfer: hyperparameters tuned on small-width models tend to remain near-optimal as width increases, substantially reducing the cost of scaling up. Extensions of $\mu$P and hyperparameter transfer to the infinite-depth setting, as well as to joint infinite-width-and-depth regimes, have been developed and analyzed (Bordelon et al., 2024b; Yang et al., 2023b; Hayou & Yang, 2023; Bordelon et al., 2024a; Dey et al., 2026; Yao et al., 2025; Mlodozeniec et al., 2025). To the best of our knowledge, however, rigorous guarantees of hyperparameter transfer remain scarce: the only formal proof we are aware of is currently limited to linear MLPs (Hayou, 2026).

**Model upscaling.** Model upscaling refers to leveraging smaller, pretrained neural networks to initialize the training of larger models within the same architecture family, typically by increasing width and/or depth.[4] The foundational work on Net2Net (Chen et al., 2016) introduced methods for upscaling that preserve the underlying parametrized function. It includes width upscaling (Net2WiderNet), which randomly duplicates weight entries, and depth upscaling (Net2DeeperNet), which inserts identity-mapping layers. It also proposes adding a small amount of noise to break symmetry, as we do in our work. While our upscaling procedure is mechanically akin to Net2WiderNet, our contributions go substantially beyond Net2Net in three ways:

1. We extend static equivalence (function preservation at initialization) to dynamic equivalence (function preservation throughout the entire training trajectory), and rigorously establish both for a broad class of architectures and optimizers, whereas Chen et al. (2016) demonstrates only MLPs and convolutional networks.

2. Our careful choice of noise and hyperparameter scaling enables theory-grounded hyperparameter transfer, greatly reducing the tuning cost for upscaled models. By contrast, Net2WiderNet provides no tuning guidance: Chen et al. (2016) leaves the symmetry-breaking noise unspecified and reports that optimal learning rates for upscaled models are about $1/10$ of those for base models in their experiments. Such shifts necessitate costly retuning at the upscaled model size, which our framework mitigates through zero-shot transfer from small systems.

3. We establish a theoretical connection between model equivalences across widths and the infinite-width limit theory, enabling a rigorous characterization of training dynamics involving upscaling and opening the door to future theoretical analysis.

Motivated by the function-preserving philosophy, recent work has empirically explored model upscaling for large language models (LLMs) (Gong et al., 2019; Chen et al., 2022; Samragh et al., 2024; Du et al., 2024; Kim et al., 2024; Zhang et al.,

---

[4]Model upscaling is also known in the literature as knowledge transfer (Chen et al., 2016; Gong et al., 2019; Chen et al., 2022) and model growth (Wang et al., 2023; Pan et al., 2023; Du et al., 2024; Samragh et al., 2024).

2024; Mallik et al., 2024) and vision transformers (Hao et al., 2025). Several studies (Chen et al., 2022; Zhang et al., 2024) propose alternative symmetry breaking by mixing parameters from upper layers. In contrast to known function-preserving transformations, Wang et al. (2023); Pan et al. (2023) develops data-driven upscaling strategies that learn mappings from smaller pretrained models to larger ones, without guarantees on equivalence or training dynamics. Generally, these empirical efforts tend to focus on specific architectures (e.g., particular transformer variants) and rely on ad-hoc heuristics, without a unifying framework applicable to general architectures and optimizers. More importantly, hyperparameter tuning for upscaled models remains an open problem across all of these approaches: practitioners either perform costly retuning at the upscaled dimension, reuse base-model hyperparameters (often suboptimal), or apply heuristic adjustments. Among these, Mallik et al. (2024) is the only prior work that also connects model upscaling with $\mu$P, but does so at an empirical level without theoretical guarantees on hyperparameter transfer. In particular, their zero-padding-based upscaling introduces non-zero-mean initializations that fall outside the regime where the infinite-width limit theory underlying $\mu$Transfer applies, so hyperparameter transfer is not expected to hold in their setting, and is neither theoretically justified nor empirically validated. Our work takes a fundamentally different approach: we provide a theoretically grounded framework that applies to general architectures and optimizers, and rather than reusing base-model hyperparameters at the upscaled scale, we transfer hyperparameters from a small upscaling system (width $n_0 \to kn_0$) to a larger one (width $n \to kn$), with both rigorous theoretical grounding (Section 3.2 and Appendix E.1) and empirical validation.

**Any-dimensional learning.** The motivation of our work—particularly the analysis of equivalence across neural-network weight spaces with varying hidden widths in Section 2—originates from a parallel line of research: any-dimensional learning. In our work, we examine the "equivalence" between narrower and wider architectures achieved by expanding a narrow network's weights through "duplication" of entries followed by suitable rescaling. We note a related form of "cross-dimensional equivalence" that we do not pursue here: expanding a narrow model's weights by zero-padding also yields a wider model that is functionally equivalent. This type of "cross-dimensional" equivalence, via duplication or zero padding, has been used to study convex sets (Levin & Chandrasekaran, 2023) and polynomial optimization (Levin & Chandrasekaran, 2025) over varying dimensions. Similar ideas have been used for any-dimensional learning, in which inputs and outputs can be objects of arbitrary size. Architectures operating on sets, graphs, or point clouds are examples of "any-dimensional neural networks:" they use a fixed number of parameters while processing inputs of arbitrary size. Correspondingly, the underlying learning tasks are likewise defined to arbitrary sizes and sometimes require cross-size equivalence via duplication or zero-padding. Along these lines, Levin & Díaz (2024) provides a general framework for constructing equivariant, any-dimensional neural networks; Díaz et al. (2025) explores these ideas in the context of kernel methods; and Levin et al. (2026) investigates size generalization for any-dimensional neural networks. Our work extends this line of research by moving from the input/output space to the weight space, establishing cross-dimensional equivalence among network parameters.

## C. Equivalence Between Models of Different Widths

### C.1. Missing Details from Section 2.2: MLPs with General Optimizers

*Proof of Proposition 2.4.* We proceed by strong induction on the training step $t$. Suppose $W_s^{\uparrow(\ell)} = k_{\ell-1}^{-1}(W_s^{(\ell)} \otimes \mathbf{1}_{k_\ell}\mathbf{1}_{k_{\ell-1}}^\top)$ holds for all $s \leq t$. We will show that it also holds for $t+1$. First, just like in Proposition 2.2, we have

$$dW_s^{\uparrow(\ell)} = k_\ell^{-1}dW_s^{(\ell)} \otimes \mathbf{1}_{k_\ell}\mathbf{1}_{k_{\ell-1}}^\top$$

for all $s \leq t$. We perform different calculations depending on the type of weight decay involved.

*Case 1: Vanilla weight decay.* Substituting the construction $\gamma^{\uparrow(\ell)} = k_\ell^m k_{\ell-1}^{-1}\gamma^{(\ell)}$, $\lambda^{\uparrow(\ell)} = k_{\ell-1}k_\ell^{-1}\lambda^{(\ell)}$, and $\varepsilon^{\uparrow(\ell)} = k_\ell^{-1}\varepsilon^{(\ell)}$ into the weight update rule (5), we obtain

$$(W_{t+1}^{\uparrow(\ell)})_{\alpha,\beta} = (W_t^{\uparrow(\ell)})_{\alpha,\beta}$$
$$- \gamma^{\uparrow(\ell)}Q_t\left((dW_0^{\uparrow(\ell)} + \lambda^{\uparrow(\ell)}W_0^{\uparrow(\ell)})_{\alpha,\beta}, \ldots, (dW_t^{\uparrow(\ell)} + \lambda^{\uparrow(\ell)}W_t^{\uparrow(\ell)})_{\alpha,\beta}; \varepsilon^{\uparrow(\ell)}\right)$$
$$= (k_{\ell-1}^{-1}W_t^{(\ell)} \otimes \mathbf{1}_{k_\ell}\mathbf{1}_{k_{\ell-1}}^\top)_{\alpha,\beta}$$
$$- k_\ell^m k_{\ell-1}^{-1}\gamma^{(\ell)}Q_t\left(\left((k_\ell^{-1}dW_0^{(\ell)} + k_{\ell-1}k_\ell^{-1}\lambda^{(\ell)}k_{\ell-1}^{-1}W_0^{(\ell)}) \otimes \mathbf{1}_{k_\ell}\mathbf{1}_{k_{\ell-1}}^\top\right)_{\alpha,\beta}, \ldots,$$

$$\left( (k_\ell^{-1} dW_t^{(\ell)} + k_{\ell-1} k_\ell^{-1} \lambda^{(\ell)} k_{\ell-1}^{-1} W_t^{(\ell)}) \otimes \mathbf{1}_{k_\ell} \mathbf{1}_{k_{\ell-1}}^\top \right)_{\alpha,\beta} ; k_\ell^{-1} \varepsilon^{(\ell)} \right).$$

By the homogeneity of $Q_t$ (of degree $m$), this yields

$$
\begin{aligned}
&= (k_{\ell-1}^{-1} W_t^{(\ell)} \otimes \mathbf{1}_{k_\ell} \mathbf{1}_{k_{\ell-1}}^\top)_{\alpha,\beta} \\
&\quad - k_{\ell-1}^{-1} \gamma^{(\ell)} Q_t \left( \left( (dW_0^{(\ell)} + \lambda^{(\ell)} W_0^{(\ell)}) \otimes \mathbf{1}_{k_\ell} \mathbf{1}_{k_{\ell-1}}^\top \right)_{\alpha,\beta}, \dots, \right. \\
&\qquad\qquad\qquad \left. \left( (dW_t^{(\ell)} + \lambda^{(\ell)} W_t^{(\ell)}) \otimes \mathbf{1}_{k_\ell} \mathbf{1}_{k_{\ell-1}}^\top \right)_{\alpha,\beta} ; \varepsilon^{(\ell)} \right) \\
&= k_{\ell-1}^{-1} (W_{t+1}^{(\ell)} \otimes \mathbf{1}_{k_\ell} \mathbf{1}_{k_{\ell-1}}^\top)_{\alpha,\beta}.
\end{aligned}
$$

*Case 2: Decoupled weight decay.* Similarly, substituting the scalings $\gamma^{\uparrow(\ell)} = k_\ell^m k_{\ell-1}^{-1} \gamma^{(\ell)}$, $\lambda^{\uparrow(\ell)} = k_{\ell-1} k_\ell^{-m} \lambda^{(\ell)}$, and $\varepsilon^{\uparrow(\ell)} = k_\ell^{-1} \varepsilon^{(\ell)}$ into the weight update rule (6), we obtain

$$
\begin{aligned}
(W_{t+1}^{\uparrow}{}^{(\ell)})_{\alpha,\beta} &= \left( 1 - \lambda^{\uparrow(\ell)} \gamma^{\uparrow(\ell)} \right) \left( (W_t^{\uparrow}{}^{(\ell)})_{\alpha,\beta} - \gamma^{\uparrow(\ell)} Q_t \left( (dW_0^{\uparrow}{}^{(\ell)})_{\alpha,\beta}, \dots, (dW_t^{\uparrow}{}^{(\ell)})_{\alpha,\beta} ; \varepsilon^{\uparrow(\ell)} \right) \right) \\
&= (1 - \lambda^{(\ell)} \gamma^{(\ell)}) \left( (k_{\ell-1}^{-1} W_t^{(\ell)} \otimes \mathbf{1}_{k_\ell} \mathbf{1}_{k_{\ell-1}}^\top)_{\alpha,\beta} \right. \\
&\quad - k_\ell^m k_{\ell-1}^{-1} \gamma^{(\ell)} Q_t \left( (k_\ell^{-1} dW_0^{(\ell)} \otimes \mathbf{1}_{k_\ell} \mathbf{1}_{k_{\ell-1}}^\top)_{\alpha,\beta}, \dots, \right. \\
&\qquad\qquad \left. \left. (k_\ell^{-1} dW_t^{(\ell)} \otimes \mathbf{1}_{k_\ell} \mathbf{1}_{k_{\ell-1}}^\top)_{\alpha,\beta} ; k_\ell^{-1} \varepsilon^{(\ell)} \right) \right).
\end{aligned}
$$

Again, by the homogeneity of $Q_t$ (of degree $m$), this yields

$$
\begin{aligned}
&= (1 - \lambda^{(\ell)} \gamma^{(\ell)}) \left( (k_{\ell-1}^{-1} W_t^{(\ell)} \otimes \mathbf{1}_{k_\ell} \mathbf{1}_{k_{\ell-1}}^\top)_{\alpha,\beta} \right. \\
&\quad - k_{\ell-1}^{-1} \gamma^{(\ell)} Q_t \left( (dW_0^{(\ell)})_{\alpha,\beta} \otimes \mathbf{1}_{k_\ell} \mathbf{1}_{k_{\ell-1}}^\top, \dots, (dW_t^{(\ell)})_{\alpha,\beta} \otimes \mathbf{1}_{k_\ell} \mathbf{1}_{k_{\ell-1}}^\top ; \varepsilon^{(\ell)} \right) \right) \\
&= k_{\ell-1}^{-1} (W_{t+1}^{(\ell)} \otimes \mathbf{1}_{k_\ell} \mathbf{1}_{k_{\ell-1}}^\top)_{\alpha,\beta}.
\end{aligned}
$$

The induction is complete in both cases, which establishes the result. $\qquad\square$

Next, we instantiate Proposition 2.4 on concrete optimization methods. In particular, we describe how to apply this result to the implementations of these methods found in the `PyTorch` library.

**Example C.1** (SGD with and without momentum)**.** The update function of stochastic gradient descent (SGD) takes the form $Q_t(x_0, \dots, x_t) = x_t$, i.e., a degree-one homogeneous map. Hence, one should choose learning rate $\gamma^{\uparrow(\ell)} := k_\ell k_{\ell-1}^{-1} \gamma^{(\ell)}$ to ensure equivalent updates between equivalent weights. For SGD with momentum $\beta$ and dampening $\tau$, the update function is given by

$$Q_t(x_0, \dots, x_t) = (1 - \tau) \sum_{s=0}^{t} \beta^{t-s} x_s.$$

This function is again degree-one homogeneous. Hence, using the same learning rate $\gamma^{\uparrow(\ell)}$, we also obtain dynamic equivalence.Further, for SGD with Nesterov momentum, i.e., (`nesterov=True` in `PyTorch`), the update function is given by

$$Q_t(x_0, \dots, x_t) = (1 + \beta - \beta\tau) x_t + (1 - \tau) \sum_{s=0}^{t-1} \beta^{t-s+1} x_s,$$

which once more is degree-one homogeneous. So, the same conclusion applies. Finally, in `PyTorch` implementation of SGD, the weight decay is not implemented in a decoupled way as in Definition 2.3, so one should choose a weight decay constant for the widened model of

$$\lambda^{\uparrow(\ell)} := k_{\ell-1} k_\ell^{-1} \lambda^{(\ell)}.$$

**Example C.2** (Adam and AdamW). The Adam optimizer (Kingma & Ba, 2015) with hyperparameters $\beta_1, \beta_2$ is also an entrywise optimizer with update function

$$Q_t(x_0, \ldots, x_t; \varepsilon) = \frac{(1 - \beta_1^t)^{-1}(1 - \beta_1) \sum_{s=0}^t \beta_1^{t-s} x_s}{\sqrt{(1 - \beta_2^t)^{-1}(1 - \beta_2) \sum_{s=0}^t \beta_2^{t-s} x_s^2} + \varepsilon},$$

where $\varepsilon > 0$ is a small constant for numerical stability. The function $Q_t$ is a degree-zero homogeneous map. Hence, for Adam, one should use the following hyperparameters for the widened model

$$\gamma^{\uparrow(\ell)} := k_{\ell-1}^{-1} \gamma^{(\ell)}, \quad \varepsilon^{\uparrow(\ell)} := k_\ell^{-1} \varepsilon^{(\ell)}.$$

The update function of AMSGrad (Reddi et al., 2018) (`amsgrad=True` in `PyTorch`) is given by

$$Q_t(x_0, \ldots, x_t; \varepsilon) = \frac{(1 - \beta_1^t)^{-1}(1 - \beta_1) \sum_{s=0}^t \beta_1^{t-s} x_s}{\sqrt{\max(v_0(x_0), \ldots, v_t(x_0, \ldots, x_t))} + \varepsilon},$$

$$v_t(x_0, \ldots, x_t) = (1 - \beta_2^t)^{-1}(1 - \beta_2) \sum_{s=0}^t \beta_2^{t-s} x_s^2,$$

which is again degree-zero homogeneous. Hence, the same choice applies here. Once more, by default, the weight decay is not implemented in a decoupled way in the Adam optimizer in `PyTorch`, so one should choose

$$\lambda^{\uparrow(\ell)} := k_{\ell-1} k_\ell^{-1} \lambda^{(\ell)}.$$

Finally, for the AdamW optimizer, i.e., `decoupled_weight_decay=True` in `Adam`, weight decay *is* implemented in a decoupled way (Loshchilov & Hutter, 2019). Hence, one should instead use

$$\lambda^{\uparrow(\ell)} := k_{\ell-1} \lambda^{(\ell)}.$$

## C.2. Missing Details from Section 2.3: Equivalence for General Network Architectures

To extend these results to all "standard" architectures, we will use NE⊗ORᴛ programs. Appendix C.2.1 starts by reviewing the version of these programs appearing in Littwin & Yang (2023), which is more general than those in earlier papers on the subject. With this background in place, we state a formal version of Theorem 2.5 in Theorem C.5. We then present a slightly more general formulation of the same result in Appendix C.2.2, followed by a proof in Appendix C.2.3.

### C.2.1. BACKGROUND: NE⊗ORᴛ PROGRAM AND THE BACKPROPAGATION PROGRAM

We briefly review the NE⊗ORᴛ language (Definition 2.6.1 in Littwin & Yang (2023)). We introduce a minor extension that allows "width" dimensions to vary across layers. In the original NE⊗ORᴛ construction used for infinite-width analysis, assuming equal layer widths was essentially a notational convenience: because all widths are taken to diverge at the same rate, any finite discrepancies between them vanish asymptotically and can be ignored. For our purpose, we explicitly allow varying widths.

**Definition C.3** (NE⊗ORᴛ program). A NE⊗ORᴛ program is an iterative procedure that generates a sequence of vectors $\boldsymbol{x}$ and a sequence of real scalars $\boldsymbol{c}$, defined inductively from an initial collection of scalars $\boldsymbol{c}^0 \subseteq \boldsymbol{c}$, an initial collection of vectors $\boldsymbol{x}^0 \subseteq \boldsymbol{x}$, and an initial set of matrices $\mathcal{W}$, by repeatedly applying any of the following allowed operations.[5]

- `Avg`: Choose a vector $x \in \boldsymbol{x}$ of dimension $n$, and append to $\boldsymbol{c}$ a scalar

$$\langle x_\alpha \rangle_\alpha = \frac{1}{n} \sum_{\alpha=1}^n x_\alpha \in \mathbb{R}.$$

---

[5]For this section, we slightly depart from Definition 2.6.1 of Littwin & Yang (2023): The initial scalars, vectors, and matrices are taken to be deterministic, rather than randomly initialized under the prescribed rules.

- `MatMul`: Choose a matrix $W \in \mathcal{W}$ and vector $x \in \boldsymbol{x}$ with compatible dimensions, and append to $\boldsymbol{x}$ the vector

$$Wx, \quad \text{or} \quad W^{\top}x. \tag{7}$$

- `OuterNonlin`: Given integers $r \geq 0, n \in \mathbb{N}$ and a function $\psi : \mathbb{R}^{|\tilde{\boldsymbol{x}}|(r+1)+l} \to \mathbb{R}$, append to $\boldsymbol{x}$ the vector

$$y \in \mathbb{R}^n, \quad y_\alpha = \langle \psi(\tilde{\boldsymbol{x}}_\alpha; \tilde{\boldsymbol{x}}_{\beta_1}; \ldots; \tilde{\boldsymbol{x}}_{\beta_r}; \boldsymbol{c}) \rangle_{\beta_1, \ldots, \beta_r} = \frac{1}{n^r} \sum_{\beta_1, \ldots, \beta_r = 1}^{n} \psi(\tilde{\boldsymbol{x}}_\alpha; \tilde{\boldsymbol{x}}_{\beta_1}; \ldots; \tilde{\boldsymbol{x}}_{\beta_r}; \boldsymbol{c}).$$

Here, $\tilde{\boldsymbol{x}} \subseteq \boldsymbol{x}$ denotes the subset of the vectors with the same dimension $n$, and we will think of $\tilde{\boldsymbol{x}}$ as a matrix with $n$-dimensional columns. We write $\tilde{\boldsymbol{x}}_\gamma$ for the $\gamma$th row of the matrix $\tilde{\boldsymbol{x}}$, and $|\tilde{\boldsymbol{x}}|$ denotes the number of columns in the matrix $\tilde{\boldsymbol{x}}$.

We emphasize that a $\text{NE} \otimes \text{OR} \top$ program itself is merely the syntactic structure of the above transformations, not their evaluations on any particular family of scalars, vectors, and matrices. It may be viewed as specified by an abstract syntax tree, for instance, with nodes representing the above operations (together with the nonlinearities $\psi$).

Consider a neural network architecture parametrizing a function $\mathbb{R}^{d_{\text{in}}} \to \mathbb{R}^{d_{\text{out}}}$ with a weight space consisting of $\ell$ matrices, $m$ vectors, and $j$ scalar weights. Suppose the matrices have dimensions $n_{1,\text{out}} \times n_{1,\text{in}}, \ldots, n_{\ell,\text{out}} \times n_{\ell,\text{in}}$, and the vectors have dimensions $n_1, \ldots, n_m$. We denote the weight space as

$$\mathcal{T}_{\boldsymbol{n}} = (\mathbb{R})^j \oplus \left( \bigoplus_{i=1}^{m} \mathbb{R}^{n_i} \right) \oplus \left( \bigoplus_{i=1}^{\ell} \mathbb{R}^{n_{i,\text{out}} \times n_{i,\text{in}}} \right),$$

which is indexed by the "width vector" that collects the hidden widths of the weights.

$$\boldsymbol{n} = (n_1, \ldots, n_m, n_{1,\text{in}}, n_{1,\text{out}}, \ldots, n_{\ell,\text{in}}, n_{\ell,\text{out}}).$$

A $\text{NE} \otimes \text{OR} \top$ program $\pi$ *represents* this neural network architecture if the following properties hold. First, $\pi$ starts with the initial set of scalars $\boldsymbol{c}_0$ consisting of the (initialized) $j$ scalar weights and the input of dimension $d_{\text{in}}$, the initial set of vectors $\boldsymbol{x}_0$ consisting of the (initialized) $m$ vector weights, and the initial set of matrices $\mathcal{W}$ consisting of the (initialized) $\ell$ matrix weights in $\mathcal{T}_{\boldsymbol{n}}$. The $\text{NE} \otimes \text{OR} \top$ program then describes the computations of all intermediate values in the architecture's forward pass. At the end, it picks vectors $(x^1, \ldots, x^{d_{\text{out}}})$ from the final set of vectors $\boldsymbol{x}$, and outputs $y \in \mathbb{R}^{d_{\text{out}}}$ with entries

$$y_i = \sum_\alpha x^i_\alpha, \quad i = 1, \ldots, d_{\text{out}}. \tag{8}$$

Appendix A of Yang (2019) shows that many common neural network components—BatchNorm, skip connections, convolution, pooling, GRU, LSTM, layer normalization, and scaled attention—are expressible in $\text{NE} \otimes \text{OR} \top$, and by composing these we see that $\text{NE} \otimes \text{OR} \top$ programs can represent standard CNN, RNN, and Transformer architectures.

Given a program $\pi$ representing an architecture, Definition 2.9.14 of Littwin & Yang (2023) shows that one can automatically construct another $\text{NE} \otimes \text{OR} \top$ program for backpropagation to compute all of the gradient vectors with respect to $x$ needed to perform gradient updates. A few times here and in the discussion below, we will talk about automatically building a new $\text{NE} \otimes \text{OR} \top$ program from a given one. Since we view $\text{NE} \otimes \text{OR} \top$ as a formal programming language, this kind of procedure should be thought of as akin to compilation of ordinary computer programs: we perform automated transformations on a $\text{NE} \otimes \text{OR} \top$ program to turn it into another $\text{NE} \otimes \text{OR} \top$ program, perhaps having different operational semantics.

**Definition C.4** (Backpropagation program). Consider any $\text{NE} \otimes \text{OR} \top$ program $\pi$ and a vector $x \in \mathbb{R}^{n_x}$ in $\pi$. Then $\pi$'s *backpropagation program* with respect to $x$ is an extension of $\pi$ defined by constructing the following objects on top of $\pi$: (Intuitively, one should interpret $d^x y = n_y \frac{\partial \langle x_\alpha \rangle_\alpha}{\partial y}$ if $y \in \mathbb{R}^{n_y}$ is a vector and $d^x c = \frac{\partial \langle x_\alpha \rangle_\alpha}{\partial c}$ if $c$ is a scalar.)

- $d^x x := \mathbf{1}_{n_x} \in \mathbb{R}^{n_x}$.

- For any `MatMul` instruction $z := Wy$ in $\pi$, we construct

$$d^{x|z} y := W^{\top} d^x z \text{ (via another MatMul).} \tag{9}$$

- For any `Avg` instruction $c := \langle x_\alpha \rangle_\alpha$ in $\pi$, we construct

$$d^{x|c}z := (d^x c)\mathbf{1}_{n_z} \in \mathbb{R}^{n_z} \text{ (via OuterNonlin)}.$$

- For any `OuterNonlin` instruction $y := \langle \psi(\tilde{\boldsymbol{x}}; \tilde{\boldsymbol{x}}_{\beta_1}, \dots, \tilde{\boldsymbol{x}}_{\beta_r}; \boldsymbol{c}) \rangle_{\beta_1, \dots, \beta_r}$, for each $i = 0, \dots, r$, let

$$\mathbf{g}^i_{\beta_0 \dots \beta_r} := d^x y \psi_i(\tilde{\boldsymbol{x}}_{\beta_0}, \dots, \tilde{\boldsymbol{x}}_{\beta_r}; \boldsymbol{c}) \in \mathbb{R}^{|\tilde{\boldsymbol{x}}|},$$

where $\psi_i : \mathbb{R}^{|\tilde{\boldsymbol{x}}|(r+1)+\ell} \to \mathbb{R}^{|\tilde{\boldsymbol{x}}|}$ yields the derivative of $\psi$ with respect to $x$ in the $i$-th slot. When $i = r+1$, we make the analogous definition for $\mathbf{g}^{r+1}_{\beta_0 \dots \beta_r} \in \mathbb{R}^{|\boldsymbol{c}|}$. We write $\boldsymbol{\beta} = (\beta_0, \dots, \beta_r)$, $\boldsymbol{\beta}[i \mapsto \alpha] = (\beta_0, \dots, \beta_{i-1}, \alpha, \beta_{i+1}, \dots, \beta_r)$, and $\boldsymbol{\beta}_{-i} = (\beta_0, \dots, \beta_{i-1}, \beta_{i+1}, \dots, \beta_r)$. Then we construct $d^{x|y}\boldsymbol{c} = (d^{x|y}c^1, \dots, d^{x|y}c^{|c|})$ and $d^{x|y}\boldsymbol{x} = (d^{x|y}x^1, \dots, d^{x|y}x^{|\boldsymbol{x}|})$ by

$$d^{x|y}\boldsymbol{c} := \langle \mathbf{g}^{r+1}_{\boldsymbol{\beta}} \rangle_{\boldsymbol{\beta}} \in \mathbb{R}^{|\boldsymbol{c}|} \quad \text{(using OuterNonlin and Avg)}$$

$$d^{x|y}\boldsymbol{x}_\alpha := \sum_{i=0}^{r} \langle \mathbf{g}^i_{\boldsymbol{\beta}[i \mapsto \alpha]} \rangle_{\boldsymbol{\beta}_{-i}} \in \mathbb{R}^{|\boldsymbol{x}|} \quad \text{(using OuterNonlin)}.$$

Explicitly,

$$d^{x|y}\boldsymbol{x}_\alpha = \langle \mathbf{g}^0_{\alpha\beta_1 \dots \beta_r} \rangle_{\beta_1 \dots \beta_r} + \langle \mathbf{g}^1_{\beta_0 \alpha \beta_2 \dots \beta_r} \rangle_{\beta_0 \beta_2 \dots \beta_r} + \dots + \langle \mathbf{g}^r_{\beta_0 \dots \beta_{r-1} \alpha} \rangle_{\beta_0 \dots \beta_{r-1}}.$$

- Finally, for every vector or scalar $y$ in $\pi$ other than $x$,

$$d^x y := \sum_u d^{x|u} y$$

where $u$ ranges over all vectors or scalars in $\pi$ whose construction used $y$.

The subprogram that constructs all of these new objects is denoted $d^x \pi$. For a $\text{NE} \otimes \text{OR} \top$ program $\pi$ representing a neural-network architecture, backpropagation involves the subprograms $d^{x^1}\pi, \dots, d^{x^{d_{\text{out}}}}\pi$, where $(x^1, \dots, x^{d_{\text{out}}})$ are vectors used to generate the output in (8).

With this background, we now write the formal version of Theorem 2.5, that we will proceed to prove in the rest of this section.

**Theorem C.5** (Formal version of Theorem 2.5). *Consider a neural network architecture represented by a $\text{NE} \otimes \text{OR} \top$ program $\pi$ satisfying the following two rules.*

1. *Its forward pass only uses matrix multiplication of the form $Wx$, not of the form $W^\top x$.*

2. *Its readout rule (8) replaces the sum by the average, i.e., $y_i = \langle x^i_\alpha \rangle_\alpha$.*

*Assume an entrywise optimizer whose update function is homogeneous of degree $m$. Consider a base model under this architecture, with widths $\boldsymbol{n} = (n_1, \dots, n_m, n_{1,\text{in}}, n_{1,\text{out}}, \dots, n_{\ell,\text{in}}, n_{\ell,\text{out}})$ and initial weights*

$$\Theta = \big( \underbrace{\theta_1, \dots, \theta_j}_{\text{scalars}}, \underbrace{x_1, \dots, x_m}_{\text{vectors}}, \underbrace{W_1, \dots, W_\ell}_{\text{matrices}} \big) \in \mathcal{T}_{\boldsymbol{n}}.$$

*It is trained with the entrywise optimizer with learning rate $\gamma$, weight decay constant $\lambda$, and additional hyperparameter $\varepsilon$.*

*Construct a widened model under the same architecture with widths*

$$\boldsymbol{N} = \boldsymbol{k} \odot \boldsymbol{n} = (k_1 n_1, \dots, k_m n_m, k_{1,\text{in}} n_{1,\text{in}}, k_{1,\text{out}} n_{1,\text{out}}, \dots, k_{\ell,\text{in}} n_{\ell,\text{in}}, k_{\ell,\text{out}} n_{\ell,\text{out}}),$$

*where the width multipliers are summarized in the vector $\boldsymbol{k} = (k_1, \dots, k_m, k_{1,\text{in}}, k_{1,\text{out}}, \dots, k_{\ell,\text{in}}, k_{\ell,\text{out}})$. Its weights $\Theta^\uparrow \in \mathcal{T}_{\boldsymbol{N}}$ are constructed using the rule in the "Widening operation" column of Table 1. That is,*

$$\Theta^\uparrow := \big( \underbrace{\theta_1, \dots, \theta_j}_{\text{scalars}}, \underbrace{x_1 \otimes \mathbf{1}_{k_1}, \dots, x_m \otimes \mathbf{1}_{k_m}}_{\text{vectors}}, \underbrace{k_{1,\text{in}}^{-1} W_1 \otimes \mathbf{1}_{k_{1,\text{out}}} \mathbf{1}_{k_{1,\text{in}}}^\top, \dots, k_{\ell,\text{in}}^{-1} W_\ell \otimes \mathbf{1}_{k_{\ell,\text{out}}} \mathbf{1}_{k_{\ell,\text{in}}}^\top}_{\text{matrices}} \big). \quad (10)$$

*The widened model is trained with the same optimizer using per-weight hyperparameters given by the "Hyperparameters" column of Table 1.*

*Then, at all training steps, both models parametrize the same function.*

### C.2.2. Formulation with scaled architecture

Theorem C.5 does not apply to architectures whose forward pass simultaneously uses both $Wx$ and $W^\top x$. This restriction is typically harmless: allowing both $W$ and $W^\top$ in the definition of the NE⊗OR⊤ program was intended to capture backpropagation. (If $y = Wx$ appears in the forward pass, then $dx = W^\top dy$ naturally appears in backpropagation.) Standard architectures do not employ $W^\top$ in the forward pass. However, extending the result to architectures that include both operations is straightforward: for the $Wx$ operation use $k_{\text{in}}^{-1} W \otimes \mathbf{1}\mathbf{1}^\top$, and for the $W^\top x$ operation use $k_{\text{out}}^{-1} W^\top \otimes \mathbf{1}\mathbf{1}^\top$, i.e., adopt distinct scaling rules for $W$ and $W^\top$. We will formally state the result in this more general setting (see Theorem C.8) because it contains insights of independent interest.

We begin with observing an equivalent formulation of Proposition 2.4 for MLPs. This formulation rescales the MLP weights so that constructing an equivalent MLP is width-independent—duplicate the weights with no further rescaling.

Define a "scaled" MLP that maps $x^{(0)}$ to $x^{(L)}$ via the recursion

$$h^{(\ell)} = n_{\ell-1}^{-1} W^{(\ell)} x^{(\ell-1)} \in \mathbb{R}^{n_\ell},$$
$$x^{(\ell)} = \phi(h^{(\ell)}) \in \mathbb{R}^{n_\ell}, \quad \text{for } \ell = 1, 2, \ldots, L.$$

We train the weights $n_{\ell-1}^{-1} W^{(\ell)}$ using the entrywise optimizer, where we adopt $\mu$P layer-wise hyperparameters when updating each $n_{\ell-1}^{-1} W^{(\ell)}$ as in Table 2, i.e.,

$$\gamma^{(\ell)} := n_\ell^m n_{\ell-1}^{-1} \overline{\gamma},$$
$$\varepsilon^{(\ell)} := n_\ell^{-1} \overline{\varepsilon},$$
$$\lambda^{(\ell)} := \begin{cases} n_{\ell-1} n_\ell^{-1} \overline{\lambda} & \text{for vanilla weight decay,} \\ n_{\ell-1} n_\ell^{-m} \overline{\lambda} & \text{for decoupled weight decay.} \end{cases}$$

Here $\overline{\gamma}, \overline{\varepsilon}, \overline{\lambda}$ denote the base learning rate, base auxiliary hyperparameter, and base weight decay constant, respectively, and the actual hyperparameters are scaled according to the layer widths.

**Corollary C.6** (Alternative statement of Proposition 2.4). *Consider a base "scaled" MLP with weight matrices $\big(W^{(\ell)} \in \mathbb{R}^{n_\ell \times n_{\ell-1}}\big)_{\ell=1}^{L}$. Construct a widened "scaled" MLP that uses the same activation function and preserves the input and output dimensions, with weights obtained by duplicating those of the base MLP as*

$$W^{\uparrow(\ell)} := W^{(\ell)} \otimes \mathbf{1}_{k_\ell} \mathbf{1}_{k_{\ell-1}}^\top \in \mathbb{R}^{N_\ell \times N_{\ell-1}} \quad \text{for } \ell = 1, \ldots, L.$$

*Here $N_\ell = k_\ell n_\ell$ for all $\ell$, and the width multipliers $k_\ell \in \mathbb{N}$ satisfy $k_0 = k_L = 1$. Suppose both models are trained with the same base hyperparameter constants $\overline{\gamma}, \overline{\varepsilon}, \overline{\lambda}$ using the entrywise optimizer, with layer-wise hyperparameters scaled accordingly. Then, at all training steps, both models parametrize the same function.*

*Proof.* By Proposition 2.4, the widened "scaled" MLP parametrizes the same function at each training step if the following holds:

$$N_{\ell-1}^{-1} W^{\uparrow(\ell)} = k_{\ell-1}^{-1}\big(n_{\ell-1}^{-1} W^{(\ell)} \otimes \mathbf{1}_{k_\ell} \mathbf{1}_{k_{\ell-1}}^\top\big),$$
$$(N_\ell^m N_{\ell-1}^{-1})\overline{\gamma} = k_\ell^m k_{\ell-1}^{-1}(n_\ell^m n_{\ell-1}^{-1})\overline{\gamma},$$
$$(N_\ell^{-1})\overline{\varepsilon} = k_\ell^{-1}(n_\ell^{-1})\overline{\varepsilon},$$
$$(N_{\ell-1} N_\ell^{-1})\overline{\lambda} = k_{\ell-1} k_\ell^{-1}(n_{\ell-1} n_\ell^{-1})\overline{\lambda} \quad \text{for vanilla weight decay,}$$
$$(N_{\ell-1} N_\ell^{-m})\overline{\lambda} = k_{\ell-1} k_\ell^{-m}(n_{\ell-1} n_\ell^{-m})\overline{\lambda} \quad \text{for decoupled weight decay.}$$

These equalities do hold exactly, establishing the claim. $\qquad \square$

To extend these results to general architectures, we first define a "scaled" architecture in which weights are rescaled with respect to width.

**Definition C.7** (Scaled architecture). A *scaled* NE⊗OR⊤ *program* is the same as the NE⊗OR⊤ program in Definition C.3, with the modification that in each `MatMul` operation (7) involving $W \in \mathbb{R}^{n_{\text{out}} \times n_{\text{in}}}$,

$$Wx \text{ is replaced by } \frac{1}{n_{\text{in}}}Wx, \quad \text{and} \quad W^{\top}x \text{ is replaced by } \frac{1}{n_{\text{out}}}W^{\top}x.$$

A *scaled neural network architecture* represented by a scaled NE⊗OR⊤ program $\pi$ is defined in the same way as before, except that at the final readout step (8), the output $y \in \mathbb{R}^{d_{\text{out}}}$ is given by

$$y_i = \langle x^i_\alpha \rangle_\alpha = \frac{1}{n^{(i)}} \sum_{\alpha=1}^{n^{(i)}} x^i_\alpha, \qquad i = 1, \ldots, d_{\text{out}}, \tag{11}$$

where each chosen vector $x^i \in \mathbb{R}^{n^{(i)}}$. The modification is that we replace the sum with the mean. Because the scaled architecture can be instantiated at multiple hidden widths, it parametrizes a family of functions

$$f_{\boldsymbol{n}} \colon \mathbb{R}^{d_{\text{in}}} \times \mathcal{T}_{\boldsymbol{n}} \to \mathbb{R}^{d_{\text{out}}},$$

indexed by the hidden-width vector $\boldsymbol{n} = (n_1, \ldots, n_m, n_{1,\text{in}}, n_{1,\text{out}}, \ldots, n_{\ell,\text{in}}, n_{\ell,\text{out}})$.

With this modification, the *scaled backpropagation program* is adjusted accordingly from Definition C.4. For any `MatMul` instruction $z := n_{\text{in}}^{-1}Wy$ in $\pi$, the construction of the gradient (9) is replaced by $d^{x|z}y := n_{\text{out}}^{-1}W^{\top}d^x z$ (via another `MatMul`). Notice that the modified backpropagation program is then a scaled NE⊗OR⊤ program.

With this definition in place, we state the general result that we will prove. One can verify that Theorem C.5 is a special case of the following theorem in which the forward pass does not use both $W$ and $W^{\top}$.

**Theorem C.8.** *Consider a scaled architecture represented by a scaled* NE⊗OR⊤ *program $\pi$, and an entrywise optimizer whose update function is homogeneous of degree $m$. Adopt the $\mu$P scaling for its hyperparameters as in Table 2. Consider a base model under this scaled architecture with widths $\boldsymbol{n} = (n_1, \ldots, n_m, n_{1,\text{in}}, n_{1,\text{out}}, \ldots, n_{\ell,\text{in}}, n_{\ell,\text{out}})$ and initial weights*

$$\Theta = \big( \underbrace{\theta_1, \ldots, \theta_j}_{\text{scalars}}, \underbrace{x_1, \ldots, x_m}_{\text{vectors}}, \underbrace{W_1, \ldots, W_\ell}_{\text{matrices}} \big) \in \mathcal{T}_{\boldsymbol{n}}.$$

*Construct a widened model under the same scaled architecture with widths*

$$\boldsymbol{N} = \boldsymbol{k} \odot \boldsymbol{n} = (k_1 n_1, \ldots, k_m n_m, k_{1,\text{in}} n_{1,\text{in}}, k_{1,\text{out}} n_{1,\text{out}}, \ldots, k_{\ell,\text{in}} n_{\ell,\text{in}}, k_{\ell,\text{out}} n_{\ell,\text{out}}),$$

*where the width multipliers are summarized in the vector $\boldsymbol{k} = (k_1, \ldots, k_m, k_{1,\text{in}}, k_{1,\text{out}}, \ldots, k_{\ell,\text{in}}, k_{\ell,\text{out}})$. Its weights $\Theta^{\uparrow} \in \mathcal{T}_{\boldsymbol{N}}$ are constructed as*

$$\Theta^{\uparrow} := \big( \underbrace{\theta_1, \ldots, \theta_j}_{\text{scalars}}, \underbrace{x_1 \otimes \mathbf{1}_{k_1}, \ldots, x_m \otimes \mathbf{1}_{k_m}}_{\text{vectors}}, \underbrace{W_1 \otimes \mathbf{1}_{k_{1,\text{out}}} \mathbf{1}_{k_{1,\text{in}}}^{\top}, \ldots, W_\ell \otimes \mathbf{1}_{k_{\ell,\text{out}}} \mathbf{1}_{k_{\ell,\text{in}}}^{\top}}_{\text{matrices}} \big).$$

*Both the base model and the widened model are trained with the entrywise optimizer using the same base hyperparameter constants $\overline{\gamma}, \overline{\varepsilon}, \overline{\lambda}$, with per-weight hyperparameters scaled as specified.*

*Then, at every training step, both models parametrize the same function. That is, for any training step and any input $\xi \in \mathbb{R}^{d_{\text{in}}}$,*

$$f_n(\xi; \Theta) = f_N(\xi; \Theta^{\uparrow}).$$

### C.2.3. PROOF OF THEOREM C.8

We begin by establishing a useful property of the scaled NE⊗OR⊤ program. Consider two runs: one initialized with a set of scalars, vectors, and matrices, and another initialized with the same scalars but with the vectors and matrices widened by duplicating entries. In the second run, the vectors and matrices generated by the program are exactly widened duplications of those obtained in the first run, while the scalars remain unchanged.

**Lemma C.9.** *Consider a scaled* NE⊗OR⊤ *program* $\pi$ *with an initial set* $c^0 \subseteq c$ *of scalars, an initial set* $x^0 \subseteq x$ *of vectors, and an initial set* $\mathcal{W}$ *of matrices. The same scaled* NE⊗OR⊤ *program can be instantiated on the initial set* $c^0 \subseteq c$ *of scalars, the widened initial set of vectors* $x^{\uparrow^0} := \{x \otimes \mathbf{1} : x \in x^0\} \subseteq x$, *and the widened initial set of matrices* $\mathcal{W}^{\uparrow} := \{W \otimes \mathbf{1}_{k_{\text{out}}} \mathbf{1}_{k_{\text{in}}}^{\top} : W \in \mathcal{W}\}$. *Let the two runs of the program generate sequences of vectors and scalars* $(x, c)$ *and* $(x^{\uparrow}, c^{\uparrow})$, *respectively. Then* $c^{\uparrow} = c$, *and* $x^{\uparrow} = \{x \otimes \mathbf{1} : x \in x\}$.

*Proof.* We show inductively that, in each step of the program, if a scalar is generated, then in both runs, the generated scalars are the same; if a vector is generated, then the vector $x^{\uparrow}$ generated in the second run and the vector $x$ generated in the first run are related by $x^{\uparrow} = x \otimes \mathbf{1}$. We consider the cases for different operations of the program.

Avg: Let the chosen vector be $x \in \mathbb{R}^n$ in the first run, corresponding to $x^{\uparrow} = x \otimes \mathbf{1} \in \mathbb{R}^N$ in the second run. Then the Avg operation generates the same scalar for the two models

$$\frac{1}{n} \sum_{\alpha=1}^{n} x_{\alpha} = \frac{1}{N} \sum_{\alpha=1}^{N} x_{\alpha}^{\uparrow} \in \mathbb{R}.$$

MatMul: Without loss of generality, consider the operation $n_{\text{in}}^{-1} W x$. The case $n_{\text{out}}^{-1} W^{\top} x$ follows in exactly the same manner. Let the chosen matrix and vector be $W \in \mathbb{R}^{n_{\text{out}} \times n_{\text{in}}}, x \in \mathbb{R}^{n_{\text{in}}}$ in the first run, corresponding to $W^{\uparrow} = W \otimes \mathbf{1}_{k_{\text{out}}} \mathbf{1}_{k_{\text{in}}}^{\top} \in \mathbb{R}^{k_{\text{out}} n_{\text{out}} \times k_{\text{in}} n_{\text{in}}}, x^{\uparrow} = x \otimes \mathbf{1}_{k_{\text{in}}} \in \mathbb{R}^{k_{\text{in}} n_{\text{in}}}$ in the second run. Then the MatMul operation in the first run generates the vector

$$n_{\text{in}}^{-1} W x,$$

and in the widened model generates the vector

$$(k_{\text{in}} n_{\text{in}})^{-1} (W \otimes \mathbf{1}_{k_{\text{out}}} \mathbf{1}_{k_{\text{in}}}^{\top})(x \otimes \mathbf{1}_{k_{\text{in}}}) = n_{\text{in}}^{-1} W x \otimes \mathbf{1}_{k_{\text{out}}}.$$

OuterNonlin: Suppose the subset of vectors $\tilde{x} \in \mathbb{R}^{n \times |\tilde{x}|}$ in the first run corresponds to $\tilde{x}^{\uparrow} = \tilde{x} \otimes \mathbf{1}_k \in \mathbb{R}^{kn \times |\tilde{x}'|}$ in the second run. Then the OuterNonlin operation generates $y^{\uparrow} \in \mathbb{R}^{kn}$ in the second run with

$$
\begin{aligned}
y_{\alpha}^{\uparrow} &= \frac{1}{(kn)^r} \sum_{\beta_1, \ldots, \beta_r=1}^{kn} \psi\left(\tilde{x}_{\alpha}^{\uparrow}; \tilde{x}_{\beta_1}^{\uparrow}; \ldots; \tilde{x}_{\beta_r}^{\uparrow}; c\right) \\
&= \frac{1}{n^r} \sum_{\beta_1, \ldots, \beta_r=1}^{n} \psi\left(\tilde{x}_{\lceil \alpha/k \rceil}; \tilde{x}_{\beta_1}; \ldots; \tilde{x}_{\beta_r}; c\right) = y_{\lceil \alpha/k \rceil}.
\end{aligned}
$$

where $y$ is the vector generated in the first run. Hence, we have $y^{\uparrow} = y \otimes \mathbf{1}_k$. $\square$

Applying Lemma C.9 to both the forward-pass and the backpropagation program suffices to complete the proof.

*Proof of Theorem C.8.* We prove the claim by strong induction on training steps $t$.

At $t = 0$, apply Lemma C.9 to the scaled NE⊗OR⊤ program $\pi$ that specifies the architecture, instantiated once with the base weights and once with the widened weights. At the final readout (11), the program selects vectors $x^1, \ldots, x^{d_{\text{out}}}$ from the set of computed vectors $x$ to form the output. By Lemma C.9, the corresponding vectors in the widened model are $(x^i \otimes \mathbf{1}_{k^{(i)}})_{i=1}^{d_{\text{out}}}$, where $k^{(i)}$ is the width multiplier of $x^i \in \mathbb{R}^{n^{(i)}}$. Since averaging is invariant under duplication, for every $i \in [d_{\text{out}}]$ we have

$$\frac{1}{n^{(i)}} \sum_{\alpha=1}^{n^{(i)}} x_{\alpha}^i = \frac{1}{n^{(i)} k^{(i)}} \sum_{\alpha=1}^{n^{(i)} k^{(i)}} (x^i \otimes \mathbf{1}_k)_{\alpha},$$

implying $f_n(\xi; \Theta) = f_N(\xi; \Theta^{\uparrow})$ at $t = 0$.

For the induction step, apply Lemma C.9 to the scaled backpropagation programs $dx^1 \pi, \ldots, dx^{d_{\text{out}}} \pi$. For each $i \in [d_{\text{out}}]$ and for each vector $u \in \mathbb{R}^{n_u}$ in $\pi$ with widened counterpart $u^{\uparrow} = u \otimes \mathbf{1}_{k_u}$, the generated backpropagated gradients satisfy

$$d^{x^i} u^{\uparrow} = d^{x^i} u \otimes \mathbf{1}_{k_u} \tag{12}$$

For each scalar $c$ in $\pi$, we similarly have

$$d^{x^i} c^{\uparrow} = d^{x^i} c. \tag{13}$$

Let $W$ denote any matrix weight, $u$ any vector weight, and $c$ any scalar weight. Write $y \in \mathbb{R}^{d_{\text{out}}}$ for the model output computed by $y_i = \langle x_{\alpha}^i \rangle_{\alpha}$ according to (11). The gradients used by the optimizer can be written as

$$
\begin{aligned}
dW = \frac{\partial \mathcal{L}}{\partial W} &= \sum_{j \in d_{\text{out}}} \frac{\partial \mathcal{L}}{\partial y_j} \frac{\partial y_j}{\partial W} \\
&= \sum_{j \in d_{\text{out}}} \frac{\partial \mathcal{L}}{\partial y_j} \left( \sum_{z = n_{\text{in}}^{-1} W h} n_{\text{in}}^{-1} \frac{\partial \langle x_{\alpha}^j \rangle_{\alpha}}{\partial z} h^{\top} + \sum_{z = n_{\text{out}}^{-1} W^{\top} h} n_{\text{out}}^{-1} \left( \frac{\partial \langle x_{\alpha}^j \rangle_{\alpha}}{\partial z} h^{\top} \right)^{\top} \right) \\
&= \sum_{j \in d_{\text{out}}} \frac{\partial \mathcal{L}}{\partial y_j} \left( \sum_{z = n_{\text{in}}^{-1} W h} n_{\text{in}}^{-1} n_{\text{out}}^{-1} (d^{x^j} z) h^{\top} + \sum_{z = n_{\text{out}}^{-1} W^{\top} h} n_{\text{out}}^{-1} n_{\text{in}}^{-1} h (d^{x^j} z)^{\top} \right)
\end{aligned}
$$

since $d^{x^j} z = n_{\text{out}} \frac{\partial \langle x_{\alpha}^j \rangle_{\alpha}}{\partial z_t}$ for $z = n_{\text{in}}^{-1} W h$. Continuing,

$$
du = \frac{\partial \mathcal{L}}{\partial u} = \sum_{j \in d_{\text{out}}} \frac{\partial \mathcal{L}}{\partial y_j} \frac{\partial y_j}{\partial u} = \sum_{j \in d_{\text{out}}} \frac{\partial \mathcal{L}}{\partial y_j} n_u^{-1} d^{x^j} u,
$$

$$
dc = \frac{\partial \mathcal{L}}{\partial c} = \sum_{j \in d_{\text{out}}} d^{x^j} c.
$$

By (12) and (13), the gradients in the widened model and the base model satisfy the relation

$$
N_{\text{in}} N_{\text{out}} \, dW^{\uparrow} = n_{\text{in}} n_{\text{out}} \, dW \otimes \mathbf{1}_{k_{\text{out}}} \mathbf{1}_{k_{\text{in}}}^{\top}, \qquad N_u \, du^{\uparrow} = n_u \, du \otimes \mathbf{1}_{k_y}, \qquad dc^{\uparrow} = dc.
$$

Finally, apply the entrywise optimizer to update $(n_{\text{in}}^{-1} W, u, c)$ in the base model and $(N_{\text{in}}^{-1} W^{\uparrow}, u^{\uparrow}, c^{\uparrow})$ in the widened model, with per-weight hyperparameters scaled as specified by $\mu$P. Here we give the proof for vanilla weight decay; the proof for decoupled weight decay is similar, so we do not include it here.

For the matrix weight $W$, since the gradient of $n_{\text{in}}^{-1} W$ is $n_{\text{in}} dW$, the update rule in the base model is

$$
\begin{aligned}
&(n_{\text{in}}^{-1} W_{t+1})_{\alpha, \beta} \\
&= (n_{\text{in}}^{-1} W_t)_{\alpha, \beta} - \gamma Q_t \Big( (n_{\text{in}} dW_0 + \lambda n_{\text{in}}^{-1} W_0)_{\alpha, \beta}, \ldots, (n_{\text{in}} dW_t + \lambda n_{\text{in}}^{-1} W_t)_{\alpha, \beta}; \varepsilon \Big)
\end{aligned}
$$

Substituting the $\mu$P for $\gamma, \lambda, \varepsilon$ and the duplication relations for $dW$ and $W$, we have by homogeneity of $Q_t$ of degree $m$

$$
\begin{aligned}
&= (n_{\text{in}}^{-1} W_t)_{\alpha, \beta} - n_{\text{out}}^m n_{\text{in}}^{-1} \overline{\gamma} \, Q_t \Big( (n_{\text{in}} dW_0 + n_{\text{in}} n_{\text{out}}^{-1} \overline{\lambda} n_{\text{in}}^{-1} W_0)_{\alpha, \beta}, \ldots, \\
&\qquad\qquad\qquad\qquad (n_{\text{in}} dW_t + n_{\text{in}} n_{\text{out}}^{-1} \overline{\lambda} n_{\text{in}}^{-1} W_t)_{\alpha, \beta}; \, n_{\text{out}}^{-1} \overline{\varepsilon} \Big) \\
&= (n_{\text{in}}^{-1} W_t)_{\alpha, \beta} - n_{\text{in}}^{-1} \overline{\gamma} \, Q_t \Big( (n_{\text{in}} n_{\text{out}} dW_0 + \overline{\lambda} W_0)_{\alpha, \beta}, \ldots, (n_{\text{in}} n_{\text{out}} dW_t + \overline{\lambda} W_t)_{\alpha, \beta}; \, \overline{\varepsilon} \Big)
\end{aligned}
$$

Equivalently,

$$
(W_{t+1})_{\alpha, \beta} = (W_t)_{\alpha, \beta} - \overline{\gamma} Q_t \Big( (n_{\text{in}} n_{\text{out}} dW_0 + \overline{\lambda} W_0)_{\alpha, \beta}, \ldots, (n_{\text{in}} n_{\text{out}} dW_t + \overline{\lambda} W_t)_{\alpha, \beta}; \, \overline{\varepsilon} \Big).
$$

Similarly, the update rule for the matrix weight in the widened model is

$$
(W_{t+1}^{\uparrow})_{\alpha, \beta} = (W_t^{\uparrow})_{\alpha, \beta} - \overline{\gamma} Q_t \Big( (N_{\text{in}} N_{\text{out}} dW_0^{\uparrow} + \overline{\lambda} W_0^{\uparrow})_{\alpha, \beta}, \ldots, (N_{\text{in}} N_{\text{out}} dW_t^{\uparrow} + \overline{\lambda} W_t^{\uparrow})_{\alpha, \beta}; \, \overline{\varepsilon} \Big).
$$

Since for all $s \leq t$, by the induction hypothesis we have $N_{\text{in}} N_{\text{out}} dW_s^{\uparrow} = n_{\text{in}} n_{\text{out}} dW_s \otimes \mathbf{1}_{k_{\text{out}}} \mathbf{1}_{k_{\text{in}}}^{\top}$ and $W_s^{\uparrow} = W_s \otimes \mathbf{1}_{k_{\text{out}}} \mathbf{1}_{k_{\text{in}}}^{\top}$, we conclude that the matrix updates satisfy

$$
W_{t+1}^{\uparrow} = W_{t+1} \otimes \mathbf{1}_{k_{\text{out}}} \mathbf{1}_{k_{\text{in}}}^{\top}.
$$

For the vector weight $u$, the update rule in the base model is

$$(u_{t+1})_\alpha = (u_t)_\alpha - \gamma\, Q_t\Big((du_0 + \lambda u_0)_\alpha, \ldots, (n_u du_t + \lambda u_t)_\alpha; \varepsilon\Big).$$

$$= (u_t)_\alpha - n_u^m \overline{\gamma}\, Q_t\Big((du_0 + n_u^{-1}\overline{\lambda} u_0)_\alpha, \ldots, (n_u du_t + n_u^{-1}\overline{\lambda} u_t)_\alpha;\ n_u^{-1}\overline{\varepsilon}\Big)$$

$$= (u_t)_\alpha - \overline{\gamma}\, Q_t\Big((n_u du_0 + \overline{\lambda} u_0)_\alpha, \ldots, (n_u du_t + \overline{\lambda} u_t)_\alpha;\ \overline{\varepsilon}\Big).$$

Similarly, the update rule in the widened model is

$$(u_{t+1}^\uparrow)_\alpha = (u_t^\uparrow)_\alpha - \overline{\gamma}\, Q_t\Big((N_u du_0^\uparrow + \overline{\lambda} u_0^\uparrow)_\alpha, \ldots, (N_u du_t^\uparrow + \overline{\lambda} u_t^\uparrow)_\alpha;\ \overline{\varepsilon}\Big).$$

Since for all $s \leq t$, we have $N_u du_s^\uparrow = n_u du_s \otimes \mathbf{1}_{k_u}$ and $u_s^\uparrow = u_s \otimes \mathbf{1}_{k_u}$, we conclude that the vector updates satisfy

$$u_{t+1}^\uparrow = u_{t+1} \otimes \mathbf{1}_{k_u}.$$

Finally, for the scalar weight $c$, it is immediate that $c_{t+1}^\uparrow = c_{t+1}$.

$$\square$$

### C.3. Explanation of *μ*P in Table 2

The $\mu$P that we stated in Table 2 is not exactly the same as the versions of $\mu$P in Littwin & Yang (2023); Yang et al. (2021). Our convention is intended to facilitate direct comparison with the widening operation that produces the equivalent model in Table 1. Below we compare these conventions with those in the prior papers and justify their equivalence.

**One degree of degeneracy.** More generally, one may introduce a width-dependent *weight multiplier* $A$ so that the actual weight relates to the width-independent rescaled weight via $W = A \cdot \overline{W}$, with $\overline{W}_{\alpha\beta} \sim \mathcal{N}\left(0, B \cdot \overline{\sigma}^2\right)$. In our convention (Table 2), we set $A = 1$ for all weight types, which is why $A$ does not appear in the table. However, keeping $A$ explicit reveals a symmetry with one degree of degeneracy in the choice of $A, B, C, D, \tilde{D}, E$, which we state as a variant of Lemma J.1 in Yang et al. (2021).

**Proposition C.10** (Symmetry of scalings). *Consider the setup of an entrywise optimizer with an update function $Q_t$ that is homogeneous of degree $m$. Suppose we adopt a parametrization prescribed by scalings $A, B, C, D, \tilde{D}, E$ as stated in Table 2.*

*For all $\theta > 0$, at any finite width, if we set*

$$A \leftarrow A\theta, \quad B \leftarrow B/\theta^2, \quad C \leftarrow C/\theta^{1+m}, \quad D \leftarrow D\theta^2, \quad \tilde{D} \leftarrow \tilde{D}\theta^{1+m}, \quad E \leftarrow E\theta,$$

*then we obtain a parametrization that is exactly equivalent.*

*Proof.* Let $W = A\theta\,\overline{W}$ with $\overline{W}_{\alpha\beta} \sim \mathcal{N}\left(0, \frac{B}{\theta^2}\overline{\sigma}^2\right)$. It follows immediately that $W_{\alpha\beta} \sim \mathcal{N}\left(0, A^2 B\,\overline{\sigma}^2\right)$, which is independent of $\theta$.

The update rule for $\overline{W}$ with learning rate $C\,\theta^{-(1+m)}\overline{\gamma}$, weight decay constant $D\,\theta^2\overline{\lambda}$, decoupled weight decay constant $\tilde{D}\,\theta^{1+m}\overline{\lambda}$, and auxiliary hyperparameter $E\theta\,\overline{\varepsilon}$ is (choose either $D = 0$ or $\tilde{D} = 0$)

$$(\overline{W}_{t+1})_{\alpha,\beta} = \left(1 - \overline{\lambda}\,\overline{\gamma}\,C\,\tilde{D}\right)(\overline{W}_t)_{\alpha,\beta} - C\,\theta^{-(1+m)}\overline{\gamma}\, Q_t\Big(\left(d\overline{W}_0 + D\theta^2\overline{\lambda}\,\overline{W}_0\right)_{\alpha,\beta}, \ldots,$$

$$\left(d\overline{W}_t + D\theta^2\overline{\lambda}\,\overline{W}_t\right)_{\alpha,\beta};\ E\theta\,\overline{\varepsilon}\Big).$$

Since $W = A\theta\,\overline{W}$, we have $d\overline{W} = A\theta\, dW$, hence

$$(A\theta)^{-1}(W_{t+1})_{\alpha,\beta}$$

$$= \left(1 - \overline{\lambda}\,\overline{\gamma}\,C\,\tilde{D}\right)(A\theta)^{-1}(W_t)_{\alpha,\beta}$$

$$- C\,\theta^{-(1+m)}\overline{\gamma}\,Q_t\Big(\big(A\theta\,dW_0 + D\theta^2\overline{\lambda}\,(A\theta)^{-1}W_0\big)_{\alpha,\beta}, \ldots, \big(A\theta\,dW_t + D\theta^2\overline{\lambda}\,(A\theta)^{-1}W_t\big)_{\alpha,\beta};\ E\theta\,\overline{\varepsilon}\Big)$$

$$= \big(1 - \overline{\lambda}\,\overline{\gamma}\,C\,\tilde{D}\big)\,(A\theta)^{-1}(W_t)_{\alpha,\beta}$$
$$- A^m C\theta^{-1}\overline{\gamma}\,Q_t\Big(\big(dW_0 + \overline{\lambda}DA^{-2}W_0\big)_{\alpha,\beta}, \ldots, \big(dW_t + \overline{\lambda}DA^{-2}W_t\big)_{\alpha,\beta};\ EA^{-1}\overline{\varepsilon}\Big),$$

(by the homogeneity of $Q_t$ of degree $m$)

Multiplying both sides by $A\theta$ yields

$$(W_{t+1})_{\alpha,\beta} = \big(1 - \overline{\lambda}\,\overline{\gamma}\,C\,\tilde{D}\big)\,(W_t)_{\alpha,\beta}$$
$$- A^{m+1}C\,\overline{\gamma}\,Q_t\Big(\big(dW_0 + \overline{\lambda}DA^{-2}W_0\big)_{\alpha,\beta}, \ldots, \big(dW_t + \overline{\lambda}DA^{-2}W_t\big)_{\alpha,\beta};\ EA^{-1}\overline{\varepsilon}\Big),$$

which is independent of $\theta$. This proves that the rescaled parametrization is exactly equivalent. $\square$

*Remark* C.11. This proposition describes *exact equivalence* at any finite width. We may also consider *asymptotic equivalence*: two scalings $(A, B, C, D, \tilde{D}, E)$ and $(A', B', C', D', \tilde{D}', E')$ are asymptotically equivalent if there exists $\theta > 0$ such that

$$\frac{A\theta}{A'}, \quad \frac{B/\theta^2}{B'}, \quad \frac{C/\theta^{1+m}}{C'}, \quad \frac{D\theta^2}{D'}, \quad \frac{\tilde{D}\,\theta^{1+m}}{\tilde{D}'}, \quad \frac{E\theta}{E'}$$

are all $\Theta(1)$ as the width(s) go to $\infty$ (if there are multiple width dimensions, we assume they go to $\infty$ at the same rate). The $\mu$P scaling is defined via its asymptotic behaviour and hence identifies a family of scalings only up to asymptotic equivalence.

**Comparison with $\mu$P in Definition 2.9.12 of Littwin & Yang (2023).** The *abcd*-parametrization (see Definition 2.9.7 of Littwin & Yang (2023)) considers a general update function $Q_t$ rather than restricting to homogeneous ones. As noted in their Remark 2.2.6, when $Q_t$ is homogeneous of degree $m$, one should interpret $n^{-d}$ as the scaling for $\varepsilon$ and $n^{dm-c}$ as the scaling for the learning rate $\gamma$.

In their Definition 2.9.12, where $\mu$P is defined, they do not consider weight decay. For vector-like weights, the parametrization translates to

$$A = 1, \quad B = 1, \quad C = n^m, \quad E = n^{-1}$$

in our convention. For matrix-like weights, the parametrization translates to

$$A = 1, \quad B = n^{-1}, \quad C = n^{m-1}, \quad E = n^{-1},$$

and it does not distinguish $n_{\text{out}}$ and $n_{\text{in}}$ as we do, because it assumes all widths go to $\infty$ at the same rate.

Weight decay is treated in Section 2.10.1, where they only consider a decoupled version (note that their notion of "decoupled" differs from ours). There they claim that $\tilde{D}$ should be set so that $C\tilde{D} = 1$. That is, $\tilde{D} = n^{-m}$ for vector-like weights, and $\tilde{D} = n^{1-m}$ for matrix-like weights.

One can verify that this choice of $A, B, C, \tilde{D}, E$ is asymptotically equivalent to what we state in Table 2, except that we add support for vanilla weight decay (controlled by $D$) and explicitly distinguish $n_{\text{in}}$ and $n_{\text{out}}$ for matrix-like weights.

**Comparison with $\mu$P in Table 8 of Yang et al. (2021).** The version of $\mu$P stated in Yang et al. (2021) does not consider a general entrywise optimizer; instead, it gives parametrizations specifically for SGD and Adam. They do not explicitly discuss weight decay, though it is included in the implementation (we will discuss this later in Appendix C.4). They also do not discuss scaling for $\varepsilon$. So we focus on the comparison of $A, B, C$.

For vector-like input weights, the first column of Table 8 of Yang et al. (2021) translates to

$$A = 1, \quad B = n_{\text{in}}^{-1}, \quad C = \begin{cases} n_{\text{out}} & \text{for SGD,} \\ 1 & \text{for Adam.} \end{cases}$$

Notice that here $n_{\text{in}}$ is constant, while $n_{\text{out}} \to \infty$ (comparable to $n$ in our convention). This is asymptotically equivalent to what we state in the "Vector-like" row of Table 2 (setting $m = 1$ for SGD and $m = 0$ for Adam).

For vector-like output weights, the second column of Table 8 of (Yang et al., 2021) translates to

$$A = 1, \quad B = 1, \quad C = \begin{cases} n_{\text{in}} & \text{for SGD,} \\ 1 & \text{for Adam,} \end{cases}$$

with an additional $n_{\text{in}}^{-1}$ output multiplier. In this case $n_{\text{in}} \to \infty$ (comparable to $n$ in our convention). This is exactly equivalent to what we state in the "Vector-like" row of Table 2.

For matrix-like weights, the third column of Table 8 of Yang et al. (2021) translates to

$$A = 1, \quad B = n_{\text{in}}^{-1}, \quad C = \begin{cases} 1 & \text{for SGD,} \\ n_{\text{in}}^{-1} & \text{for Adam.} \end{cases}$$

This is exactly equivalent to what we state in the "Matrix-like" row of Table 2.

### C.4. Experimental Verification of Equivalence

To verify Theorem 2.5, we perform experiments on MLPs, ResNets, and Transformers with variants of SGD, Adam, and AdamW. For each model, we first train a base model from scratch for several epochs under $\mu$P, then construct a widened model whose weights are obtained using the rule in Table 1. We next create an optimizer for the widened model and transfer the internal state from the base model to the widened model (explained in Appendix D). Finally, we train both the base and widened models under $\mu$P using the same base hyperparameters and compare their equivalence after each training step.

All models pass this verification. We directly use the $\mu$P package introduced in Yang et al. (2021), with slight modifications to recover our specific scaling in Table 2 that is essential for exact equivalence. We explain these modifications below.

**Modification of the $\mu$P package** We summarize in Table 3 the exact scaling implemented in the $\mu$P package and our minor modifications (highlighted in blue). Specifically, we add support for the default weight decay in Adam, whereas originally only the decoupled weight decay (same as in AdamW) was implemented. Moreover, we implement the correct scaling for $\varepsilon$ for Adam and AdamW.

*Remark* C.12. Below are some further details about this implementation:

1. Table 8 of Yang et al. (2021) claims to document the scalings used in the $\mu$P implementation, but it specifies only the entries for $A, B, C$. The scalings $\tilde{D}$ for Adam and $D$ for SGD were implemented but not explicitly specified in that paper. We document their explicit forms in the $\mu$P implementation and list them in Table 3 (in black and purple). Table 8 of Yang et al. (2021) also states the SGD learning-rate scaling for hidden (matrix) weights to be 1, but the actual implementation uses $n_{\text{out}}/n_{\text{in}}$, which we therefore reflect here.

2. The $\mu$P implementation assumes `decoupled_weight_decay=True` in PyTorch for Adam, in which case Adam is equivalent to AdamW. Consequently, it implements only the $\tilde{D}$ scaling and not $D$. We modify `MuAdam` to also implement the $D$ scaling corresponding to the default, coupled weight decay in Adam.

3. The original implementation does not scale $\varepsilon$, even though later theoretical analysis in Littwin & Yang (2023) recommends scaling $\varepsilon$; see Remark 2.2.6 of Littwin & Yang (2023). We modify `MuAdam` to implement this scaling as in Table 2.

4. The $\mu$P implementation requires a specified *base model width*, and all scalings are adjusted so that the base model behaves identically to the standard parametrization. Concretely, this is equivalent to multiplying an additional constant on top of what is shown in Table 3.

## D. Details of the Upscaling Algorithm

We explain the omitted details of the upscaling algorithm presented in Meta-algorithm 1.

**Simple example: MLP trained with SGD.** For concreteness, we detail the algorithm for the simple example of an MLP trained with vanilla SGD used in Section 2. Note that vanilla SGD does not maintain any internal optimizer state (e.g., momentum), so there is nothing to duplicate or rescale in the optimizer's checkpoint in Step 3 of Meta-algorithm 1.

*Table 3.* $\mu$P scaling in its implementation. Purple text highlights the key differences between $\mu$P scaling and the standard parametrization that is the default in `PyTorch` (written in parentheses in black when different). Blue text highlights our modifications on top of the original implementation, where these scalings were not considered.

| | Input vector (inf $\times$ fin) | | Matrix (inf $\times$inf) | | Output vector (fin $\times$inf) | |
|---|---|---|---|---|---|---|
| Multiplier of weight $W = A \cdot \overline{W}$ | 1 | | 1 | | $\frac{1}{n_{\text{in}}}$ | (1) |
| Init variances $\overline{W}_{\alpha\beta} \sim \mathcal{N}\left(0, B \cdot \overline{\sigma}^2\right)$ | $\frac{1}{n_{\text{in}}}$ | | $\frac{1}{n_{\text{in}}}$ | | 1 | $(\frac{1}{n_{\text{in}}})$ |
| SGD learning rate scaling $\gamma = C \cdot \overline{\gamma}$ | $n_{\text{out}}$ | (1) | $\frac{n_{\text{out}}}{n_{\text{in}}}$ | (1) | $n_{\text{in}}$ | (1) |
| SGD weight decay scaling $\lambda = D \cdot \overline{\lambda}$ | $\frac{1}{n_{\text{out}}}$ | (1) | $\frac{n_{\text{in}}}{n_{\text{out}}}$ | (1) | $\frac{1}{n_{\text{in}}}$ | (1) |
| Adam learning rate scaling $\gamma = C \cdot \overline{\gamma}$ | 1 | | $\frac{1}{n_{\text{in}}}$ | (1) | 1 | |
| Adam weight decay scaling $\lambda = D \cdot \overline{\lambda}$ | $\frac{1}{n_{\text{out}}}$ | (1) | $\frac{n_{\text{in}}}{n_{\text{out}}}$ | (1) | $\frac{1}{n_{\text{in}}}$ | (1) |
| AdamW weight decay scaling $\lambda = \tilde{D} \cdot \overline{\lambda}$ | 1 | | $n_{\text{in}}$ | (1) | 1 | |
| Adam, AdamW $\varepsilon$ scaling $\varepsilon = E \cdot \overline{\varepsilon}$ | $\frac{1}{n_{\text{out}}}$ | (1) | $\frac{1}{n_{\text{out}}}$ | (1) | $\frac{1}{n_{\text{in}}}$ | (1) |

---

**Algorithm 2** Upscaling for MLPs under SGD

---

**Input:** Base MLP weights $(W^{(\ell)} \in \mathbb{R}^{n_\ell \times n_{\ell-1}})_{\ell=1}^L$; expansion multipliers $(k_\ell \in \mathbb{N})_{\ell=1}^{L-1}$.

**Output:** Trained upscaled MLP $(W^{\uparrow (\ell)} \in \mathbb{R}^{N_\ell \times N_{\ell-1}})_{\ell=1}^L$.

**set** Tune $\overline{\sigma_\Delta}$ and $\overline{\gamma^\uparrow}$ by running the steps below on a small system (widths $n_0 \to k_\ell n_0$ with $n_0 \ll n_1$), selecting the values that minimize the terminal training loss.

**set** $k_0 \leftarrow 1$, $k_L \leftarrow 1$; $N_\ell \leftarrow k_\ell n_\ell$ for all $\ell = 0, \ldots, L$.

**set** $\sigma_\Delta^{(1)} \leftarrow \overline{\sigma_\Delta}$, $\sigma_\Delta^{(L)} \leftarrow \overline{\sigma_\Delta}$; $\sigma_\Delta^{(\ell)} \leftarrow N_{\ell-1}^{-1/2}\, \overline{\sigma_\Delta}$ for all $\ell = 2, \ldots, L-1$.

  **for** $\ell = 1$ to $L$ **do**

    Sample $\Delta^{(\ell)} \in \mathbb{R}^{N_\ell \times N_{\ell-1}}$ with $\Delta_{ij}^{(\ell)} \sim \mathcal{N}\left(0, \sigma_\Delta^{(\ell)^2}\right)$.

    $W^{\uparrow (\ell)} \leftarrow k_{\ell-1}^{-1} W^{(\ell)} \otimes \mathbf{1}_{k_\ell} \mathbf{1}_{k_{\ell-1}}^\top\ +\ \Delta^{(\ell)}$.

  **end for**

  Under $\mu$P, run SGD on the upscaled MLP initialized with $(W^{\uparrow (\ell)})$, using a $\mu$P-scaled learning rate based on $\overline{\gamma^\uparrow}$.

---

**Transfer weights: implementation of Step 1.** With the modified $\mu$P package in place, we implement weight transfer from the base to the widened model via a generic routine that works for standard architectures that we consider in our experiments. Concretely, the routine operates on both `named_parameters` (model weights) and, when present, `named_buffers` (e.g., BatchNorm running statistics).

For each tensor in `named_parameters` saved in the checkpoint, we query its $\mu$P `infshape` to classify it as scalar-like, vector-like, or matrix-like, and then duplicate/rescale according to the rules in the "Widening operation" column of Table 1.

For `named_buffers`, usually simple duplication and copy-over is sufficient. In particular, BatchNorm `running_mean` and `running_var` are vector-like and are duplicated without rescaling, while scalar buffers such as `num_batches_tracked` are copied as-is. Typically no matrix-like buffers occur, so no rescaling is needed.

**Transfer optimizer's internal state: implementation of Step 3.** One detail not elaborated in Meta-algorithm 1 is how to transfer the optimizer's internal state, as in Step 3. In Theorem 2.5, for simplicity of analysis, we construct the widened model before any training and show that the widened and base models undergo equivalent updates when the hyperparameters are set appropriately. However, in practical upscaling, we perform widening mid-training. To ensure that the widened model (without adding any noise) is updated equivalently to the base model, as if no upscaling had occurred, it is necessary to transfer the optimizer's internal state.

We implemented a generic routine for transferring optimizer, which works for SGD, Adam, and AdamW (and their variants).

Extension to other optimizers is possible but may require additional adjustments. The procedure mirrors the transfer of weights. For each tensor in `named_parameters` saved in the optimizer's checkpoint, we query its $\mu$P `infshape` to classify it as scalar-like, vector-like, or matrix-like, and then duplicate and rescale accordingly. Scalar-like tensors are copied as is. For vector-like and matrix-like tensors, in SGD the relevant state is the saved `momentum`, and in Adam/AdamW this is the saved `exp_avg` or `exp_avg_sq`, for the corresponding vector-like or matrix-like weight. Recall from the proof of Proposition 2.2 that the gradients of the equivalent widened model satisfy

$$dW^{\uparrow} = k_{\text{out}}^{-1} dW \otimes \mathbf{1}_{k_{\text{out}}} \mathbf{1}_{k_{\text{in}}}^{\top} \quad \text{if } W \text{ is matrix-like;}$$

$$dW^{\uparrow} = k^{-1} dW \otimes \mathbf{1}_{k} \quad \text{if } W \text{ is vector-like.}$$

Saved `momentum` and `exp_avg` should be transferred in the same way as the gradient $dW$. We treat the `exp_avg_sq` case separately, as it requires a distinct rescaling: the scaling factors $k_{\text{out}}^{-1}$ or $k^{-1}$ should be squared, consistent with second-moment accumulation.

**Practical considerations for weight, buffer, and optimizer-state transfer.** Our algorithm transfers model weights, `named_buffers`, and the optimizer's internal state. Transferring weights is the most essential step, as it determines the function computed by the upscaled model. Transferring `named_buffers` and optimizer state further guarantees that, with zero added noise, training continues exactly as if no upscaling had occurred; we expect this to stabilize early training by preserving momentum and second-moment estimates from pretraining. When optimizer checkpoints are unavailable (e.g., when upscaling a published model), it is acceptable to omit this transfer.

**Alternative algorithm: signal-normalized noise.** In Meta-algorithm 1, Step 2 adds noise whose standard deviation is controlled by a single base constant $\overline{\sigma_{\Delta}}$ shared across all layers. However, in architectures with heterogeneous hidden dimensions (e.g., ResNet, where the number of channels varies across stages), the widened weights at different layers can have substantially different spectral norms, so a single $\overline{\sigma_{\Delta}}$ may inject too much noise in some layers and too little in others.

To address this, we introduce an alternative noise injection scheme that normalizes the noise relative to the signal on a per-layer basis. Given any architecture and optimizer, the modification applies to Steps 0 and 2 of Meta-algorithm 1; all other steps (weight transfer, optimizer state transfer, and training) remain unchanged.

*Modification to Step 2 (noise injection).* For each weight tensor $W$ in the model, let $\widetilde{W}$ denote its widened version constructed according to Table 1. Instead of adding noise with a globally controlled standard deviation, we set the upscaled weight to

$$W^{\uparrow} := \widetilde{W} + t \frac{\|\widetilde{W}\|}{\|\Delta\|} \Delta,$$

where $\|\cdot\|$ denotes the spectral norm, $\Delta$ is a random tensor of the same shape as $\widetilde{W}$ sampled with the same width-dependent scaling that $\mu$P prescribes for random initialization (with $\overline{\sigma_{\Delta}} = 1$), and $t \geq 0$ is a hyperparameter controlling the noise level relative to the signal. The rescaling ensures that the signal and noise terms have equal spectral norms when $t = 1$; sweeping $t$ modulates the relative noise level. A practical benefit of this formulation is that the natural sweep range for $t$ is $[0, 1]$: the value $t = 0$ recovers exact weight transfer (no noise), while $t = 1$ adds noise whose spectral norm equals that of the signal, which is already a substantial perturbation. In contrast, the original algorithm requires prior experience or preliminary experiments to determine the appropriate sweep range for $\overline{\sigma_{\Delta}}$, since its effect depends on the (architecture- and training-dependent) scale of the widened weights.

*Modification to Step 0 (hyperparameter tuning).* The hyperparameter $t$ does not necessarily transfer reliably across widths. Instead, after tuning $t$ on a small system, one computes the effective per-weight noise base constant $\overline{\sigma_{\Delta,W}} = t \|\widetilde{W}\|/\|\Delta\|$ for each weight tensor $W$. Since $\overline{\sigma_{\Delta,W}}$ is expected to transfer across widths, these values are recorded and used in lieu of a single $\overline{\sigma_{\Delta}}$ when performing the actual upscaling at the target width.

For the specific case of an MLP with SGD (cf. Algorithm 2), the modified noise injection step becomes

$$W^{\uparrow(\ell)} := k_{\ell-1}^{-1} W^{(\ell)} \otimes \mathbf{1}_{k_\ell} \mathbf{1}_{k_{\ell-1}}^{\top} + t \frac{\left\| k_{\ell-1}^{-1} W^{(\ell)} \otimes \mathbf{1}_{k_\ell} \mathbf{1}_{k_{\ell-1}}^{\top} \right\|}{\left\| \Delta^{(\ell)} \right\|} \Delta^{(\ell)} \in \mathbb{R}^{N_\ell \times N_{\ell-1}},$$

where $\Delta^{(\ell)} \in \mathbb{R}^{N_\ell \times N_{\ell-1}}$ has i.i.d. entries $\Delta_{ij}^{(\ell)} \sim \mathcal{N}\left(0, \sigma_\Delta^{(\ell)2}\right)$ with $\sigma_\Delta^{(1)} = \sigma_\Delta^{(L)} = 1$ and $\sigma_\Delta^{(\ell)} = N_{\ell-1}^{-1/2}$ for $\ell = 2, \dots, L - 1$. The corresponding per-layer effective noise constant is

$$\overline{\sigma_{\Delta, W^{(\ell)}}} = t \, \frac{\left\| k_{\ell-1}^{-1} W^{(\ell)} \otimes \mathbf{1}_{k_\ell} \mathbf{1}_{k_{\ell-1}}^\top \right\|}{\left\| \Delta^{(\ell)} \right\|} = t \sqrt{\frac{k_\ell}{k_{\ell-1}}} \, \frac{\left\| W^{(\ell)} \right\|}{\left\| \Delta^{(\ell)} \right\|}.$$

# E. Infinite-Width Training Dynamics for Upscaling

We present in Appendix E.1 a modified NE⊗OR⊤ program tailored to analyzing infinite-width limits under mid-training upscaling. In Appendix E.2, we prove its correctness. The proof strategy is to show that the modified program can be expressed as compositions of operations in the original NE⊗OR⊤ program, thereby reducing the analysis to the standard framework and enabling the application of Tensor Program techniques. Finally, to illustrate the usefulness of our modified NE⊗OR⊤ program for analyzing infinite-width limits with upscaling, we work through two simple examples in Appendix E.3 and Appendix E.4, from which we derive several insights.

## E.1. Main Result: Modified Tensor Program

A SHORT RECAP OF THE ORIGINAL NE⊗OR⊤ PROGRAM

The original NE⊗OR⊤ program, introduced in Littwin & Yang (2023), captures infinite-width training dynamics for standard neural network architectures. First, one writes all neural computations during training in the NE⊗OR⊤ language, including both forward and backward passes at each training step. Given this NE⊗OR⊤ program, one then defines the "ket" construction recursively: each vector appearing in the program (for example, layer preactivations) is represented as a random variable (called a "ket") that encodes its distribution as the width $n \to \infty$, and the limiting model outputs or losses at each step are expressed as expectations of functions of these random variables. This reduction yields a deterministic system of "limit equations" describing the evolution of activation and weight distributions over time. Finally, the Master Theorem provides the theoretical foundation, rigorously characterizing the sense in which finite-width dynamics converge to the limiting object defined by the ket construction. We now present our modified NE⊗OR⊤ program, which retains this structure.

MODIFIED NE⊗OR⊤ PROGRAM LANGUAGE

We characterize the infinite-width limit $n \to \infty$ for the following training-and-upscaling procedure: Consider a network architecture specified in the NE⊗OR⊤ language with hidden width $n$. After training for $T$ epochs at width $n$, we upscale to a wider model with hidden width $N = kn$, where $k$ is a fixed constant, and then continue training for an additional $T'$ epochs. (Because we focus on asymptotic training dynamics, it suffices to take all hidden layers to have the same width, as in the original NE⊗OR⊤ construction; this contrasts with Appendix C.2, where we allow heterogeneous hidden widths and expansion factors to study exact finite-width equivalence.)

Rather than extending the language to directly implement the upscaling operation, we seek the minimal modification of NE⊗OR⊤ that preserves its structure as a composition of matrix multiplications and elementwise nonlinearities while enabling the expression of upscaling. By the analysis in Section 2, the training dynamics at width $n$ over the first $T$ epochs can be represented equivalently at width $kn$. Consequently, instead of changing dimension mid-training, we model the entire procedure at the higher dimension $kn$ by lifting the pre-upscaling stage to that dimension. We therefore define a modified NE⊗OR⊤ program whose only deviation from Definition 2.6.1 of Littwin & Yang (2023) is that all hidden-layer dimensions are $nk$, where $n$ is the pre-upscaled hidden width and $k$ is the multiplier.

**Definition E.1.** The modified NE⊗OR⊤ program generates a sequence $\boldsymbol{x}$ of $\mathbb{R}^{nk}$-vectors and a sequence $\boldsymbol{c}$ of $\mathbb{R}$-scalars inductively defined via one of the following ways from an initial set $\boldsymbol{c}^0 \subseteq \boldsymbol{c}$ of random scalars, an initial set $\boldsymbol{x}^0 \subseteq \boldsymbol{x}$ of random $\mathbb{R}^{nk}$ vectors, and an initial set $\mathcal{W}$ of random $\mathbb{R}^{(nk) \times (nk)}$ matrices.

With a slight abuse of notation, we sometimes treat $\boldsymbol{x}, \boldsymbol{c}$ as sets, and at other times regard of $\boldsymbol{c}$ as a vector and $\boldsymbol{x}$ as a matrix with the $\mathbb{R}^{nk}$ vectors as columns; then $\boldsymbol{c}^0$ is just a subvector of $\boldsymbol{c}$ and $\boldsymbol{x}^0$ is a submatrix of $\boldsymbol{x}$. At each step of the program, one can:

`Avg`: Choose a vector $x \in \boldsymbol{x}$ (think of $x$ as a column in $\boldsymbol{x} \in \mathbb{R}^{nk \times |\boldsymbol{x}|}$) and append to $\boldsymbol{c}$ a scalar

$$\langle x_\alpha \rangle_\alpha = \frac{1}{nk} \sum_{\alpha=1}^{nk} x_\alpha \in \mathbb{R}. \tag{14}$$

`MatMul`: Choose a matrix $W \in \mathcal{W}$ and vector $x \in \boldsymbol{x}$, and append to $\boldsymbol{x}$ the vector

$$Wx \in \mathbb{R}^{nk} \quad \text{or} \quad W^\top x \in \mathbb{R}^{nk}.$$

`OuterNonlin`: Choose an integer $r \geq 0$ and a function $\psi : \mathbb{R}^{|\boldsymbol{x}|(r+1)+l} \to \mathbb{R}$, and append to $\boldsymbol{x}$ the vector

$$y \in \mathbb{R}^{nk}, \quad y_\alpha = \langle \psi(\boldsymbol{x}_\alpha; \boldsymbol{x}_{\beta_1}; \dots; \boldsymbol{x}_{\beta_r}; \boldsymbol{c}) \rangle_{\beta_1, \dots, \beta_r} = \frac{1}{(nk)^r} \sum_{\beta_1, \dots, \beta_r = 1}^{nk} \psi(\boldsymbol{x}_\alpha; \boldsymbol{x}_{\beta_1}; \dots; \boldsymbol{x}_{\beta_r}; \boldsymbol{c}) \tag{15}$$

where $\boldsymbol{x}_\gamma \in \mathbb{R}^{|\boldsymbol{x}|}$ is the $\gamma$-th row in $\boldsymbol{x}$ as a matrix and $|\boldsymbol{x}|$ is the number of vectors in $\boldsymbol{x}$.

We consider two canonical initialization regimes that encompass both Gaussian and non-Gaussian cases. In contrast to the original setup in Littwin & Yang (2023), we allow two classes of random matrices. The first class consists of matrices of size $nk$ with i.i.d. entries, which model the noise injected during upscaling. The second class consists of matrices of size $nk$ obtained by duplicating a size-$n$ random matrix with i.i.d. entries, which model (an already-widened version of) the weights at the start of training. Analogously, the random vectors at initialization are partitioned into the same two classes. It is straightforward to verify that any network architecture expressible in the original NE⊗OR⊤ program admits an upscaling procedure that can be formulated within the modified NE⊗OR⊤ framework defined here.

**Assumption E.2** (Gaussian). The following three hold.

1. The initial set of random matrices is defined as $\mathcal{W} = \mathcal{W}_1 \cup \mathcal{W}_2$. Every entry of each $W \in \mathcal{W}_1$ is sampled i.i.d. from $\mathcal{N}\left(0, \frac{1}{n}\right)$. Each matrix $W' \in \mathcal{W}_2$ is formed as $W' = W \otimes \mathbf{1}_k \mathbf{1}_k^\top$, where $W$ has i.i.d. entries sampled from $\mathcal{N}\left(0, \frac{1}{n}\right)$.

2. The initial set of vectors is defined as $\boldsymbol{x}^0 = \boldsymbol{x}^{0,1} \cup \boldsymbol{x}^{0,2}$. Every entry of each $x \in \boldsymbol{x}^{0,1}$ is sampled i.i.d. from $\mathcal{N}(0, 1)$. Each vector $x' \in \boldsymbol{x}^{0,2}$ is formed by $x' = x \otimes \mathbf{1}_k$, where $x$ has i.i.d. entries sampled from $\mathcal{N}(0, 1)$.

3. The initial scalars $c^0$ converge almost surely to 0.

4. All functions $\psi$ used in `OuterNonlin` are pseudo-Lipschitz.

**Assumption E.3** (Non-Gaussian). Suppose the same items as Assumption E.2 but with 1) and 4) replaced by the following two.

1*. There exists a sequence $\nu_3, \nu_4, \dots > 0$ such that (1) each $W \in \mathcal{W}_1$ has independent entries drawn from distributions with zero mean, variance $\frac{1}{n}$ and all higher $t$-th moment bounded by $\nu_t n^{-t/2}$; (2) each $W' \in \mathcal{W}_2$ is formed by $W' = W \otimes \mathbf{1}_k \mathbf{1}_k^\top$, where $W$ has independent entries drawn from distributions with zero mean, variance $\frac{1}{n}$, and all higher $t$-th moment bounded by $\nu_t n^{-t/2}$.

4*. All functions $\psi$ used in `OuterNonlin` are polynomially smooth.

We further require initial scalars $\boldsymbol{c}^0$ to have moments of all orders bounded in $n$.

THE "KET" CONSTRUCTION

In the infinite-width limit ($n \to \infty$), the training dynamics of a neural network can be analyzed using a calculus of random variables. Without upscaling, in the original NE⊗OR⊤ program, the preactivation vector in each layer is distributed like $n$ i.i.d. random variables. Given a NE⊗OR⊤ program, Littwin & Yang (2023) constructs the associated limit objects using the ket notation, which are random variables that track these preactivations at every step of training. We proceed analogously with upscaling; the difference is that, in this case, each layer's preactivations behave like $n$ blocks consisting of random vectors of dimension $k$, where these small vectors may have depending entries, but different vectors are i.i.d. Hence, we need to keep track of random $k$-vectors rather than just scalar random variables.

We first clarify the notation, which is based on but slightly deviates from that in Littwin & Yang (2023).

**Notation**

- For a vector $x \in \mathbb{R}^{nk}$, for $i \in [k]$ we write

$$x_{(i)} = (x_i, x_{i+k}, x_{i+2k} \dots) \in \mathbb{R}^n, \tag{16}$$

  We are concerned with the situation where $x$ has $n$ i.i.d. $k$-blocks; equivalently, each $x_{(i)}$ has i.i.d. entries.

  Similarly, for a matrix $W \in \mathbb{R}^{nk \times nk}$, for $i, j \in [k]$ we write

$$W_{(i,j)} := \left( W_{i+(r-1)k, \, j+(c-1)k} \right)_{r,c=1}^{n} \in \mathbb{R}^{n \times n}.$$

  This $W_{(i,j)}$ collects, in an interleaved manner, the $(i, j)$-th entries from each $k \times k$ block of $W$ across the $n \times n$ grid of blocks. We are concerned with the situation where $W$ has $n \times n$ i.i.d. $k \times k$ blocks; equivalently, each $W_{(i,j)}$ has i.i.d. entries.

- For a vector $x \in \mathbb{R}^{nk}$ with $n$ i.i.d. $k$-blocks, we write $|x\rangle \in \mathbb{R}^k$ for a random vector such that $x$ look like $n$ i.i.d. samples from $|x\rangle$. We write $\langle x \mid y \rangle$ for the $k \times k$ matrix where

$$\langle x \mid y \rangle_{ij} := \mathbb{E}[|x\rangle_i |y\rangle_j] \tag{17}$$

  We are concerned with the situation where $x, y$ look like $n$ i.i.d. samples from $|x\rangle, |y\rangle$ respectively, in the sense that, for any two such vectors $x, y \in \mathbb{R}^{nk}$, we have

$$\lim_{n \to \infty} \frac{(x_{(i)})^\top (y_{(j)})}{n} = \langle x \mid y \rangle_{ij}.$$

  This implies that

$$\lim_{n \to \infty} \frac{x^\top y}{nk} = \frac{\mathrm{Tr} \langle x \mid y \rangle}{k}.$$

  Our notation differs from that of the original NE⊗OR⊤ program: there, $|x\rangle$ denotes a scalar random variable, and $\langle x \mid y \rangle := \mathbb{E}\, |x\rangle |y\rangle \in \mathbb{R}$ captures the limit of $x^\top y$. In that setting, one considers the simpler case where $x \in \mathbb{R}^n$ consists of $n$ i.i.d. samples from $|x\rangle$. We further note that our use of $\langle x \mid y \rangle_{ij}$ departs somewhat from the conventional bra–ket notation in physics; we retain this form to remain as close as possible to the original NE⊗OR⊤ program.

  In later parts of the appendix, we will slightly overload the bra–ket notation and use $\langle \cdot \mid \cdot \rangle$ uniformly in both the original and modified NE⊗OR⊤. The intended meaning should be clear from context—namely, whether $x, y \in \mathbb{R}^n$ or $\mathbb{R}^{nk}$ and whether $|x\rangle, |y\rangle$ are scalars or $k$-vectors.

- Recall that $\boldsymbol{x}$ represents a matrix with $\mathbb{R}^{nk}$-vectors as columns, collecting all the random vectors that are recursively generated from the modified NE⊗OR⊤. We write $x^i, i = 1, \dots, |\boldsymbol{x}|$ as its columns (the vectors in the collection), and $\boldsymbol{x}_\alpha, \alpha = 1 \dots, nk$ as its rows (the $\alpha$-th coordinate of all the vectors in the collection).

- For the current set of vectors $\boldsymbol{x} = (x^1, \dots, x^{|\boldsymbol{x}|}) \in \mathbb{R}^{nk \times |\boldsymbol{x}|}$, we write

$$|\boldsymbol{x}\rangle = \left( |x^1\rangle, \dots, |x^{|\boldsymbol{x}|}\rangle \right) \in \mathbb{R}^{k \times |\boldsymbol{x}|}. \tag{18}$$

  We write $|\boldsymbol{x}\rangle_\alpha \in \mathbb{R}^{|\boldsymbol{x}|}$ for the $\alpha$-th row of $|\boldsymbol{x}\rangle$.

With the notation in place, we now introduce the ket-based construction for our modified NE⊗OR⊤ program. This construction tracks the distribution of preactivations at each layer at every training step in the infinite-width limit.

**Definition E.4** (Ket Construction). We recursively define the random $k$-vector $|x\rangle \in \mathbb{R}^k$ (a multi-vector *ket*) for each vector $x \in \mathbb{R}^{nk}$ and deterministic number $\mathring{\theta}$ for each scalar $\theta$ in the program. For a vector $Wx$ produced by MatMul, we also define random $k$-vectors $|Wx\hat{\rangle}$ and $|Wx\dot{\rangle}$ (called *hat-ket* and *dot-ket* respectively) such that $|Wx\rangle = |Wx\hat{\rangle} + |Wx\dot{\rangle}$. Their recursive definitions are given below.

- `Init`: For a vector $x \in \boldsymbol{x}^0$, define the ket

$$|x\rangle := \begin{cases} \mathcal{N}(0, I_k) \in \mathbb{R}^k & \text{if } x \in \boldsymbol{x}^{0,1}, \\ \mathcal{N}(0, 1) \otimes \mathbf{1}_k \in \mathbb{R}^k & \text{if } x \in \boldsymbol{x}^{0,2}. \end{cases} \quad (19)$$

Also, for the scalars, let $\mathring{\boldsymbol{c}}^0 := 0 \in \mathbb{R}^{|\boldsymbol{c}^0|}$.

- `Avg`: If $\theta$ is generated by `Avg` as in (14), then

$$\mathring{\theta} := \frac{1}{k} \sum_{i=1}^{k} \mathbb{E} |x\rangle_i = \frac{1}{k} \operatorname{Tr} \langle x \mid \mathbf{1} \rangle, \quad (20)$$

where the equality follows from our notation of $\langle \bullet \mid \bullet \rangle$ defined in (17).

- `OuterNonlin`: If $y$ is generated by `OuterNonlin` as in (15), then for $\alpha \in [k]$,

$$|y\rangle_\alpha := f(|\boldsymbol{x}\rangle_\alpha) \quad \text{where} \quad f : \mathbb{R}^{|\boldsymbol{x}|} \to \mathbb{R}, \quad f(\boldsymbol{y}) := \frac{1}{k^r} \sum_{\beta_1, \dots, \beta_r = 1}^{k} \mathbb{E}\left[ \psi(\boldsymbol{y}; |\boldsymbol{x}\rangle_{\beta_1}^{\boxed{1}}; \cdots ; |\boldsymbol{x}\rangle_{\beta_r}^{\boxed{r}}; \mathring{\boldsymbol{c}}) \right]. \quad (21)$$

Here, as we have mentioned in the notation part, $|\boldsymbol{x}\rangle_\alpha$ refers to the $\alpha$-th row of the random matrix $|\boldsymbol{x}\rangle \in \mathbb{R}^{k \times |\boldsymbol{x}|}$ that collects the current set of multi-vector kets. Also, $|\boldsymbol{x}\rangle^{\boxed{1}}, \dots, |\boldsymbol{x}\rangle^{\boxed{r}}$ represents $r$ i.i.d. copies of $|\boldsymbol{x}\rangle$.

- `Hat`: All hat-kets are jointly Gaussian with zero-mean and covariance

$$\operatorname{Cov}(|W\hat{x}\rangle, |U\hat{y}\rangle) = \begin{cases} \mathbb{I}(W = U) \operatorname{Tr} \langle x \mid y \rangle I_k & \text{if } W \in \mathcal{W}_1, \\ \mathbb{I}(W = U)(\mathbf{1}_k^\top \langle x \mid y \rangle \mathbf{1}_k)\mathbf{1}_k \mathbf{1}_k^\top & \text{if } W \in \mathcal{W}_2. \end{cases} \quad (22)$$

Here $\mathbb{I}(W = U) = 1$ if and only if $W$ and $U$ are the same matrix as symbol in the program and $0$ otherwise.

- `Dot`: By the construction presented here, each entry of any ket, $|x\rangle_\alpha$, for $\alpha \in [k]$, is always a deterministic function of the set of hat-ket entries $|\hat{y}\rangle_\beta$, where $\beta \in [k]$. As such, $\frac{\partial |x\rangle_\alpha}{\partial |\bullet\rangle_\alpha}$ can be defined symbolically. Moreover, it should be independent of $\alpha$ due to permutation symmetry. Hence, we can use $\frac{\partial |x\rangle}{\partial |\bullet\rangle}$ to denote this value.

Then, every dot-ket can be expressed as a linear combination of previous kets, represented by the following equation:

$$|W\dot{x}\rangle := \begin{cases} k \sum_{y \in \boldsymbol{x}} |y\rangle \mathbb{E}\left[ \frac{\partial |x\rangle}{\partial |W^\top \hat{y}\rangle} \right] & \text{if } W \in \mathcal{W}_1, \\ k \sum_{y \in \boldsymbol{x}} \sum_{j \in [k]} |y\rangle_j \mathbf{1}_k \mathbb{E}\left[ \frac{\partial |x\rangle}{\partial |W^\top \hat{y}\rangle} \right] & \text{if } W \in \mathcal{W}_2. \end{cases} \quad (23)$$

THE MASTER THEOREM

Finally, we state the Master theorem for the modified $\text{NE} \otimes \text{OR} \top$, which is practically the same as in Littwin & Yang (2023).

**Theorem E.5** (Master theorem). *Consider modified $\text{NE} \otimes \text{OR} \top$ with Gaussian or non-Gaussian set-up. Then, as $n \to \infty$, its scalars $c$ satisfy*

$$\boldsymbol{c} \overset{a.s.}{\to} \mathring{\boldsymbol{c}}.$$

*In the non-Gaussian set-up, this convergence also happens in $L^p$ for every $p \in [1, \infty)$. In either setup, if the initial scalars are all $\tilde{O}(n^{-1/2})$, then*

$$\boldsymbol{c} - \mathring{\boldsymbol{c}} = \tilde{O}(n^{-1/2})$$

*as well.*

*Here, for a random sequence $\boldsymbol{c} = \{\boldsymbol{c}(n)\}_{n \geq 1}$ of fixed-sized vectors, we write $\boldsymbol{c} = \tilde{O}(n^h)$ if $n^{-h-\varepsilon} \boldsymbol{c} \overset{a.s.}{\to} 0$ for every $\varepsilon > 0$.*

### E.2. Proof of the Master Theorem for the Modified NE⊗OR⊤

We now prove the main results stated in the previous section. Our strategy is to construct an original NE⊗OR⊤ program that represents the modified NE⊗OR⊤ program (Definition E.1) under either Assumption E.2 or Assumption E.3. This construction provides an explicit correspondence between the two formulations, enabling a direct transfer of the analytical framework. Consequently, the ket construction for the modified NE⊗OR⊤ (Definition E.4) follows directly from the ket construction of the original NE⊗OR⊤ .

*Proof of Theorem E.5.* We begin with a program written in the modified NE⊗OR⊤ . Our objective is to iteratively construct an original NE⊗OR⊤ program (Littwin & Yang, 2023) that expresses the same operations and generates equivalent ket construction. We will use $\mathcal{W}, \boldsymbol{x}, \boldsymbol{c}$ to denote the sets of matrices, vectors, and scalars in the modified NE⊗OR⊤ ; and use $\mathcal{W}', \boldsymbol{x}', \boldsymbol{c}'$ to denote the sets in the constructed original NE⊗OR⊤ .

For simplicity, we introduce two additional operations in the original NE⊗OR⊤ : (1) removing a vector from the set $\boldsymbol{x}'$ and (2) removing a scalar from the set $\boldsymbol{c}'$. In the ket construction, the corresponding limiting objects are also removed. These modifications do not affect the expressivity of the original NE⊗OR⊤ ; their primary purpose is to eliminate intermediate quantities and maintain clean sets.

We now consider the cases of different operations of the modified NE⊗OR⊤ to iteratively reconstruct the original NE⊗OR⊤ program. In each step of the construction, we will prove the following property:

P1 The generated scalars and vectors match: $\boldsymbol{c}' = \boldsymbol{c}$, and $\boldsymbol{x}'$ consists of vectors $x_{(i)} \in \mathbb{R}^n$ for $i \in [k], x \in \boldsymbol{x}$. (Recall the notation $x_{(i)}$ defined in (16).) Alternatively as matrices (where we assume appropriate permutation of columns), $\boldsymbol{x}' \in \mathbb{R}^{n \times k|\boldsymbol{x}|}$ is a reshape of $\boldsymbol{x} \in \mathbb{R}^{nk \times |\boldsymbol{x}|}$.

P2 Moreover, the ket constructions match. Firstly, $\mathring{\boldsymbol{c}} = \mathring{\boldsymbol{c}}'$; Secondly, for vector $x \in \boldsymbol{x}$, its ket $|x\rangle \in \mathbb{R}^k$ in the modified NE⊗OR⊤ as defined in Definition E.4 satisfies $|x\rangle_i = |x_{(i)}\rangle$, where $x_{(i)} \in \boldsymbol{x}'$ with $|x_{(i)}\rangle$ being the corresponding ket in the original NE⊗OR⊤ . Alternatively as matrices (where we assume appropriate permutation of columns), $|\boldsymbol{x}\rangle \in \mathbb{R}^{k \times |\boldsymbol{x}|}$ is a reshape of the vector $|\boldsymbol{x}'\rangle \in \mathbb{R}^{k|\boldsymbol{x}|}$.

The Master Theorem then follows immediately from the Master Theorem of the original NE⊗OR⊤ .

- `Init`: Given initial sets $\boldsymbol{x}^0 \subseteq \mathbb{R}^{nk}$ in the program written in modified NE⊗OR⊤ , we construct an initial set $\boldsymbol{x}^{0'}$ for the original NE⊗OR⊤ in the following way:
    - For each vector $x \in \boldsymbol{x}^{0,1} \subseteq \mathbb{R}^{nk}$, where $x$ has i.i.d. entries drawn from $\mathcal{N}(0, 1)$, we add the vectors $x_{(1)}, \ldots, x_{(k)} \in \mathbb{R}^n$ to the initial set $\boldsymbol{x}^{0'}$.
    - For each vector $x \in \boldsymbol{x}^{0,2} \subseteq \mathbb{R}^{nk}$, which consists of constant $k$-blocks (i.e., $x_{(1)} = \ldots = x_{(k)}$), we add only $x_{(1)}$ to the initial set $\boldsymbol{x}^{0'}$. Later, after the initialization is complete, we will use `OuterNonlin` operations to add $k - 1$ identical copies of $x_{(1)}$ to ensure that all of $x_{(1)}, \ldots, x_{(k)}$ are included in $\boldsymbol{x}'$.

Given the initial set of matrices $\mathcal{W}$ in the program written in our NE⊗OR⊤ , we construct an initial set $\mathcal{W}'$ for the original NE⊗OR⊤ as follows:

- For each matrix $W \in \mathcal{W}_1 \subseteq \mathbb{R}^{nk \times nk}$, which has i.i.d. entries drawn from $\mathcal{N}(0, 1)$, we add the matrices $W_{(1,1)}, \ldots, W_{(k,k)} \in \mathbb{R}^{n \times n}$ to the initial set $\mathcal{W}'$.
- For each matrix $W \in \mathcal{W}_2 \subseteq \mathbb{R}^{nk \times nk}$, which has constant $k \times k$ blocks (i.e., $W_{(1,1)} = \ldots = W_{(k,k)}$), we simply add $W_{(1,1)}$ to the initial set $\mathcal{W}'$.

Finally, for initial scalars $\boldsymbol{c}^0$ in the program written in modified NE⊗OR⊤ , we also use it for the initial scalars for the original NE⊗OR⊤ . That is, let $\boldsymbol{c}^{0'} := \boldsymbol{c}^0$.

After this step of construction, Property P1 immediately satisfies. For each $x \in \boldsymbol{x}^{0,1}$ in modified NE⊗OR⊤ , its ket construction is $|x\rangle \sim \mathcal{N}(0, I_k)$ as defined in (19). This coincides with $(|x_{(1)}\rangle, \ldots, |x_{(k)}\rangle)$, the corresponding ket construction in the constructed original NE⊗OR⊤ . For each $x \in \boldsymbol{x}^{0,2}$ in modified NE⊗OR⊤ , its ket construction is $|x\rangle \sim \mathcal{N}(0, 1) \otimes \mathbf{1}_k$ as defined in (19). This again aligns with $(|x_{(1)}\rangle, \ldots, |x_{(k)}\rangle)$ from the constructed original NE⊗OR⊤ . Hence Property P2 is also satisfied.

- $\texttt{Avg}$: The $\texttt{Avg}$ operation in modified $\text{NE} \otimes \text{OR} \top$ can be expressed in the original $\text{NE} \otimes \text{OR} \top$ as follows: By the inductive hypothesis P1, the chosen vector $x \in \boldsymbol{x}$ has corresponding vectors $x_{(1)}, \ldots, x_{(k)}$ in $\boldsymbol{x}'$. We perform $k$ $\texttt{Avg}$ operations on each of these vectors, which appends $k$ scalars to $\boldsymbol{c}'$. Subsequently, we use an additional $\texttt{OuterNonlin}$ operation followed by another $\texttt{Avg}$ operation to compute the average of the $k$ scalars and append the result to $\boldsymbol{c}'$. Finally, we remove all the intermediate vectors and scalars generated, and leave only the final result in $\boldsymbol{c}'$. This ensure that after this step of construction, Property P1 satisfies.

The ket construction of the constructed original $\text{NE} \otimes \text{OR} \top$ generates deterministic values $\mathbb{E} |x_{(1)}\rangle, \ldots, \mathbb{E} |x_{(k)}\rangle$ during the $k$ $\texttt{Avg}$ operations. It then adds another deterministic value $\frac{1}{k} \sum_{i=1}^{k} \mathbb{E} |x_{(i)}\rangle$ from the final $\texttt{Avg}$ operation. The removals ensure that only the last scalar is added to $\mathring{\boldsymbol{c}}'$. Meanwhile, the ket construction for $\texttt{Avg}$ in the modified $\text{NE} \otimes \text{OR} \top$ generates $\frac{1}{k} \sum_{i=1}^{k} \mathbb{E} |x\rangle_i$ as defined in (20). By the inductive hypothesis, $|x\rangle_i = |x_{(i)}\rangle$ for all $i$. Hence, P2 also satisfies.

- $\texttt{OuterNonlin}$: For each $i \in [k]$, let $\boldsymbol{x}_{(i)}$ denote the set of $\mathbb{R}^n$ vectors formed by taking $x_{(i)}$ for all $x \in \boldsymbol{x}$. Equivalently, consider this as a matrix $\boldsymbol{x}_{(i)} \in \mathbb{R}^{n \times |\boldsymbol{x}|}$ formed by the $i$-th, $(i+k)$-th, ... rows of $\boldsymbol{x} \in \mathbb{R}^{nk \times |\boldsymbol{x}|}$. By the inductive hypothesis P1, $\boldsymbol{x}'$ consists of vectors $x_{(i)}, i \in [k], x \in \boldsymbol{x}$. Let $\boldsymbol{x}'_{(i)}$ denote the subset of $\boldsymbol{x}'$ consisting of $\{x_{(i)} : x \in \boldsymbol{x}\}$. As matrices, $\boldsymbol{x}'_{(i)} = \boldsymbol{x}_{(i)}$.

The $\texttt{OuterNonlin}$ operation (15) can be expressed as

$$y_{(i)} \in \mathbb{R}^n, \quad (y_{(i)})_\alpha = \frac{1}{(nk)^r} \sum_{i_1, \ldots, i_r = 1}^{k} \sum_{\beta_1, \ldots, \beta_r = 1}^{n} \psi \left( (\boldsymbol{x}_{(i)})_\alpha; (\boldsymbol{x}_{(i_1)})_{\beta_1}; \ldots; (\boldsymbol{x}_{(i_r)})_{\beta_r}; \boldsymbol{c} \right)$$

$$= \frac{1}{n^r} \sum_{\beta_1, \ldots, \beta_r = 1}^{n} \phi^{(i)} \left( \boldsymbol{x}'_\alpha; \boldsymbol{x}'_{\beta_1}; \ldots; \boldsymbol{x}'_{\beta_r}; \boldsymbol{c}' \right), \quad i = 1, \ldots, k$$

where the function

$$\phi^{(i)} : \mathbb{R}^{|\boldsymbol{x}|k(r+1)+l} \to \mathbb{R} \quad \text{is given by}$$

$$\phi^{(i)}(\boldsymbol{y}_0; \ldots; \boldsymbol{y}_r; \boldsymbol{c}) = \frac{1}{k^r} \sum_{i_1, \ldots, i_r = 1}^{k} \psi \left( \xi_i(\boldsymbol{y}_0); \xi_{i_1}(\boldsymbol{y}_1); \ldots; \xi_{i_r}(\boldsymbol{y}_r); \boldsymbol{c}' \right).$$

Here $\xi_i : \mathbb{R}^{k|\boldsymbol{x}|} \to \mathbb{R}^{|\boldsymbol{x}|}$ denotes the operation that reshapes the input vector into a matrix in $\mathbb{R}^{k \times |\boldsymbol{x}|}$ and returns its $i$-th row. This shows that, $\texttt{OuterNonlin}$ in our modified $\text{NE} \otimes \text{OR} \top$ can be expressed as $k$ $\texttt{OuterNonlin}$ operations in the original $\text{NE} \otimes \text{OR} \top$. Moreover, Property P1 is satisfied after this step of construction.

The original $\text{NE} \otimes \text{OR} \top$ generates $k$ kets: for $i \in [k]$, $|y_{(i)}\rangle := f^{(i)}(|\boldsymbol{x}'\rangle)$ where the function $f^{(i)} : \mathbb{R}^{k|\boldsymbol{x}|} \to \mathbb{R}$ is given by

$$f^{(i)}(\boldsymbol{y}) = \mathbb{E} \, \phi^{(i)} \left( \boldsymbol{y}; |\boldsymbol{x}'\rangle^{\boxed{1}}; \ldots; |\boldsymbol{x}'\rangle^{\boxed{r}}; \mathring{\boldsymbol{c}}' \right)$$

$$= \frac{1}{k^r} \sum_{i_1, \ldots, i_r = 1}^{k} \mathbb{E} \, \psi \left( \xi_i(\boldsymbol{y}); \xi_{i_1}(|\boldsymbol{x}'\rangle^{\boxed{1}}); \ldots; \xi_{i_r}(|\boldsymbol{x}'\rangle^{\boxed{r}}); \mathring{\boldsymbol{c}}' \right)$$

By inductive hypothesis P2, we have $\xi_i(|\boldsymbol{x}'\rangle) = |\boldsymbol{x}'_{(i)}\rangle = |\boldsymbol{x}\rangle_i$, the $i$-th row of $|\boldsymbol{x}\rangle \in \mathbb{R}^{k \times |\boldsymbol{x}|}$. Moreover, $\mathring{\boldsymbol{c}}' = \mathring{\boldsymbol{c}}$. Hence,

$$= \frac{1}{k^r} \sum_{i_1, \ldots, i_r = 1}^{k} \mathbb{E} \, \psi \left( \xi_i(\boldsymbol{y}); |\boldsymbol{x}\rangle_{i_1}^{\boxed{1}}; \ldots; |\boldsymbol{x}\rangle_{i_r}^{\boxed{r}}; \mathring{\boldsymbol{c}} \right).$$

On the other hand, the modified $\text{NE} \otimes \text{OR} \top$ generates ket $|y\rangle$ as defined in (21), with $|y\rangle_i$ exactly matches $|y_{(i)}\rangle$ from above. Hence, P2 still satisfies after this step of construction.

- $\texttt{MatMul}$: Without loss of generality, consider the matrix multiplication between $W \in \mathcal{W} \subseteq \mathbb{R}^{nk \times nk}$ and $x \in \boldsymbol{x} \subseteq \mathbb{R}^{nk}$. (The case of $W^\top x$ follows in exactly the same way.) Then, for any $i \in [k]$ and $t \in [n]$,

$$\left( (Wx)_{(i)} \right)_t = (Wx)_{(t-1)n+i} = \sum_{s \in [nk]} W_{(t-1)n+i, s} x_s$$

$$= \sum_{j \in [k]} \sum_{s \in [n]} (W_{(i,j)})_{t,s} (x_{(j)})_s = \sum_{j \in [k]} (W_{(i,j)} x_{(j)})_t.$$

This implies that for all $i \in [k]$,

$$(Wx)_{(i)} = \sum_{j \in [k]} W_{(i,j)} x_{(j)}.$$

*Case 1:* $W \in \mathcal{W}_1$.

In this case, $W_{(i,j)}$ are i.i.d. matrices for $i, j \in [k]$. MatMul in our modified $\mathrm{NE} \otimes \mathrm{OR} \top$ can be expressed as $k^2$ MatMul operations in the original $\mathrm{NE} \otimes \mathrm{OR} \top$:

$$W_{(i,j)} x_{(j)}, \quad i, j \in [k].$$

This is followed by $k$ OuterNonlin operations to calculate $\sum_{j \in [k]} W_{(i,j)} x_{(j)}$ for each $i \in [k]$, and to delete the previously appended $k^2$ vectors $\{W_{(i,j)} x_{(j)} : i, j \in [k]\}$ from $\boldsymbol{x}'$. This ensures that property P1 continues to be satisfied after this step of construction.

The original $\mathrm{NE} \otimes \mathrm{OR} \top$ generated the following limiting objects: first, the MatMul operations generate $k^2$ hat-kets, $\left\{ |W_{(i,j)} x_{(j)} \hat{\rangle} : i, j \in [k] \right\}$ and $k^2$ dot-kets, $\left\{ |W_{(i,j)} x_{(j)} \dot{\rangle} : i, j \in [k] \right\}$. For any $Uz$ in our modified $\mathrm{NE} \otimes \mathrm{OR} \top$, and for all $i, j, s, t \in [k]$,

$$\mathrm{Cov}(|W_{(i,j)} x_{(j)} \hat{\rangle}, |U_{(s,t)} z_{(t)} \hat{\rangle}) = \mathbb{I}(W = U) \mathbf{1}(i = s, j = t) \mathbb{E} |x_{(j)}\rangle |z_{(j)}\rangle .$$

Moreover,

$$|W_{(i,j)} x_{(j)} \dot{\rangle} := \sum_{y' \in \boldsymbol{x}'} |y'\rangle \mathbb{E} \frac{\partial |x_{(j)}\rangle}{\partial |(W_{(i,j)})^\top y' \hat{\rangle}}$$

$$= \sum_{y' \in \boldsymbol{x}'} |y'\rangle \mathbb{E} \frac{\partial |x_{(j)}\rangle}{\partial |(W^\top)_{(j,i)} y' \hat{\rangle}}$$

$$= \sum_{y \in \boldsymbol{x}} |y_{(i)}\rangle \mathbb{E} \frac{\partial |x_{(j)}\rangle}{\partial |(W^\top)_{(j,i)} y_{(i)} \hat{\rangle}}$$

since each $y' \in \boldsymbol{x}'$ comes from $y' = y_{(i)}$ for some $y \in \boldsymbol{x}, i \in [k]$, and $(W^\top)_{(j,i)}$ only interacts with $y_{(i)}$.

Following the MatMul, the OuterNonlin operations generate, for each $i \in [k]$,

$$|{\textstyle\sum_{j \in [k]}} W_{(i,j)} x_{(j)}\rangle = \sum_{j \in [k]} |W_{(i,j)} x_{(j)} \hat{\rangle} + \sum_{j \in [k]} |W_{(i,j)} x_{(j)} \dot{\rangle}.$$

To show Property P2 is still satisfied after this step of construction, we need to check that the kets generated in the modified $\mathrm{NE} \otimes \mathrm{OR} \top$, $|Wx \hat{\rangle}, |Wx \dot{\rangle}$ as defined in (22) and (23) matches the kets generated in the original $\mathrm{NE} \otimes \mathrm{OR} \top$:

$$|Wx \hat{\rangle} = \left( \sum_{j \in [k]} |W_{(1,j)} x_{(j)} \hat{\rangle}, \dots, \sum_{j \in [k]} |W_{(k,j)} x_{(j)} \hat{\rangle} \right),$$

$$|Wx \dot{\rangle} = \left( \sum_{j \in [k]} |W_{(1,j)} x_{(j)} \dot{\rangle}, \dots, \sum_{j \in [k]} |W_{(k,j)} x_{(j)} \dot{\rangle} \right).$$

First, consider the hat part. For any $Wz$ used in the modified $\mathrm{NE} \otimes \mathrm{OR} \top$, the corresponding $k \times k$ covariance matrix between the Gaussian vectors on the RHS has the $(i, j)$-th entry being

$$\mathrm{Cov}(\sum_{t \in [k]} |W_{(i,t)} x_{(t)} \hat{\rangle}, \sum_{s \in [k]} |W_{(j,s)} z_{(s)} \hat{\rangle}) = \mathbf{1}(i = j) \sum_{t \in [k]} \mathbb{E} |x_{(t)}\rangle |z_{(t)}\rangle = \mathbf{1}(i = j) \mathrm{Tr} \langle x \mid z \rangle .$$

This exactly matches our definition of $\text{Cov}(|Wx\hat{\rangle}, |Wz\hat{\rangle})_{ij}$ in (22) for $W \in \mathcal{W}_1$.

Second, consider the dot part. The $i$-th entry of the vector on the RHS is equal to

$$\sum_{j\in[k]} |W_{(i,j)}x_{(j)}\dot{\rangle} = \sum_{y\in\boldsymbol{x}} |y_{(i)}\rangle \sum_{j\in[k]} \mathbb{E}\frac{\partial\,|x_{(j)}\rangle}{\partial|(W^\top)_{(j,i)}y_{(i)}\hat{\rangle}}.$$

By induction hypothesis, $|W^\top y\hat{\rangle}_j = \sum_{i\in[k]}|(W^\top)_{(j,i)}y_{(i)}\hat{\rangle}$, $|x\rangle_j = |x_{(j)}\rangle$, and $|y\rangle_i = |y_{(i)}\rangle$ so

$$= \sum_{y\in\boldsymbol{x}} |y\rangle_i \sum_{j\in[k]} \mathbb{E}\frac{\partial\,|x\rangle_j}{\partial|W^\top y\hat{\rangle}_j}$$

$$= k\sum_{y\in\boldsymbol{x}} |y\rangle_i\, \mathbb{E}\frac{\partial\,|x\rangle_1}{\partial|W^\top y\hat{\rangle}_1}.$$

This is exactly the $i$-th entry of $|Wx\dot{\rangle}$ in the modified NE⊗ORT by our definition in (23) for $W \in \mathcal{W}_1$.

*Case 2: $W \in \mathcal{W}_2$.*

In this case, $W_{(i,j)}$ are identical for $i,j \in [k]$. MatMul in our modified NE⊗ORT can be expressed as, first an OuterNonlin operation to calculate, $\sum_{j\in[k]} x_{(j)}$, followed by MatMul operations in the original NE⊗ORT: $W_{(1,1)}\sum_{j\in[k]} x_{(j)}$, and finally delete the previously appended vector. Finally, we use one more OuterNonlin operation to add $k-1$ identical copies of the final vector. This ensures that property P1 continues to be satisfied after this step of construction.

The original NE⊗ORT generated the following limiting objects: First, the OuterNonlin operation generates ket $|\sum_{j\in[k]} x_{(j)}\rangle = \sum_{j\in[k]} |x_{(j)}\rangle$; then, MatMul operations generate hat-kets, $|W_{(1,1)}\sum_j x_{(j)}\hat{\rangle}$, and dot-kets, $|W_{(1,1)}\sum_j x_{(j)}\dot{\rangle}$. For any $Uz$ in our modified NE⊗ORT, we have

$$\text{Cov}(|W_{(1,1)}\textstyle\sum_i x_{(i)}\hat{\rangle}, |U_{(1,1)}\textstyle\sum_j z_{(j)}\hat{\rangle}) = \mathbb{I}(W=U)\sum_{i,j} \langle x_{(i)} \mid z_{(j)}\rangle.$$

Moreover,

$$|W_{(1,1)}\textstyle\sum_i x_{(i)}\dot{\rangle} := \sum_{y'\in\boldsymbol{x}'} \sum_{i\in[k]} |y'\rangle\, \mathbb{E}\frac{\partial\,|x_{(i)}\rangle}{\partial|(W_{(1,1)})^\top y'\hat{\rangle}}$$

$$= \sum_{y\in\boldsymbol{x}} \sum_{j\in[k]} \sum_{i\in[k]} |y_{(j)}\rangle\, \mathbb{E}\frac{\partial\,|x_{(i)}\rangle}{\partial|(W^\top)_{(1,1)}y_{(j)}\hat{\rangle}}.$$

since each $y' \in \boldsymbol{x}'$ comes from $y' = y_{(j)}$ for some $y \in \boldsymbol{x}, j \in [k]$, and $(W^\top)_{(1,1)}$ interacts with every $y_{(j)}$.

Following the MatMul, the OuterNonlin operations generate a random variable

$$|W_{(1,1)}\textstyle\sum_i x_{(i)}\rangle = |W_{(1,1)}\textstyle\sum_i x_{(i)}\hat{\rangle} + |W_{(1,1)}\textstyle\sum_i x_{(i)}\dot{\rangle}.$$

To show Property P2 is still satisfied after this step of construction, we need to check that the kets generated in the modified NE⊗ORT, $|Wx\hat{\rangle}, |Wx\dot{\rangle}$ as defined in (22) and (23) matches the kets generated in the original NE⊗ORT:

$$|Wx\hat{\rangle} = |W_{(1,1)}\textstyle\sum_i x_{(i)}\hat{\rangle}\mathbf{1}_k, \qquad \text{and} \qquad |Wx\dot{\rangle} = |W_{(1,1)}\textstyle\sum_i x_{(i)}\dot{\rangle}\mathbf{1}_k.$$

First, let us consider the hat ket. For any $Wz$ used in the modified NE⊗ORT, the corresponding $k \times k$ covariance matrix between the Gaussian vectors on the RHS has constant entries which are given by

$$\text{Cov}(|W_{(1,1)}\textstyle\sum_i x_{(i)}\hat{\rangle}, |W_{(1,1)}\textstyle\sum_j z_{(j)}\hat{\rangle}) = \sum_{i,j} \langle x_{(i)} \mid z_{(j)}\rangle = \mathbf{1}_k^\top \langle x \mid z\rangle \mathbf{1}_k$$

This exactly matches our definition of $\text{Cov}(|Wx\hat\rangle, |Wz\hat\rangle)$ in (22) for $W \in \mathcal{W}_2$.

Second, let us consider the dot-ket. The vector on the RHS has constant entry, which is

$$|W_{(1,1)} \textstyle\sum_i x_{(i)}\dot\rangle = \sum_{y\in \boldsymbol{x}} \sum_{j\in[k]} \sum_{i\in[k]} |y_{(j)}\rangle \, \mathbb{E} \frac{\partial\,|x_{(i)}\rangle}{\partial|(W^\top)_{(1,1)} y_{(j)}\hat\rangle}$$

$$= \sum_{y\in \boldsymbol{x}} \sum_{j\in[k]} |y\rangle_j \sum_{i\in[k]} \mathbb{E} \frac{\partial\,|x\rangle_i}{\partial|W^\top y\hat\rangle_i}$$

since by induction hypothesis, $|W^\top y\hat\rangle_j = |(W^\top)_{(1,1)} \sum_i y_{(i)}\hat\rangle$, $|x\rangle_j = |x_{(j)}\rangle$ and $|y\rangle_j = |y_{(j)}\rangle$. Continuing,

$$= k \sum_{y\in \boldsymbol{x}} \sum_{j\in[k]} |y\rangle_j \, \mathbb{E} \frac{\partial\,|x\rangle_1}{\partial|W^\top y\hat\rangle_1}.$$

This is exactly the entries of $|Wx\dot\rangle$ in the modified $\textsc{Ne}\otimes\textsc{or}\top$ (23) for $W \in \mathcal{W}_2$. $\qquad\square$

### E.3. Example: 3-Layer MLP Without Nonlinearities and with Input and Output Weights Frozen

**Set-up.** Consider a 3-layer MLP without nonlinearity initialized and trained under $\mu$P in Table 2. The forward pass is given by

$$h^{(1)} = Cx, \quad h^{(2)} = Bh^{(1)}, \quad y = \frac{1}{n} Ah^{(2)},$$

where $A \in \mathbb{R}^{1\times n}, B \in \mathbb{R}^{n\times n}, C \in \mathbb{R}^{n\times 1}$ are the weights. The random initialization at $t = 0$ is

$$(A_0)_{ij} \sim \mathcal{N}\left(0, \overline\sigma^2\right), \quad (B_0)_{ij} \sim \mathcal{N}\left(0, n^{-1}\overline\sigma^2\right), \quad (C_0)_{ij} \sim \mathcal{N}\left(0, \overline\sigma^2\right),$$

We fix the input and output weights $A, C$ and only train the hidden weight $B$. For simplicity of notation, we assume full gradient descent on a dataset of size 1, but the same works for any dataset of fixed size; it can also be extended SGD on mini-batches (see Setup 2.3.1 in Littwin & Yang (2023)).

We first train this MLP using vanilla SGD for training steps $t < T$. We write $A_t$, $B_t$, $C_t$, $h_t^{(1)}$, $h_t^{(2)}$, and $y_t$ for the corresponding quantities at step $t$. We write $A_t^\uparrow, B_t^\uparrow, C_t^\uparrow, h_t^{\uparrow(1)}, h_t^{\uparrow(2)}$ for the equivalent widened version of width $N = nk$, as specified in Table 1. That is,

$$A_t^\uparrow := A_t \otimes \mathbf{1}_k^\top, \quad B_t^\uparrow := k^{-1} B_t \otimes \mathbf{1}_k \mathbf{1}_k^\top, \quad C_t^\uparrow := C_t \otimes \mathbf{1}_k, \quad h_t^{\uparrow(1)} := h_t^{(1)} \otimes \mathbf{1}_k, \quad h_t^{\uparrow(2)} := h_t^{(2)} \otimes \mathbf{1}_k.$$

In the backpropagation, the gradient is computed by

$$dB_t = dh_t^{(2)} (h_t^{(1)})^\top = n^{-1} \mathcal{L}'(y_t) x A_0^\top C_0^\top,$$

The weight is updated by $B_{t+1} = B_t - \overline\gamma dB_t$. Other weights $A, C$ are not trained, so we take $A_t = A_0, C_t = C_0$ for all $t < T$.

At step $T$, we apply upscaling. Specifically, we set

$$A_T^\uparrow := A_{T-1}^\uparrow + \Delta_A, \quad B_T^\uparrow := B_{T-1}^\uparrow + \Delta_B, \quad C_T^\uparrow := C_{T-1}^\uparrow + \Delta_C,$$

where matrices $\Delta_A \in \mathbb{R}^{1\times N}, \Delta_B \in \mathbb{R}^{N\times N}, \Delta_C \in \mathbb{R}^{N\times 1}$ have i.i.d. entries

$$(\Delta_A)_{ij} \sim \mathcal{N}\left(0, \overline{\sigma_{\Delta,A}}^2\right), \quad (\Delta_B)_{ij} \sim \mathcal{N}\left(0, N^{-1}\overline{\sigma_{\Delta,B}}^2\right), \quad (\Delta_C)_{ij} \sim \mathcal{N}\left(0, \overline{\sigma_{\Delta,C}}^2\right).$$

To facilitate theoretical analysis of the role of noise, here we assume that the weights $A$, $B$, and $C$ are perturbed by additive noise with potentially different magnitudes.

Finally, we train the upscaled model for training step $t \geq T$. We apply gradient descent on $B_t^\uparrow$.

$$B_{t+1}^\uparrow = B_t^\uparrow - \overline{\gamma^\uparrow} dB_t^\uparrow,$$

where the gradient

$$dB_t^{\uparrow} = dh_t^{\uparrow(2)}(h_t^{\uparrow(1)})^{\top} = N^{-1}\mathcal{L}'(y_t)xA_T^{\uparrow\top}C_T^{\uparrow\top}.$$

Meanwhile, take $A_t = A_T, C_t = C_T$ for all $t > T$.

Note that the size-$n$ quantities stop to be tracked at step $t = T$. All the size-$N$ quantities are all denoted with $\uparrow$, and they exist for all training steps.

**Before upscaling.** We apply the original $\text{NE}\otimes\text{OR}\top$ from Littwin & Yang (2023) to analyze the infinite-width training dynamics prior to upscaling. At each training step $t < T$, we have

$$h_t^{(1)} = C_0 x, \quad h_t^{(2)} = \left(B_0 - \bar{\gamma}n^{-1}x\sum_{s=0}^{t-1}\mathcal{L}'(y_s)A_0^{\top}C_0^{\top}\right)h_t^{(1)}, \quad y_t = \frac{1}{n}A_0 h_t^{(2)}.$$

The first pre-activation $h_t^{(1)}$ remains constant for all $t$, and its distribution is tracked by

$$|h_t^{(1)}\rangle = x\,|C_0\rangle \sim \mathcal{N}\left(0, x^2\bar{\sigma}^2\right).$$

It follows that

$$h_t^{(2)} = B_0 h_0^{(1)} - \bar{\gamma}x^2\sum_{s=0}^{t-1}\mathcal{L}'(y_s)\frac{C_0^{\top}C_0}{n}A_0^{\top}.$$

Hence, the distribution of the second pre-activation $h_t^{(2)}$ is tracked by

$$|h_t^{(2)}\rangle = |B_0 h_0^{(1)}\rangle - \bar{\gamma}x^2\sum_{s=0}^{t-1}\mathcal{L}'(\mathring{y}_s)\underbrace{\langle C_0 \mid C_0\rangle}_{=\bar{\sigma}^2}|A_0\rangle,$$

where $|B_0 h_0^{(1)}\rangle \sim \mathcal{N}\left(0, x^2\bar{\sigma}^4\right)$ and $|A_0\rangle \sim \mathcal{N}\left(0, \overline{\sigma^2}\right)$ are independent. The output $y_t$ converges to the deterministic scalar

$$\mathring{y}_t = \langle A_0 \mid h_t^{(2)}\rangle = -\bar{\gamma}x^2\bar{\sigma}^4\sum_{s=0}^{t-1}\mathcal{L}'(\mathring{y}_s). \tag{24}$$

In this simple model, the infinite-width training dynamics is fully characterized by the recursion (24), and no intermediate distributions need to be tracked.

**After upscaling.** We analyze the post-upscaling regime using our modified $\text{NE}\otimes\text{OR}\top$. At each training step $t \geq T$, we have

$$h_t^{\uparrow(1)} = (C_0^{\uparrow} + \Delta_C)x,$$

$$h_t^{\uparrow(2)} = \left(B_0^{\uparrow} - \bar{\gamma}N^{-1}x\sum_{s=0}^{T-2}\mathcal{L}'(y_s)A_0^{\uparrow\top}C_0^{\uparrow\top}\right.$$

$$\left. + \Delta_B - \overline{\gamma^{\uparrow}}N^{-1}x\sum_{s=T}^{t-1}\mathcal{L}'(y_s)(A_0^{\uparrow} + \Delta_A)^{\top}(C_0^{\uparrow} + \Delta_C)^{\top}\right)h_t^{\uparrow(1)},$$

$$y_t = \frac{1}{N}(A_0^{\uparrow} + \Delta_A)h_t^{\uparrow(2)}.$$

The first pre-activation $h_t^{\uparrow(1)}$ remains constant for all $t \geq T$, and its distribution is tracked by

$$|h_t^{\uparrow(1)}\rangle = x\,|C_0^{\uparrow}\rangle + x\,|\Delta_C\rangle,$$

where

$$|C_0^{\uparrow}\rangle \sim \mathcal{N}\left(0, \bar{\sigma}^2\right)\mathbf{1}_k, \quad |\Delta_C\rangle \sim \mathcal{N}\left(0, \overline{\sigma_{\Delta,C}}^2 I_k\right).$$

It follows that

$$h_t^{\uparrow(2)} = B_0^{\uparrow} h_T^{\uparrow(1)} - \overline{\gamma} x \sum_{s=0}^{T-2} \mathcal{L}'(y_s) A_0^{\uparrow\top} \frac{C_0^{\uparrow\top} h_T^{\uparrow(1)}}{N}$$

$$+ \Delta_B h_T^{\uparrow(1)} - \overline{\gamma^{\uparrow}} x \sum_{s=T}^{t-1} \mathcal{L}'(y_s)(A_0^{\uparrow} + \Delta_A)^{\top} \frac{(C_0^{\uparrow} + \Delta_C)^{\top} h_T^{\uparrow(1)}}{N}.$$

Since $\frac{1}{k} \operatorname{Tr} \langle C_0^{\uparrow} \mid h_T^{\uparrow(1)} \rangle = x^2 \overline{\sigma}^2$ and $\frac{1}{k} \operatorname{Tr} \langle C_0^{\uparrow} + \Delta_C \mid h_T^{\uparrow(1)} \rangle = x^2 (\overline{\sigma}^2 + \overline{\sigma_{\Delta,C}}^2)$, the distribution of the second pre-activation $h_t^{\uparrow(2)}$ is tracked by

$$|h_t^{\uparrow(2)}\rangle = |B_0^{\uparrow} h_T^{\uparrow(1)}\rangle - \overline{\gamma} x^2 \overline{\sigma}^2 \sum_{s=0}^{T-2} \mathcal{L}'(\mathring{y}_s) |A_0^{\uparrow}\rangle$$

$$+ |\Delta_B h_T^{\uparrow(1)}\rangle - \overline{\gamma^{\uparrow}} x^2 (\overline{\sigma}^2 + \overline{\sigma_{\Delta,C}}^2) \sum_{s=T}^{t-1} \mathcal{L}'(\mathring{y}_s) \left( |A_0^{\uparrow}\rangle + |\Delta_A\rangle \right),$$

where $|B_0^{\uparrow} h_T^{\uparrow(1)}\rangle$, $|\Delta_B h_T^{\uparrow(1)}\rangle$, $|A_0^{\uparrow}\rangle$, and $|\Delta_A\rangle$ are independent Gaussian vectors. The output $y_t$ converges to the deterministic scalar

$$\mathring{y}_t = \langle A_0^{\uparrow} \mid h_t^{\uparrow(2)} \rangle + \langle \Delta_A \mid h_t^{\uparrow(2)} \rangle = -\overline{\gamma} x^2 \overline{\sigma}^4 \sum_{s=0}^{T-2} \mathcal{L}'(\mathring{y}_s) - \overline{\gamma^{\uparrow}} x^2 (\overline{\sigma}^2 + \overline{\sigma_{\Delta,C}}^2)(\overline{\sigma}^2 + \overline{\sigma_{\Delta,A}}^2) \sum_{s=T}^{t-1} \mathcal{L}'(\mathring{y}_s). \quad (25)$$

**Conclusion.** Comparing (24) and (25) shows that the infinite-width training dynamics before and after upscaling coincide if we have the following relation among the hyperparameters:

$$\overline{\gamma}\,\overline{\sigma}^4 = \overline{\gamma^{\uparrow}}(\overline{\sigma}^2 + \overline{\sigma_{\Delta,C}}^2)(\overline{\sigma}^2 + \overline{\sigma_{\Delta,A}}^2).$$

In particular, if $\overline{\sigma_{\Delta,A}} = \overline{\sigma_{\Delta,C}} = 0$ (i.e., no noise added to the frozen weights $A$ or $C$, with noise added only to $B$), then for any $\overline{\sigma_{\Delta,B}}$, choosing $\overline{\gamma^{\uparrow}} = \overline{\gamma}$ (the same learning-rate base constant after upscaling) yields equivalent dynamics. Further, even if we add a substantial amount of noise, it is possible to adjust the learning rates accordingly so that the infinite-width limit is preserved. We note that this equivalence does not imply that upscaling is inconsequential, because the above equivalence only pertains to infinite-width limits, and we expect increased width to reduce finite-width errors and thus yield performance closer to those infinite-width limits.

### E.4. Example: 4-Layer MLP Without Nonlinearities and with Input and Output Weights Frozen

**Set-up.** We now consider a slightly more complex MLP that adds one additional trainable hidden layer relative to the previous architecture. The forward pass at training step $t$ is

$$h_t^{(1)} = Dx, \quad h_t^{(2)} = Ch_t^{(1)}, \quad h_t^{(3)} = Bh_t^{(2)}, \quad y_t = n^{-1}Ah_t^{(3)},$$

where $A \in \mathbb{R}^{1 \times n}$, $B \in \mathbb{R}^{n \times n}$, $C \in \mathbb{R}^{n \times n}$, and $D \in \mathbb{R}^{n \times 1}$ denote the layer weights. At $t = 0$, the random initialization is

$$(A_0)_{ij} \sim \mathcal{N}\left(0, \overline{\sigma}^2\right), \quad (B_0)_{ij} \sim \mathcal{N}\left(0, n^{-1}\overline{\sigma}^2\right), \quad (C_0)_{ij} \sim \mathcal{N}\left(0, n^{-1}\overline{\sigma}^2\right), \quad (D_0)_{ij} \sim \mathcal{N}\left(0, \overline{\sigma}^2\right).$$

We fix the input and output weights $A$ and $D$ and train only the hidden-layer weights $B$ and $C$.

For steps $t < T$, we optimize the width-$n$ model using vanilla SGD. The gradients are computed via backpropagation as

$$dh_t^{(3)} = n^{-1}\mathcal{L}'(y_t)A_t^{\top}, \quad dh_t^{(2)} = B_t^{\top} dh_t^{(3)}, \quad dh_t^{(1)} = C_t^{\top} dh_t^{(2)},$$

$$dB_t = dh_t^{(3)}(h_t^{(2)})^{\top}, \quad dC_t = dh_t^{(2)}(h_t^{(1)})^{\top},$$

and the weights are updated with learning rate $\overline{\gamma}$ according to

$$B_{t+1} = B_t - \overline{\gamma}dB_t, \quad C_{t+1} = C_t - \overline{\gamma}dC_t.$$

We write $\uparrow$ for the equivalent widened version of width $N = nk$, as specified in Table 1. That is, for $t \leq T$,

$$A_t^{\uparrow} := A_t \otimes \mathbf{1}_k^{\top}, \quad B_t^{\uparrow} := k^{-1}B_t \otimes \mathbf{1}_k\mathbf{1}_k^{\top}, \quad C_t^{\uparrow} := k^{-1}C_t \otimes \mathbf{1}_k\mathbf{1}_k^{\top}, \quad D_t^{\uparrow} := D_t \otimes \mathbf{1}_k;$$

$$h_t^{\uparrow(j)} := h_t^{(j)} \otimes \mathbf{1}_k, \quad dh_t^{\uparrow(j)} := k^{-1}dh_t^{(j)} \otimes \mathbf{1}_k \quad \text{for } j = 1, 2, 3;$$

$$dB_t^{\uparrow} := k^{-1}dB_t \otimes \mathbf{1}_k\mathbf{1}_k^{\top}, \quad dC_t^{\uparrow} := k^{-1}dC_t \otimes \mathbf{1}_k\mathbf{1}_k^{\top}.$$

At step $T$, we apply upscaling to width $N$. Specifically, we set

$$A_T^{\uparrow} := A_{T-1}^{\uparrow} + \Delta_A, \quad B_T^{\uparrow} := B_{T-1}^{\uparrow} + \Delta_B, \quad C_T^{\uparrow} := C_{T-1}^{\uparrow} + \Delta_C, \quad D_T^{\uparrow} := D_{T-1}^{\uparrow} + \Delta_D,$$

where the increment matrices $\Delta_A \in \mathbb{R}^{1 \times N}, \Delta_B \in \mathbb{R}^{N \times N}, \Delta_C \in \mathbb{R}^{N \times N}$, and $\Delta_D \in \mathbb{R}^{N \times 1}$ have i.i.d. entries distributed as

$$(\Delta_A)_{ij} \sim \mathcal{N}\left(0, \overline{\sigma_{\Delta,A}}^2\right), \quad (\Delta_B)_{ij} \sim \mathcal{N}\left(0, N^{-1}\overline{\sigma_{\Delta,B}}^2\right),$$

$$(\Delta_C)_{ij} \sim \mathcal{N}\left(0, N^{-1}\overline{\sigma_{\Delta,C}}^2\right), \quad (\Delta_D)_{ij} \sim \mathcal{N}\left(0, \overline{\sigma_{\Delta,D}}^2\right).$$

Finally, for all $t \geq T$, we train the upscaled model. The gradients are computed by

$$dh_t^{\uparrow(3)} = N^{-1}\mathcal{L}'(y_t)A_t^{\uparrow\top}, \quad dh_t^{\uparrow(2)} = B_t^{\uparrow\top}dh_t^{\uparrow(3)}, \quad dh_t^{\uparrow(1)} = C_t^{\uparrow\top}dh_t^{\uparrow(2)},$$

$$dB_t^{\uparrow} = dh_t^{\uparrow(3)}(h_t^{\uparrow(2)})^{\top}, \quad dC_t^{\uparrow} = dh_t^{\uparrow(2)}(h_t^{\uparrow(1)})^{\top},$$

and the weights are updated with learning rate $\overline{\gamma^{\uparrow}}$ as

$$B_{t+1}^{\uparrow} = B_t^{\uparrow} - \overline{\gamma^{\uparrow}}dB_t^{\uparrow}, \quad C_{t+1}^{\uparrow} = C_t^{\uparrow} - \overline{\gamma^{\uparrow}}dC_t^{\uparrow}.$$

**Before upscaling.** We apply the original NE⊗OR⊤ of Littwin & Yang (2023) to characterize the infinite-width training dynamics prior to upscaling. For each training step $t < T$, the following forward- and backward-propagation relations hold.

$$h_t^{(1)} = D_0 x,$$

$$h_t^{(2)} = \left(C_0 - \overline{\gamma}\sum_{s=0}^{t-1} dh_s^{(2)}(h_s^{(1)})^{\top}\right)h_t^{(1)},$$

$$h_t^{(3)} = \left(B_0 - \overline{\gamma}\sum_{s=0}^{t-1} dh_s^{(3)}(h_s^{(2)})^{\top}\right)h_t^{(2)},$$

$$y_t = n^{-1}A_0 h_t^{(3)},$$

$$dh_t^{(3)} = n^{-1}\mathcal{L}'(y_t)A_t^{\top},$$

$$dh_t^{(2)} = \left(B_0 - \overline{\gamma}\sum_{s=0}^{t-1} dh_s^{(3)}(h_s^{(2)})^{\top}\right)^{\top}dh_t^{(3)},$$

$$dh_t^{(1)} = \left(C_0 - \overline{\gamma}\sum_{s=0}^{t-1} dh_s^{(2)}(h_s^{(1)})^{\top}\right)^{\top}dh_t^{(2)}.$$

We track the distributions of these vectors using kets: [6]

$$|h_t^{(1)}\rangle = x|D_0\rangle \sim \mathcal{N}\left(0, x^2\overline{\sigma}^2\right),$$

---

[6]Here $|dh_t^{(j)}\rangle$ tracks the distribution of $n\,dh_t^{(j)}/\mathcal{L}'(y_t)$ for $j = 1, 2, 3$.

$$|h_t^{(2)}\rangle = |C_0 h_0^{(1)}\hat{\rangle} - \overline{\gamma}\sum_{s=0}^{t-1}\mathcal{L}'(\mathring{y}_s)\underbrace{\langle h_s^{(1)} \mid h_t^{(1)}\rangle}_{=x^2\overline{\sigma}^2}|dh_s^{(2)}\rangle,$$

$$|h_t^{(3)}\rangle = |B_0 h_t^{(2)}\hat{\rangle} + |B_0 h_t^{(2)}\dot{\rangle} - \overline{\gamma}\sum_{s=0}^{t-1}\mathcal{L}'(\mathring{y}_s)\langle h_s^{(2)} \mid h_t^{(2)}\rangle\underbrace{|dh_s^{(3)}\rangle}_{=|A_0\rangle},$$

$$\mathring{y}_t = \langle A_0 \mid h_t^{(3)}\rangle,$$

$$|dh_t^{(3)}\rangle = |A_0\rangle,$$

$$|dh_t^{(2)}\rangle = |B_0^\top dh_0^{(3)}\hat{\rangle} - \overline{\gamma}\sum_{s=0}^{t-1}\mathcal{L}'(\mathring{y}_s)\underbrace{\langle dh_s^{(3)} \mid dh_t^{(3)}\rangle}_{=\overline{\sigma}^2}|h_s^{(2)}\rangle,$$

$$|dh_t^{(1)}\rangle = |C_0^\top dh_t^{(2)}\hat{\rangle} + |C_0^\top dh_t^{(2)}\dot{\rangle} - \overline{\gamma}\sum_{s=0}^{t-1}\mathcal{L}'(\mathring{y}_s)\langle dh_s^{(2)} \mid dh_t^{(2)}\rangle\underbrace{|h_s^{(1)}\rangle}_{=x|D_0\rangle}.$$

By substitution, $|h_t^{(2)}\rangle$ satisfies the recursion

$$|h_t^{(2)}\rangle = |C_0 h_0^{(1)}\hat{\rangle} - \overline{\gamma}x^2\overline{\sigma}^2\sum_{s=0}^{t-1}\mathcal{L}'(\mathring{y}_s)\left(|B_0^\top dh_0^{(3)}\hat{\rangle} - \overline{\gamma}\,\overline{\sigma}^2\sum_{\ell=0}^{s-1}\mathcal{L}'(\mathring{y}_\ell)|h_\ell^{(2)}\rangle\right).$$

Therefore, $|h_t^{(2)}\rangle$ admits the decomposition

$$|h_t^{(2)}\rangle = M_t|C_0 h_0^{(1)}\hat{\rangle} + N_t|B_0^\top dh_0^{(3)}\hat{\rangle}. \tag{26}$$

The coefficients $M_t$ and $N_t$ satisfy the recursion

$$M_t = 1 + \overline{\gamma}^2 x^2 \overline{\sigma}^4 \sum_{s=1}^{t-1}\mathcal{L}'(\mathring{y}_s)\sum_{\ell=0}^{s-1}\mathcal{L}'(\mathring{y}_\ell)M_\ell, \quad N_t = -\overline{\gamma}x^2\overline{\sigma}^2\sum_{s=0}^{t-1}\mathcal{L}'(\mathring{y}_s) + \overline{\gamma}^2 x^2 \overline{\sigma}^4 \sum_{s=1}^{t-1}\mathcal{L}'(\mathring{y}_s)\sum_{\ell=0}^{s-1}\mathcal{L}'(\mathring{y}_\ell)N_\ell. \tag{27}$$

Using the definition of $|\bullet\dot{\rangle}$, since $B_0/\overline{\sigma}$ has i.i.d. entries in $\mathcal{N}\left(0, n^{-1}\right)$, we have

$$|B_0 h_t^{(2)}\dot{\rangle} = \overline{\sigma}\,\mathbb{E}\left[\frac{\partial\,|h_t^{(2)}\rangle}{\partial\,|\overline{\sigma}^{-1}B_0^\top dh_0^{(3)}\rangle}\right]|dh_0^{(3)}\rangle = \overline{\sigma}^2 N_t\,|A_0\rangle.$$

Moreover, the bra-ket evaluates to

$$\langle h_s^{(2)} \mid h_t^{(2)}\rangle = M_s M_t \overline{\sigma}^4 x^2 + N_s N_t \overline{\sigma}^4 =: \textcircled{1}.$$

Finally, we obtain

$$\mathring{y}_t = \overline{\sigma}^2 N_t\langle A_0 \mid A_0\rangle - \overline{\gamma}\sum_{s=0}^{t-1}\mathcal{L}'(\mathring{y}_s)\langle h_s^{(2)} \mid h_t^{(2)}\rangle\langle A_0 \mid A_0\rangle$$
$$= \overline{\sigma}^4 N_t - \overline{\gamma}\,\overline{\sigma}^2\sum_{s=0}^{t-1}\mathcal{L}'(\mathring{y}_s)\textcircled{1}. \tag{28}$$

**After upscaling.** We apply the modified $\text{NE}\otimes\text{OR}\top$ to characterize the infinite-width training dynamics after upscaling. For $t \geq T$, the forward- and backward-propagation relations are

$$h_t^{\uparrow(1)} = (D_0^\uparrow + \Delta_D)x,$$
$$h_t^{\uparrow(2)} = \left(C_0^\uparrow - \overline{\gamma}\sum_{s=0}^{T-2}dh_s^{\uparrow(2)}(h_s^{\uparrow(1)})^\top + \Delta_C - \overline{\gamma^\uparrow}\sum_{s=T}^{t-1}dh_s^{\uparrow(2)}(h_s^{\uparrow(1)})^\top\right)h_t^{\uparrow(1)},$$

$$h_t^{\uparrow(3)} = \left(B_0^\uparrow - \overline{\gamma}\sum_{s=0}^{T-2} dh_s^{\uparrow(3)}(h_s^{\uparrow(2)})^\top + \Delta_B - \overline{\gamma^\uparrow}\sum_{s=T}^{t-1} dh_s^{\uparrow(3)}(h_s^{\uparrow(2)})^\top\right)h_t^{\uparrow(2)},$$

$$y_t = N^{-1}(A_0^\uparrow + \Delta_A)h_t^{\uparrow(3)},$$

$$dh_t^{\uparrow(3)} = N^{-1}\mathcal{L}'(y_t)(A_0^\uparrow + \Delta_A)^\top,$$

$$dh_t^{\uparrow(2)} = \left(B_0^\uparrow - \overline{\gamma}\sum_{s=0}^{T-2} dh_s^{\uparrow(3)}(h_s^{\uparrow(2)})^\top + \Delta_B - \overline{\gamma^\uparrow}\sum_{s=T}^{t-1} dh_s^{\uparrow(3)}(h_s^{\uparrow(2)})^\top\right)^\top dh_T^{\uparrow(3)},$$

$$dh_t^{\uparrow(1)} = \left(C_0^\uparrow - \overline{\gamma}\sum_{s=0}^{T-2} dh_s^{\uparrow(2)}(h_s^{\uparrow(1)})^\top + \Delta_C - \overline{\gamma^\uparrow}\sum_{s=T}^{t-1} dh_s^{\uparrow(2)}(h_s^{\uparrow(1)})^\top\right)^\top dh_t^{\uparrow(2)}.$$

We track the distributions of the preactivations and backpropagated signals using the multi-vector kets: [7]

$$|h_t^{\uparrow(1)}\rangle = x\left(|D_0^\uparrow\rangle + |\Delta_D\rangle\right),$$

$$|h_t^{\uparrow(2)}\rangle = |C_0^\uparrow h_T^{\uparrow(1)}\rangle^\wedge - \overline{\gamma}\sum_{s=0}^{T-2}\mathcal{L}'(\mathring{y}_s)\underbrace{\frac{\mathrm{Tr}\,\langle h_s^{\uparrow(1)}\mid h_T^{\uparrow(1)}\rangle}{k}}_{=x^2\overline{\sigma}^2}|dh_s^{\uparrow(2)}\rangle$$

$$+ |\Delta_C h_T^{\uparrow(1)}\rangle^\wedge - \overline{\gamma^\uparrow}\sum_{s=T}^{t-1}\mathcal{L}'(\mathring{y}_s)\underbrace{\frac{\mathrm{Tr}\,\langle h_s^{\uparrow(1)}\mid h_T^{\uparrow(1)}\rangle}{k}}_{=x^2(\overline{\sigma}^2+\sigma_{\Delta,D}{}^2)}|dh_s^{\uparrow(2)}\rangle,$$

$$|h_t^{\uparrow(3)}\rangle = |B_0^\uparrow h_t^{\uparrow(2)}\rangle^\wedge + |B_0^\uparrow h_t^{\uparrow(2)}\rangle^\cdot - \overline{\gamma}\sum_{s=0}^{T-2}\mathcal{L}'(\mathring{y}_s)\frac{\mathrm{Tr}\,\langle h_s^{\uparrow(2)}\mid h_t^{\uparrow(2)}\rangle}{k}\underbrace{|dh_s^{\uparrow(3)}\rangle}_{=|A_0^\uparrow\rangle}$$

$$+ |\Delta_B h_t^{\uparrow(2)}\rangle^\wedge + |\Delta_B h_t^{\uparrow(2)}\rangle^\cdot - \overline{\gamma^\uparrow}\sum_{s=T}^{t-1}\mathcal{L}'(\mathring{y}_s)\frac{\mathrm{Tr}\,\langle h_s^{\uparrow(2)}\mid h_t^{\uparrow(2)}\rangle}{k}\underbrace{|dh_s^{\uparrow(3)}\rangle}_{=|A_0^\uparrow\rangle+|\Delta_A\rangle},$$

$$\mathring{y}_t = \frac{\mathrm{Tr}\,\langle A_0^\uparrow\mid h_t^{\uparrow(3)}\rangle + \mathrm{Tr}\,\langle\Delta_A\mid h_t^{\uparrow(3)}\rangle}{k},$$

$$|dh_t^{\uparrow(3)}\rangle = |A_0^\uparrow\rangle + |\Delta_A\rangle,$$

$$|dh_t^{\uparrow(2)}\rangle = |B_0^{\uparrow\top} dh_T^{\uparrow(3)}\rangle^\wedge - \overline{\gamma}\sum_{s=0}^{T-2}\mathcal{L}'(\mathring{y}_s)\underbrace{\frac{\mathrm{Tr}\,\langle dh_s^{\uparrow(3)}\mid dh_t^{\uparrow(3)}\rangle}{k}}_{=\overline{\sigma}^2}|h_s^{\uparrow(2)}\rangle$$

$$+ |\Delta_B^\top dh_T^{\uparrow(3)}\rangle - \overline{\gamma^\uparrow}\sum_{s=T}^{t-1}\mathcal{L}'(\mathring{y}_s)\underbrace{\frac{\mathrm{Tr}\,\langle dh_s^{\uparrow(3)}\mid dh_t^{\uparrow(3)}\rangle}{k}}_{=\overline{\sigma}^2+\sigma_{\Delta,A}{}^2}|h_s^{\uparrow(2)}\rangle,$$

$$|dh_t^{\uparrow(1)}\rangle = |C_0^{\uparrow\top} dh_t^{\uparrow(2)}\rangle^\wedge + |C_0^{\uparrow\top} dh_t^{\uparrow(2)}\rangle^\cdot - \overline{\gamma}\sum_{s=0}^{T-2}\mathcal{L}'(\mathring{y}_s)\frac{\mathrm{Tr}\,\langle dh_s^{\uparrow(2)}\mid dh_t^{\uparrow(2)}\rangle}{k}\underbrace{|h_s^{\uparrow(1)}\rangle}_{=x|D_0^\uparrow\rangle}$$

$$+ |\Delta_C^\top dh_t^{\uparrow(2)}\rangle^\wedge + |\Delta_C^\top dh_t^{\uparrow(2)}\rangle^\cdot - \overline{\gamma^\uparrow}\sum_{s=T}^{t-1}\mathcal{L}'(\mathring{y}_s)\frac{\mathrm{Tr}\,\langle dh_s^{\uparrow(2)}\mid dh_t^{\uparrow(2)}\rangle}{k}\underbrace{|h_s^{\uparrow(1)}\rangle}_{=x(|D_0^\uparrow\rangle+|\Delta_D\rangle)}.$$

---

[7]As before, $|dh_t^{\uparrow(j)}\rangle$ tracks the distribution of $N\,dh_t^{\uparrow(j)}/\mathcal{L}'(y_t)$ for $j = 1, 2, 3$.

By substitution, $|h_t^{\uparrow(2)}\rangle$ satisfies the recursion

$$|h_t^{\uparrow(2)}\rangle = |C_0^\uparrow h_T^{\uparrow(1)}\widehat{\rangle} - \overline{\gamma}x^2\overline{\sigma}^2 \sum_{s=0}^{T-2} \mathcal{L}'(\mathring{y}_s)\left(|B_0^{\uparrow\top} dh_0^{\uparrow(3)}\widehat{\rangle} - \overline{\gamma}\,\overline{\sigma}^2 \sum_{\ell=0}^{s-1} \mathcal{L}'(\mathring{y}_\ell)\,|h_\ell^{\uparrow(2)}\rangle\right) + |\Delta_C h_T^{\uparrow(1)}\widehat{\rangle}$$

$$- \overline{\gamma^\uparrow}x^2(\overline{\sigma}^2 + \overline{\sigma_{\Delta,D}}^2) \sum_{s=T}^{t-1} \mathcal{L}'(\mathring{y}_s)\left(|B_0^{\uparrow\top} dh_T^{\uparrow(3)}\widehat{\rangle} - \overline{\gamma}\,\overline{\sigma}^2 \sum_{\ell=0}^{T-2} \mathcal{L}'(\mathring{y}_\ell)\,|h_\ell^{\uparrow(2)}\rangle + |\Delta_B^\top dh_T^{\uparrow(3)}\widehat{\rangle}\right.$$

$$\left. - \overline{\gamma^\uparrow}(\overline{\sigma}^2 + \overline{\sigma_{\Delta,A}}^2) \sum_{\ell=T}^{s-1} \mathcal{L}'(\mathring{y}_\ell)\,|h_\ell^{\uparrow(2)}\rangle\right).$$

Recall from (26) that for $t \leq T - 2$ we have

$$|h_t^{(2)}\rangle = M_t|C_0 h_0^{(1)}\widehat{\rangle} + N_t|B_0^\top dh_0^{(3)}\widehat{\rangle},$$

which translates to

$$|h_t^{\uparrow(2)}\rangle = M_t|C_0^\uparrow h_0^{\uparrow(1)}\widehat{\rangle} + N_t|B_0^{\uparrow\top} dh_0^{\uparrow(3)}\widehat{\rangle}.$$

Therefore, for $t \geq T$, we obtain the decomposition

$$|h_t^{\uparrow(2)}\rangle = M_t|C_0^\uparrow h_0^{\uparrow(1)}\widehat{\rangle} + M_t'|C_0^\uparrow h_T^{\uparrow(1)}\widehat{\rangle} + M_t'|\Delta_C h_T^{\uparrow(1)}\widehat{\rangle}$$

$$+ N_t|B_0^{\uparrow\top} dh_0^{\uparrow(3)}\widehat{\rangle} + N_t'|B_0^{\uparrow\top} dh_T^{\uparrow(3)}\widehat{\rangle} + N_t'|\Delta_B^\top dh_T^{\uparrow(3)}\widehat{\rangle}.$$

The coefficients satisfy

$$M_t = \overline{\gamma}^2 x^2 \overline{\sigma}^4 \sum_{s=1}^{T-2} \mathcal{L}'(\mathring{y}_s) \sum_{\ell=0}^{s-1} \mathcal{L}'(\mathring{y}_\ell) M_\ell$$

$$+ \overline{\gamma^\uparrow}x^2(\overline{\sigma}^2 + \overline{\sigma_{\Delta,D}}^2) \sum_{s=T}^{t-1} \mathcal{L}'(\mathring{y}_s)\left(\overline{\gamma}\,\overline{\sigma}^2 \sum_{\ell=0}^{T-2} \mathcal{L}'(\mathring{y}_\ell) M_\ell + \overline{\gamma^\uparrow}(\overline{\sigma}^2 + \overline{\sigma_{\Delta,A}}^2) \sum_{\ell=T}^{s-1} \mathcal{L}'(\mathring{y}_\ell) M_\ell\right),$$

$$M_t' = 1 + \overline{\gamma^\uparrow}^2 x^2(\overline{\sigma}^2 + \overline{\sigma_{\Delta,D}}^2)(\overline{\sigma}^2 + \overline{\sigma_{\Delta,A}}^2) \sum_{s=T}^{t-1} \mathcal{L}'(\mathring{y}_s) \sum_{\ell=T}^{s-1} \mathcal{L}'(\mathring{y}_\ell) M_\ell',$$

$$N_t = -\overline{\gamma}x^2\overline{\sigma}^2 \sum_{s=0}^{T-2} \mathcal{L}'(\mathring{y}_s) + \overline{\gamma}^2 x^2 \overline{\sigma}^4 \sum_{s=1}^{T-2} \mathcal{L}'(\mathring{y}_s) \sum_{\ell=0}^{s-1} \mathcal{L}'(\mathring{y}_\ell) N_\ell$$

$$+ \overline{\gamma^\uparrow}x^2(\overline{\sigma}^2 + \overline{\sigma_{\Delta,D}}^2) \sum_{s=T}^{t-1} \mathcal{L}'(\mathring{y}_s)\left(\overline{\gamma}\overline{\sigma}^2 \sum_{\ell=0}^{T-2} \mathcal{L}'(\mathring{y}_\ell) N_\ell + \overline{\gamma^\uparrow}(\overline{\sigma}^2 + \overline{\sigma_{\Delta,A}}^2) \sum_{\ell=T}^{s-1} \mathcal{L}'(\mathring{y}_\ell) N_\ell\right),$$

$$N_t' = -\overline{\gamma^\uparrow}x^2(\overline{\sigma}^2 + \overline{\sigma_{\Delta,D}}^2) \sum_{s=T}^{t-1} \mathcal{L}'(\mathring{y}_s)\left(1 - \overline{\gamma^\uparrow}(\overline{\sigma}^2 + \overline{\sigma_{\Delta,A}}^2) \sum_{\ell=T}^{s-1} \mathcal{L}'(\mathring{y}_\ell) N_\ell'\right).$$

$$(29)$$

Using the definition of $|\bullet\dot{\rangle}$ in (23), and since $kB_0^\uparrow/\overline{\sigma} \in \mathcal{W}_2$ and $\sqrt{k}\,\Delta_B/\overline{\sigma_{\Delta,B}} \in \mathcal{W}_1$, we have

$$
|B_0^\uparrow h_t^{\uparrow(2)}\dot{\rangle} = k^{-1}\overline{\sigma}k\;\mathbb{E}\left[\frac{\partial\,|h_t^{\uparrow(2)}\rangle}{\partial|k\overline{\sigma}^{-1}B_0^{\uparrow\top}dh_0^{\uparrow(3)}\dot{\rangle}}\right]\sum_j |dh_0^{\uparrow(3)}\rangle_j\,\mathbf{1}_k
$$

$$
+\,k^{-1}\overline{\sigma}k\;\mathbb{E}\left[\frac{\partial\,|h_t^{\uparrow(2)}\rangle}{\partial|k\overline{\sigma}^{-1}B_0^{\uparrow\top}dh_T^{\uparrow(3)}\dot{\rangle}}\right]\sum_j |dh_T^{\uparrow(3)}\rangle_j\,\mathbf{1}_k
$$

$$
=\overline{\sigma}\left(k^{-1}\overline{\sigma}N_t\sum_j |dh_0^{\uparrow(3)}\rangle_j\,\mathbf{1}_k + k^{-1}\overline{\sigma}N_t'\sum_j |dh_T^{\uparrow(3)}\rangle_j\,\mathbf{1}_k\right)
$$

$$
=\overline{\sigma}^2 N_t\,|A_0^\uparrow\rangle + \overline{\sigma}^2 N_t'\left(|A_0^\uparrow\rangle + \frac{\sum_j |\Delta_A\rangle_j}{k}\mathbf{1}_k\right),
$$

$$
|\Delta_B h_t^{\uparrow(2)}\dot{\rangle} = \overline{\sigma_{\Delta,B}}k^{-1/2}k\;\mathbb{E}\left[\frac{\partial|h_t^\uparrow\rangle^{(2)}}{\partial|k^{1/2}\overline{\sigma_{\Delta,B}}^{-1}\Delta_B^\top dh_T^{\uparrow(3)}\rangle}\right]|dh_T^{\uparrow(3)}\rangle
$$

$$
=\overline{\sigma_{\Delta,B}}k^{-1/2}kN_t'k^{-1/2}\overline{\sigma_{\Delta,B}}\left(|A_0^\uparrow\rangle + |\Delta_A\rangle\right)
$$

$$
=\overline{\sigma_{\Delta,B}}^2 N_t'\left(|A_0^\uparrow\rangle + |\Delta_A\rangle\right).
$$

Moreover, for $s \le T-2$ and $t \ge T$, we have

$$
\langle h_s^{\uparrow(2)} \mid h_t^{\uparrow(2)}\rangle
$$

$$
= M_s M_t\overline{\sigma}^2 k^{-2}\left(\mathbf{1}_k^\top \langle h_0^{\uparrow(1)} \mid h_0^{\uparrow(1)}\rangle\,\mathbf{1}_k\right)\mathbf{1}_k\mathbf{1}_k^\top + M_s M_t'\overline{\sigma}^2 k^{-2}\left(\mathbf{1}_k^\top \langle h_0^{\uparrow(1)} \mid h_T^{\uparrow(1)}\rangle\,\mathbf{1}_k\right)\mathbf{1}_k\mathbf{1}_k^\top
$$

$$
+\,N_s N_t\overline{\sigma}^2 k^{-2}\left(\mathbf{1}_k^\top \langle dh_0^{\uparrow(3)} \mid dh_0^{\uparrow(3)}\rangle\,\mathbf{1}_k\right)\mathbf{1}_k\mathbf{1}_k^\top + N_s N_t'\overline{\sigma}^2 k^{-2}\left(\mathbf{1}_k^\top \langle dh_0^{\uparrow(3)} \mid dh_T^{\uparrow(3)}\rangle\,\mathbf{1}_k\right)\mathbf{1}_k\mathbf{1}_k^\top
$$

$$
=\left(M_s(M_t + M_t')\overline{\sigma}^4 x^2 + N_s(N_t + N_t')\overline{\sigma}^4\right)\mathbf{1}_k\mathbf{1}_k^\top.
$$

Hence,

$$
\frac{\mathrm{Tr}\,\langle h_s^{\uparrow(2)} \mid h_t^{\uparrow(2)}\rangle}{k} = M_s(M_t + M_t')\overline{\sigma}^4 x^2 + N_s(N_t + N_t')\overline{\sigma}^4 =: \text{①}.
$$

Similarly, for $s,t \ge T$, we have

$$
\langle h_s^{\uparrow(2)} \mid h_t^{\uparrow(2)}\rangle
$$

$$
= M_s M_t\overline{\sigma}^2 k^{-2}\left(\mathbf{1}_k^\top \langle h_0^{\uparrow(1)} \mid h_0^{\uparrow(1)}\rangle\,\mathbf{1}_k\right)\mathbf{1}_k\mathbf{1}_k^\top + N_s N_t\overline{\sigma}^2 k^{-2}\left(\mathbf{1}_k^\top \langle dh_0^{\uparrow(3)} \mid dh_0^{\uparrow(3)}\rangle\,\mathbf{1}_k\right)\mathbf{1}_k\mathbf{1}_k^\top
$$

$$
+\,M_s' M_t'\overline{\sigma}^2 k^{-2}\left(\mathbf{1}_k^\top \langle h_T^{\uparrow(1)} \mid h_T^{\uparrow(1)}\rangle\,\mathbf{1}_k\right)\mathbf{1}_k\mathbf{1}_k^\top + N_s' N_t'\overline{\sigma}^2 k^{-2}\left(\mathbf{1}_k^\top \langle dh_T^{\uparrow(3)} \mid dh_T^{\uparrow(3)}\rangle\,\mathbf{1}_k\right)\mathbf{1}_k\mathbf{1}_k^\top
$$

$$
+\,M_s' M_t'\overline{\sigma_{\Delta,C}}^2 k^{-1}\,\mathrm{Tr}\,\langle h_T^{\uparrow(1)} \mid h_T^{\uparrow(1)}\rangle\,I_k + N_s' N_t'\overline{\sigma_{\Delta,B}}^2 k^{-1}\,\mathrm{Tr}\,\langle dh_T^{\uparrow(3)} \mid dh_T^{\uparrow(3)}\rangle\,I_k
$$

$$
+\,(M_s M_t' + M_s' M_t)\overline{\sigma}^2 k^{-2}\left(\mathbf{1}_k^\top \langle h_0^{\uparrow(1)} \mid h_T^{\uparrow(1)}\rangle\,\mathbf{1}_k\right)\mathbf{1}_k\mathbf{1}_k^\top
$$

$$
+\,(N_s N_t' + N_s' N_t)\overline{\sigma}^2 k^{-2}\left(\mathbf{1}_k^\top \langle dh_0^{\uparrow(3)} \mid dh_T^{\uparrow(3)}\rangle\,\mathbf{1}_k\right)\mathbf{1}_k\mathbf{1}_k^\top
$$

$$
=\Big((M_s M_t + M_s M_t' + M_s' M_t)\overline{\sigma}^4 x^2 + (N_s N_t + N_s N_t' + N_s' N_t)\overline{\sigma}^4
$$

$$
+M_s' M_t'\overline{\sigma}^2(\overline{\sigma}^2 + k^{-1}\overline{\sigma_{\Delta,D}}^2)x^2 + N_s' N_t'\overline{\sigma}^2(\overline{\sigma}^2 + k^{-1}\overline{\sigma_{\Delta,A}}^2)\Big)\mathbf{1}_k\mathbf{1}_k^\top
$$

$$
+\left(M_s' M_t'\overline{\sigma_{\Delta,C}}^2(\overline{\sigma}^2 + \overline{\sigma_{\Delta,D}}^2)x^2 + N_s' N_t'\overline{\sigma_{\Delta,B}}^2(\overline{\sigma}^2 + \overline{\sigma_{\Delta,A}}^2)\right)I_k.
$$

Consequently,

$$\frac{\operatorname{Tr}\langle h_s^{(2)} \mid h_t^{(2)}\rangle}{k} = x^2\overline{\sigma}^4(M_s + M_s')(M_t + M_t') + \overline{\sigma}^4(N_s + N_s')(N_t + N_t')$$
$$+ M_s'M_t'\left(k^{-1}\overline{\sigma}^2\overline{\sigma_{\Delta,D}}^2 + \overline{\sigma}^2\overline{\sigma_{\Delta,C}}^2 + \overline{\sigma_{\Delta,C}}^2\overline{\sigma_{\Delta,D}}^2\right)x^2$$
$$+ N_s'N_t'\left(k^{-1}\overline{\sigma}^2\overline{\sigma_{\Delta,A}}^2 + \overline{\sigma}^2\overline{\sigma_{\Delta,B}}^2 + \overline{\sigma_{\Delta,A}}^2\overline{\sigma_{\Delta,B}}^2\right)$$
$$=: \text{②'}.$$

Finally, by independence of various hat-kets, we obtain the post-upscaling readout

$$
\begin{aligned}
\mathring{y}_t &= \frac{\operatorname{Tr}\langle A_0^{\uparrow} \mid h_t^{\uparrow(3)}\rangle + \operatorname{Tr}\langle \Delta_A \mid h_t^{\uparrow(3)}\rangle}{k} \\
&= k^{-1}\operatorname{Tr}\left(\left(-\overline{\gamma}\sum_{s=0}^{T-2}\mathcal{L}'(\mathring{y}_s)\text{①} - \overline{\gamma^{\uparrow}}\sum_{s=T}^{t-1}\mathcal{L}'(\mathring{y}_s)\text{②'} + \overline{\sigma}^2(N_t + N_t') + \overline{\sigma_{\Delta,B}}^2 N_t'\right)\langle A_0^{\uparrow} \mid A_0^{\uparrow}\rangle\right) \\
&\quad + k^{-1}\operatorname{Tr}\left(\left(-\overline{\gamma^{\uparrow}}\sum_{s=T}^{t-1}\mathcal{L}'(\mathring{y}_s)\text{②'} + \overline{\sigma_{\Delta,B}}^2 N_t'\right)\langle \Delta_A \mid \Delta_A\rangle + \overline{\sigma}^2 N_t'\langle k^{-1}\sum_j(\Delta_A)_j\mathbf{1}_k \mid \Delta_A\rangle\right) \\
&= \overline{\sigma}^2\left(-\overline{\gamma}\sum_{s=0}^{T-2}\mathcal{L}'(\mathring{y}_s)\text{①} - \overline{\gamma^{\uparrow}}\sum_{s=T}^{t-1}\mathcal{L}'(\mathring{y}_s)\text{②'} + \overline{\sigma}^2(N_t + N_t') + \overline{\sigma_{\Delta,B}}^2 N_t'\right) \\
&\quad + \overline{\sigma_{\Delta,A}}^2\left(-\overline{\gamma^{\uparrow}}\sum_{s=T}^{t-1}\mathcal{L}'(\mathring{y}_s)\text{②'} + N_t'(k^{-1}\overline{\sigma}^2 + \overline{\sigma_{\Delta,B}}^2)\right) \\
&= \overline{\sigma}^4(N_t + N_t') + (\overline{\sigma}^2\overline{\sigma_{\Delta,B}}^2 + k^{-1}\overline{\sigma}^2\overline{\sigma_{\Delta,A}}^2 + \overline{\sigma_{\Delta,A}}^2\overline{\sigma_{\Delta,B}}^2)N_t' \\
&\quad - \overline{\sigma}^2\overline{\gamma}\sum_{s=0}^{T-2}\mathcal{L}'(\mathring{y}_s)\text{①} - (\overline{\sigma}^2 + \overline{\sigma_{\Delta,A}}^2)\overline{\gamma^{\uparrow}}\sum_{s=T}^{t-1}\mathcal{L}'(\mathring{y}_s)\text{②'}.
\end{aligned}
\tag{30}
$$

**Conclusion.** Adding one more hidden layer immediately complicates the computation of the infinite-width dynamics, but the system remains analyzable. Since in the previous section we found for a simpler model that, even when adding noise while upscaling, it is possible to exactly maintain the infinite-width limit of training dynamics, it is natural to ask whether that remains possible in this more complicated architecture.

Comparing the coefficient recursions before and after upscaling, (27) and (29), we observe that if

$$\overline{\gamma^{\uparrow}}^2\left(\overline{\sigma}^2 + \overline{\sigma_{\Delta,D}}^2\right)\left(\overline{\sigma}^2 + \overline{\sigma_{\Delta,A}}^2\right) = \overline{\gamma}^2\,\overline{\sigma}^4,$$

an analogous relation to that from the previous section, then the recursions for $(M_t + M_t')$ and $(N_t + N_t')$ coincide with their pre-upscaling counterparts (in particular, this holds when $\overline{\sigma_{\Delta,A}} = \overline{\sigma_{\Delta,D}} = 0$ and $\overline{\gamma^{\uparrow}} = \overline{\gamma}$). Further, comparing the final recursions (28) and (30), full agreement of infinite-width limits requires also $\overline{\sigma_{\Delta,B}} = \overline{\sigma_{\Delta,C}} = 0$, in which case both of the terms labeled ① and ② above match ①. In this (degenerate) case where no noise is added, the training dynamics after upscaling are identical to those before upscaling, which is expected given the equivalence we showed in Section 2. Otherwise, whenever we add noise during upscaling, the infinite-width training dynamics are expected to be altered after upscaling, and no choice of hyperparameters can exactly preserve the infinite-width limit.

One may view this as an illustration that tuning hyperparameters for upscaled training is a non-trivial task distinct from tuning for ordinary training. Indeed, on the one hand and as we have discussed, we must use non-zero noise in upscaling in order for training after upscaling to yield models that are actually utilizing their increased width and learning a larger class of functions than those parametrized by narrow models. On the other hand, we see from the above calculations that, even in quite simple architectures, once we use non-zero noise, we cannot hope for upscaled training to share the infinite-width limit of its training dynamics with ordinary training. Thus, in particular, it seems that in order to tune hyperparameters for upscaled training we must, as we do in our proposed method, actually simulate upscaling itself on smaller models, rather than merely taking hyperparameters tuned on non-upscaled training and systematically modifying them in some way (unless one were to explicitly describe the limiting behavior of upscaled training in terms of that of non-upscaled training, which, even with the NE⊗OR⊤ program tools, appears to be a prohibitively complicated mathematical task).

# F. Experiments

## F.1. MLPs

**Dataset.** We use the Forest Cover Type dataset (Blackard & Dean, 1999) from the UCI Machine Learning Repository, a tabular dataset for multiclass classification into 7 forest cover types based on attributes such as elevation, aspect, slope, hillshade, soil type, and additional environmental variables. The dataset comprises 581,012 samples with 54 features, including both continuous and binary variables. We apply standard preprocessing by performing a stratified 80–20 train–test split and normalizing the continuous features, while leaving the binary features unchanged. We select this dataset because MLPs achieve strong performance on tabular data of this type, and its relatively large sample size yields a sufficiently challenging task in which increasing MLP width and thus model capacity improves predictive accuracy. Therefore, it is well suited for our exploration of model upscaling.

**Model.** We employ a standard MLP with bias terms and ReLU activations, comprising 4 layers whose hidden width is shared and set to $n$. Specifically, this is MLP defined in (1) with $L = 4, n_1 = n_2 = n_3 = n$, and $\phi = \text{ReLU}$. The $\mu P$ package is used to configure weight initialization and learning-rate scaling to ensure width-consistent training.

**Optimizer and training.** We experiment with both SGD and AdamW, and we use the $\mu P$ implementations of these optimizers to obtain the appropriate scaling with respect to network width. For SGD, we apply weight decay with a base coefficient of $10^{-4}$ (with $\mu P$ scaling) to stabilize training. For AdamW, we set $\beta = (0.9, 0.999)$, $\epsilon = 10^{-8}$, and weight decay $10^{-4}$, each applied with the corresponding $\mu P$ scaling. We tune the learning-rate base constant and the magnitude of added noise. We use a batch size of 2000 and train for 500 epochs.

**Experiment procedure.**

(1) Under $\mu P$, we begin with Sweep 1 over learning rates $\overline{\gamma}$ at width $n_0 = 100$, selecting the best learning rate based on the training loss after 500 epochs.

(2) We then train a base model of width $n = 500$ from scratch for 500 epochs using the best learning rate $\overline{\gamma}$ identified in Sweep 1. We also train a wide model of width $kn = 2000$ from scratch using the same hyperparameters, to serve as a baseline for comparison with the upscaled model.

(3) Next, setting width multiplier $k = 4$, we select the width-$n_0$ checkpoint achieving the lowest training loss in Sweep 1 and perform Sweep 2 for upscaling: we construct an upscaled model of width $kn_0 = 400$, vary the noise std base constant $\overline{\sigma}$ and learning rate $\overline{\gamma^\uparrow}$, and train for 500 epochs.

(4) Finally, we apply the best noise level and learning rate $\overline{\gamma^\uparrow}$ found in Sweep 2 to upscale the base model to width $kn = 2000$ and train it for 500 epochs.

**Results.** Figure 4 shows training and validation curves for an MLP trained with SGD and weight decay, which are omitted from the main paper. In this setting, Sweep 1 selects a learning-rate base constant of $\overline{\gamma} = 0.4$. Sweep 2 selects a learning-rate base constant of $\overline{\gamma} = 0.1$ and a noise std base constant of $\overline{\sigma} = 5$.

Figure 5 visualizes the hyperparameter sensitivity of the upscaled model in Sweep 2. The heatmap shows the minimum training loss achieved over a grid of learning rates and noise levels $\overline{\sigma_\Delta}$, with the optimal configuration marked by a red star. The right panel shows training loss curves at the best learning rate, with each curve corresponding to a different noise level indicated by the colorbar. Performance first improves as noise increases—starting from a worse initialization but converging to a better final value, as additional model capacity is exploited—and then degrades gracefully at higher noise levels. This indicates the existence of an optimal noise magnitude whose selection meaningfully impacts performance, whereas prior work does not address this choice.

Analogous results for AdamW are shown in Figure 6, with a subset reported in the main paper. Here we show the entire training curve, with a zoomed-in view of the training loss to better highlight the differences between the upscaled and from-scratch curves. In this setting, Sweep 1 selects a learning-rate base constant of $\overline{\gamma} = 0.1$. Sweep 2 selects a learning-rate base constant of $\overline{\gamma} = 0.013$ and a noise std base constant of $\overline{\sigma} = 4$. On a side note, AdamW is more stable than SGD and generally attains higher validation accuracy on this task. Meanwhile the validation loss of the upscaled model exhibits overfitting, suggesting that the model becomes increasingly overconfident over the course of training and may be

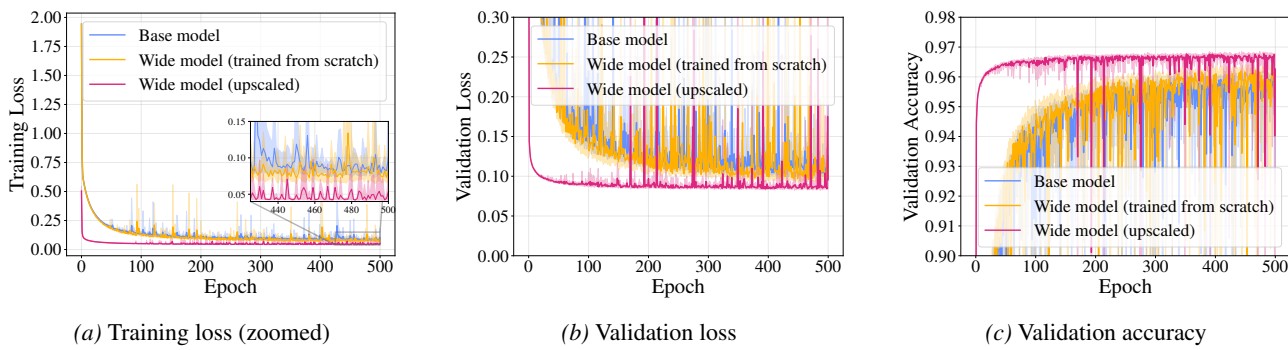

*(a)* Training loss (zoomed)          *(b)* Validation loss          *(c)* Validation accuracy

*Figure 4.* Training and validation curves comparing upscaling to training from scratch for an MLP trained with SGD and weight decay. The wide models have width $kn = 2000$; the base model (width $n = 500$) is included for reference. Curves show the mean across five random runs, with ranges spanning the minimum to the maximum across runs. The training loss panel is zoomed in on the y-axis to highlight differences between the two curves.

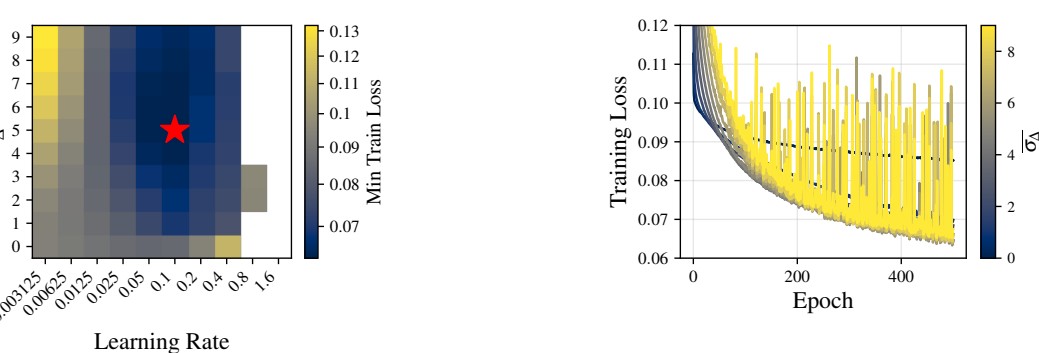

*(a)* Sweep heatmap over learning rate and noise $\overline{\sigma_\Delta}$          *(b)* Training loss at best learning rate across noise levels

*Figure 5.* Hyperparameter sweep for MLP with SGD. Left: minimum training loss over a grid of learning rates and noise magnitudes $\overline{\sigma_\Delta}$; the red star marks the optimum and empty squares indicate unstable runs that diverged. Right: training loss curves at the best learning rate for varying noise levels (colorbar).

miscalibrated. Importantly, our theory of model upscaling focuses on training dynamics, specifically the behavior of the training curves. Understanding of generalization under upscaling requires a separate analytical treatment.

Figure 7 shows the analogous sweep for AdamW. The same qualitative pattern is observed: an optimal noise level exists, with performance degrading at both lower and higher values.

### F.2. ResNet

**Dataset.**    We evaluate on CIFAR-100 (Krizhevsky & Hinton, 2009), an image dataset of $32 \times 32$ color images with 100 classes. For training, we apply standard data augmentation comprising a 4-pixel padding followed by a random crop to $32 \times 32$ and a random horizontal flip, after which we normalize using per-channel means and standard deviations computed on the training set.

**Model.**    We adopt the standard 18-layer ResNet from He et al. (2016) and vary its width, i.e., the number of feature channels per stage. In the standard ResNet-18, the convolutional stem outputs 64 channels, and the four residual stages use 64, 128, 256, and 512 channels, respectively. We consider width-multiplier variants: for example, the $2\times$ model has all widths multiplying by 2, i.e. it uses 128, 256, 512, and 1024 channels. We use the $\mu$P library to configure weight initialization and learning-rate scaling to ensure width-consistent training dynamics. In particular, we configure the implementation so that the standard ($1\times$) model exhibits identical behavior under $\mu$P and under the standard parametrization. See, e.g., Remark C.12.

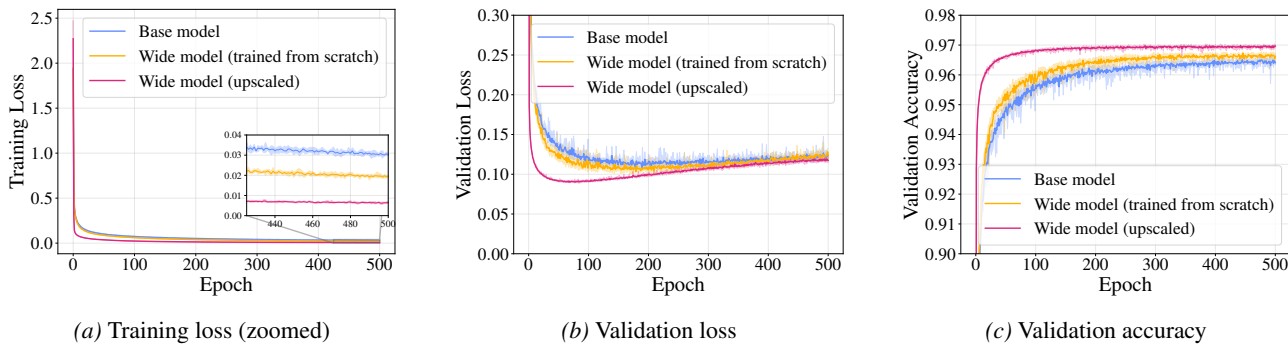

*(a)* Training loss (zoomed)    *(b)* Validation loss    *(c)* Validation accuracy

*Figure 6.* Training and validation curves comparing upscaling to training from scratch for an MLP trained with AdamW. The wide models have width $kn = 2000$; the base model (width $n = 500$) is included for reference. Curves show the mean across five random runs, with ranges spanning the minimum to the maximum across runs. The training loss panel is zoomed in on the y-axis to highlight differences between the two curves.

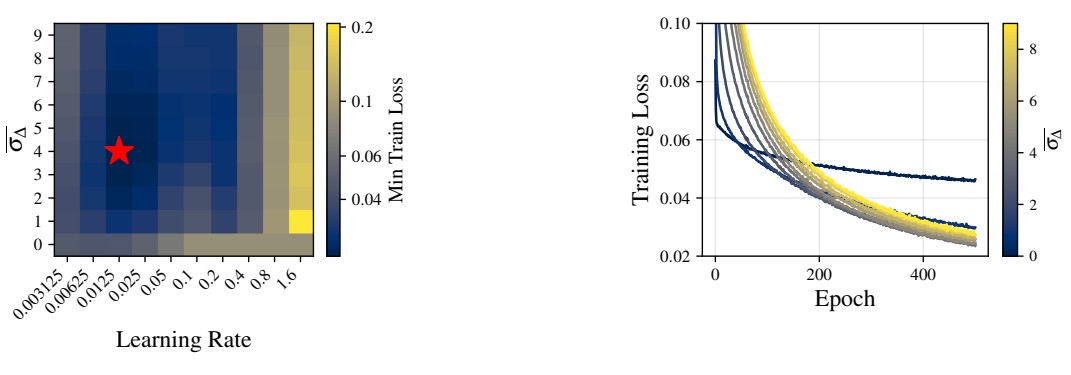

*(a)* Sweep heatmap over learning rate and noise $\overline{\sigma_\Delta}$    *(b)* Training loss at best learning rate across noise levels

*Figure 7.* Hyperparameter sweep for MLP with AdamW. Left: minimum training loss over a grid of learning rates and noise magnitudes $\overline{\sigma_\Delta}$; the red star marks the optimum. Right: training loss curves at the best learning rate for varying noise levels (colorbar).

**Optimizer and training.** Following He et al. (2016), we train with SGD using momentum 0.9 and weight decay $10^{-4}$. We do not use Adam because it seems to have worse generalization behavior as compared to SGD. We tune the learning rate base constant and the magnitude of injected noise, when applicable. We use a batch size of 128 and train for 100 epochs.

**Experiment procedure.** Because ResNet-18 has heterogeneous hidden dimensions across stages (64, 128, 256, 512 channels), adding the same amount of noise to every layer leads to suboptimal performance. We therefore use the additive rescaled noise scheme described in Appendix D, which normalizes the injected noise relative to the spectral norm of the widened weights at each layer.

(1) Under $\mu$P, we begin with Sweep 1 over learning rates $\overline{\gamma}$ using the $0.5\times$ models, selecting the best learning rate based on the training loss after 100 epochs.

(2) We then train a $2\times$ base model from scratch for 100 epochs using the best learning rate $\overline{\gamma}$ identified in Sweep 1. We also train a wide $4\times$ model from scratch using the same hyperparameters, to serve as a baseline for comparison with the upscaled model.

(3) Next, setting width multiplier $k = 2$, we select the $0.5\times$ model checkpoint achieving the lowest training loss in Sweep 1 and perform Sweep 2 for upscaling: we construct an upscaled $1\times$ model, vary the relative noise level $t$ and learning rate $\overline{\gamma^\uparrow}$, and train for 100 epochs.

(4) Finally, using the best $t$ found in Sweep 2, we compute the effective per-weight noise base constant $\overline{\sigma_{\Delta,W}}$ for each parameter (as described in Appendix D) and transfer these values to upscale the $2\times$ base model to $4\times$, training for 100

epochs with learning rate $\overline{\gamma^\uparrow}$ from Sweep 2.

**Experiment results.** Figure 8 shows training and validation curves for ResNets trained with SGD (using weight decay and momentum), with a subset of results reported in the main paper. Here we show the entire training curve, with a zoomed-in view of the training loss to better highlight the differences between the upscaled and from-scratch curves. In this setting, Sweep 1 selects a learning-rate base constant of $\overline{\gamma} = 0.01$. Sweep 2 selects a learning-rate base constant of $\overline{\gamma^\uparrow} = 0.003$ and a relative noise level of $t = 0.4$. For training loss, the upscaled model converges faster and reaches lower terminal loss; however, in both validation loss and validation accuracy, the wide model trained from scratch outperforms the upscaled model.

Our theory addresses training dynamics and does not make predictions about generalization, so the weaker test performance does not contradict our theoretical framework. The generalization gap observed here is a well-known property of overparameterized CNNs trained on limited data: CIFAR-100 has only 500 images per class, placing models deep in the overparameterized regime where the difference in test performance is dominated by the generalization gap rather than by any difference in training dynamics. Importantly, this phenomenon is likely not specific to our upscaling method but inherent to model upscaling in such settings: even with hyperparameters tuned directly at the target scale (i.e., following the Net2Net baseline), it is not certain that upscaling would yield better generalization in this regime, since the upscaled and from-scratch models differ substantially in both their training trajectories and the implicit regularization they induce. Practitioners should therefore exercise caution when applying upscaling in settings where validation and training curves diverge substantially. Understanding generalization under upscaling remains an interesting open question orthogonal to the hyperparameter transfer guarantees our paper provides.

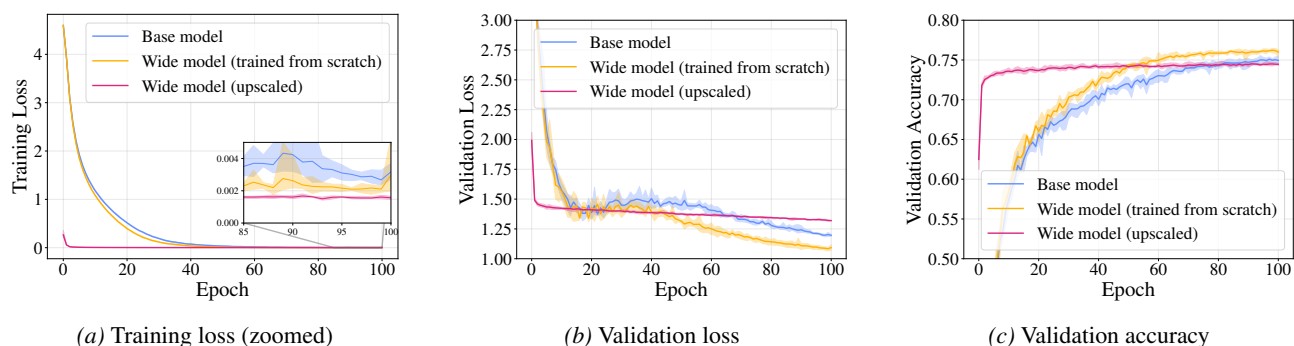

*(a)* Training loss (zoomed)  *(b)* Validation loss  *(c)* Validation accuracy

*Figure 8.* Training and validation curves comparing upscaling to training from scratch for ResNet trained with SGD. The wide models are $4\times$ ResNets; the base model ($2\times$) is included for reference. Curves show the mean across five random runs, with ranges spanning the minimum to the maximum across runs. The training loss panel is zoomed in on the y-axis to highlight differences between the two curves.

Figure 9 shows the analogous sweep for ResNet with SGD, using relative noise level $t$. The same pattern holds: an optimal noise level exists, and the choice meaningfully impacts performance.

### F.3. GPT-2

**Dataset.** We evaluate on the `CC-MAIN-2013-20` subset[8] of the FineWeb-Edu dataset (Penedo et al., 2024). We leverage the official tokenizer of GPT-2, which has a vocabulary size of 50,257. We construct a training split of 11.8B tokens and a validation split of 5.5M tokens.

**Model.** We adopt the standard GPT-2 architecture (Radford et al., 2019) and vary its hidden dimension. Specifically, we use the GPT-2 small configuration with 12 layers and 12 attention heads. We define the $20\times$ model as GPT-2 small with 320 dimensions per head, and create the $10\times$ and $1\times$ models by scaling down the head dimension to 160 and 16, respectively. All models share the same number of layers and attention heads.

**Optimizer and training.** We utilize the AdamW optimizer with $(\beta_1, \beta_2) = (0.90, 0.95)$, using a weight decay of 0.1 and gradient clipping at 1.0. We set an effective batch size of 0.5M tokens per step and train the model for 10,000 steps,

---

[8]https://huggingface.co/datasets/HuggingFaceFW/fineweb-edu/viewer/CC-MAIN-2013-20

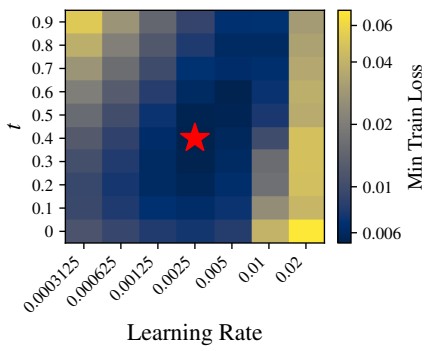

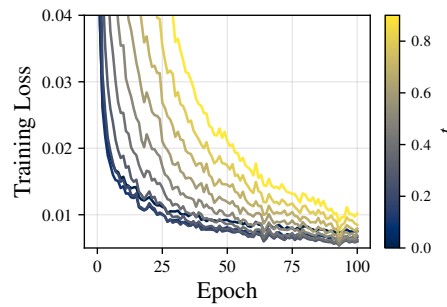

*(a)* Sweep heatmap over learning rate and noise level $t$

*(b)* Training loss at best learning rate across noise levels

*Figure 9.* Hyperparameter sweep for ResNet with SGD. Left: minimum training loss over a grid of learning rates and relative noise levels $t$; the red star marks the optimum. Right: training loss curves at the best learning rate for varying noise levels (colorbar).

totalling 5B tokens. A constant learning rate schedule is employed. For efficiency, we adapt mixed-precision training (bf16 with TF32-enabled `matmul`). All experiments are conducted on 2 NVIDIA H200 GPUs (141 GiB each) using PyTorch Distributed Data Parallel (DDP).

**Experiment procedure.**

(1) Under $\mu$P, we begin with Sweep 1 over learning rates $\overline{\gamma}$ using the $1\times$ models, selecting the best learning rate based on end-of-training validation negative log likelihood (NLL).

(2) We then train a $10\times$ base model from scratch for $10,000$ steps using the best learning rate $\overline{\gamma}$ identified in Sweep 1. We also train a wide $20\times$ model from scratch using the same hyperparameters, to serve as a baseline for comparison with the upscaled model.

(3) Next, setting width multiplier $k = 2$, we select the $1\times$ model checkpoint in Sweep 1 and perform Sweep 2 for upscaling: we construct an upscaled $2\times$ model, vary the noise std base constant $\overline{\sigma}$ and learning rate $\overline{\gamma^{\uparrow}}$, and train for $10,000$ steps.

(4) Finally, we apply the best noise level and learning rate $\overline{\gamma^{\uparrow}}$ found in Sweep 2 to upscale the $10\times$ base model to $20\times$ and train it for $10,000$ steps.

**Results.** Figure 2 (last column) in the main paper shows training and validation loss for the GPT-2 experiment described above. Figure 10 shows the entire training curve, with a zoomed-in view to better highlight the differences between the upscaled and from-scratch curves. In this setting, Sweep 1 selects a learning-rate base constant of $\overline{\gamma} = 0.0039$. Sweep 2 selects a learning-rate base constant of $\overline{\gamma^{\uparrow}} = 0.0020$ and a noise std base constant of $\overline{\sigma} = 0.005$. The upscaled model converges faster and achieves lower terminal training and validation loss than training from scratch.

### F.4. Evaluation Criteria

We evaluate our method along two axes: the cost of hyperparameter tuning (where we compare against Net2Net) and the total training cost of the upscaled model (where we compare against training from scratch). Below, we first describe how FLOPs are estimated for each architecture, then detail each comparison and the assumptions involved.

**FLOPs estimation.** We estimate training FLOPs (forward plus backward pass) for each model using the standard $6N$ approximation (Kaplan et al., 2020): the factor of six accounts for two multiply-accumulate operations in the forward pass and four in the backward pass (two for the input gradient and two for the weight gradient).

For the MLP, let $N$ denote the total number of weight-matrix parameters (excluding biases), $d_h$ the hidden dimension, and $L$ the number of layers. The cost per sample is $\mathcal{F}_{\text{MLP}} = 6 \cdot (d_{\text{in}} \cdot d_h + (L - 2) \cdot d_h^2 + d_h \cdot d_{\text{out}})$. For ResNet-18, we sum the multiply-accumulate operations (MACs) over all convolutional and linear layers; each convolutional layer contributes

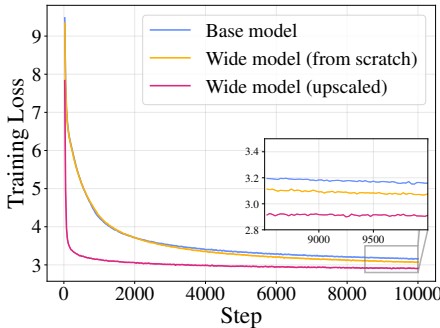

*Figure 10.* Training loss (zoomed) for GPT-2 trained with AdamW. The wide model has 320 dimensions per head ($20\times$); the base model (160 dimensions per head, $10\times$) is included for reference.

$C_{\text{in}} \times C_{\text{out}} \times k^2 \times H_{\text{out}} \times W_{\text{out}}$ MACs, giving $\mathcal{F}_{\text{ResNet}} = 6 \cdot \sum_\ell \text{MACs}_\ell$ per sample. For GPT-2, following Chowdhery et al. (2023), the FLOPs per token are $\mathcal{F}_{\text{GPT-2}} = 6N + 12LHQT$, where the first term covers all weight-matrix multiplications and the second term accounts for the quadratic attention computation: each query of dimension $Q$ is dotted with $T$ keys across $H$ heads and $L$ layers. Because attention operates on activations rather than stored parameters, its cost is not captured by the $6N$ term alone.

**Tuning cost comparison with Net2Net.** A key advantage of our method over Net2Net (Chen et al., 2016) is the ability to perform hyperparameter tuning at the small width $kn_0$ instead of the expensive target width $kn$. Table 4 reports the resulting speedup for each experiment: an $S\times$ speedup means one can run $S\times$ as many tuning runs under the same compute budget, as measured by the approximate FLOPs per run. Because transformer parameter counts scale as $\Theta(n^2)$ (due to width-squared dominance in the embedding and attention projection matrices), the speedup grows roughly as $(n/n_0)^2$; at practical ratios much larger than our experimental $n/n_0 = 10$, substantially greater savings are expected.

|  | **MLP** $(n/n_0 = 5)$ | **ResNet** $(n/n_0 = 4)$ | **GPT-2** $(n/n_0 = 10)$ |
|---|---|---|---|
| Net2Net (tune at target scale) | $1.0\times$ | $1.0\times$ | $1.0\times$ |
| $\mu$pscaling (ours, tune at small scale) | $23.6\times$ | $16.0\times$ | $48.2\times$ |

*Table 4.* Tuning-cost FLOPs speedup of hyperparameter transfer vs. tuning the full-scale target model directly, at each model's experiment compression ratio. Speedup $= \mathcal{F}(kn) / \mathcal{F}(kn_0)$.

**Accounting for tuning cost in the upscaled vs. from-scratch comparison.** The preceding comparison quantifies the tuning cost *relative to Net2Net*; we now address a separate question: should the tuning cost be charged to the upscaled model when comparing total training cost against training from scratch? We note that the relative tuning cost decreases as $n/n_0$ grows, because tuning is performed at width $kn_0$ while the target trains at width $kn$. When $n/n_0$ is sufficiently large, even fully charging the tuning cost to the upscaled model yields a total well below training from scratch.

We demonstrate this with our GPT-2 experiment ($n/n_0 = 10$), where the cost breakdown is as follows:

- Training from scratch: 37,929,809 TFLOPs.
- Upscaled training (to reach the from-scratch terminal loss): 6,539,099 TFLOPs.
- Hyperparameter tuning ($5 \times 5$ grid at width $kn_0 = 384$): 19,679,775 TFLOPs.
- Total (upscaled + tuning): 26,218,874 TFLOPs.

Even including the full hyperparameter tuning cost, the speedup is about $1.4\times$. We emphasize that the $5 \times 5$ grid search is a conservative upper bound on the tuning cost: in practice, more efficient strategies such as random search or fewer trials would reduce this further. At even larger scales ($n/n_0 \gg 10$), the tuning cost becomes negligible relative to the target training cost, making the savings even more pronounced.

**Accounting for the base-model training cost in the upscaled vs. from-scratch comparison.**   Beyond hyperparameter tuning, one may also ask whether the base-model training cost should be attributed to the upscaled model. In the main body, we evaluate in the setting where a trained base model is already available and its training cost is treated as sunk. This is justified because model upscaling can be readily applied to open-source checkpoints from repositories such as HuggingFace, in which case the base-model training cost is naturally a sunk cost. Accordingly, this evaluation criterion has been adopted in many prominent prior works on upscaling, including Net2Net (Chen et al., 2016), bert2BERT (Chen et al., 2022), LiGO (Wang et al., 2023), and Mango (Pan et al., 2023).

Here, we additionally consider the other extreme: fully including the base-model training cost, i.e., comparing the cost of (base-model training + upscaled-model training) against training the wide model from scratch. We note that this pessimistic accounting is only valid if one trains the small model *solely* for the purpose of upscaling and does so *exactly once*. In practice, the base model is often useful on its own (e.g., for serving smaller-scale inference) and can be reused for multiple rounds of upscaling, so the effective cost attributable to any single upscaling event is lower. The actual benefit therefore depends on the specific application scenario and likely lies between these two extremes.

Even under this pessimistic accounting, upscaling remains more compute-efficient in all four settings:

- **MLP (SGD):** Base-model training: 740 TFLOPs; upscaled model reaches from-scratch min loss at 598 TFLOPs; total cost = 1,338 TFLOPs. Wide model from scratch: 11,326 TFLOPs. Speedup: $\approx$8.5$\times$.
- **MLP (AdamW):** Base-model training: 740 TFLOPs; upscaled model reaches from-scratch min loss at 3,031 TFLOPs; total cost = 3,771 TFLOPs. Wide model from scratch: 11,326 TFLOPs. Speedup: $\approx$3.0$\times$.
- **ResNet:** Base-model training: 66,547 TFLOPs; upscaled model reaches from-scratch min train loss at 132,985 TFLOPs; total cost = 199,532 TFLOPs. Wide model from scratch: 265,970 TFLOPs. Speedup: $\approx$1.3$\times$.
- **GPT-2:** Base-model training: 10,615,488 TFLOPs; upscaled model reaches from-scratch min loss at 6,539,099 TFLOPs; total cost = 17,154,587 TFLOPs. Wide model from scratch: 37,929,809 TFLOPs. Speedup: $\approx$2.2$\times$.

### F.5. Verification of Hyperparameter Transfer

We experimentally validate that, under Meta-algorithm 1, hyperparameters transfer across model widths, consistent with the behavior reported in Yang et al. (2021) without upscaling.

**MLP.**   The following MLP results are omitted from the main paper. We use the same MLP architecture, dataset, and optimizer defaults as in Appendix F.1, with the sole modification of training for 100 epochs to expedite experimentation.

For SGD with weight decay, Figure 11(a,b) reports the experiment results. Panel (a) fixes the noise std base constant at $\overline{\sigma_\Delta} = 0.75$ (near-optimal) and varies the upscaled-model learning-rate base constant $\overline{\gamma^\uparrow}$. Panel (b) fixes the learning-rate base constant at $\overline{\gamma^\uparrow} = 0.1$ and varies the amount of added noise, controlled by the noise std base constant $\overline{\sigma_\Delta}$.

Analogous results for AdamW appear in Figure 11(c,d). Panel (c) fixes $\overline{\sigma_\Delta} = 1$ and varies $\overline{\gamma^\uparrow}$. Panel (d) fixes $\overline{\gamma^\uparrow} = 0.0005$ and varies $\overline{\sigma_\Delta}$.

In all cases, the optimal hyperparameters generally transfer across widths.

**GPT-2.**   We run hyperparameter transfer verification experiments using a smaller model with 8 layers and 8 attention heads. To further control the model size, we train a BPE tokenizer with an 8,192-word vocabulary on FineWeb-edu. As in F.3, we fix the number of heads and vary only the per-head dimension. We train base models with hidden sizes $n \in \{64, 128, 256, 512\}$ (head dimensions $\{8, 16, 32, 64\}$), then upscale each by $k = 2$ to $N \in \{128, 256, 512, 1024\}$ (head dimensions $\{16, 32, 64, 128\}$). Across these settings, we sweep injected noise levels and learning rates. All base models are trained under $\mu$P with the same hyperparameters.

**Discussion of the noise sensitivity profile.**   The noise sweep panels (Figure 3(b) and Figure 11(b,d)) reveal an asymmetric sensitivity profile around the optimal $\overline{\sigma_\Delta}$. On the right side, choosing $\overline{\sigma_\Delta}$ slightly above the optimum leads to rapid performance degradation, indicating that excessive noise overwhelms the signal inherited from the base model. On the left side, the curves are relatively flat below the optimum. This flatness reflects the following theoretical property: as established in Section 3, setting $\overline{\sigma_\Delta} = 0$ causes the upscaled model to evolve equivalently to the base model, as if no upscaling had occurred (dynamic equivalence). Consequently, values of $\overline{\sigma_\Delta}$ well below the optimum effectively reduce to continued training of the base model with an adjusted learning rate, which for these tasks already achieves a terminal loss not far

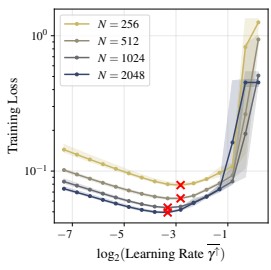 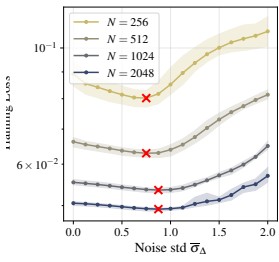 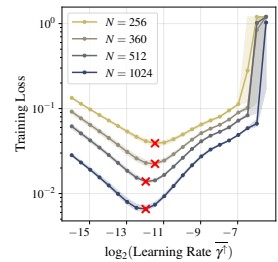 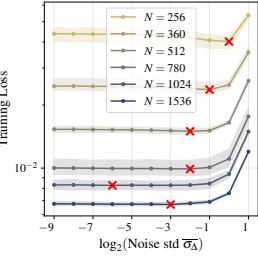

*(a)* MLP under SGD: Learning-rate sweep with fixed additive noise.

*(b)* MLP under SGD: Noise sweep with fixed learning rate.

*(c)* MLP under AdamW: Learning-rate sweep with fixed additive noise.

*(d)* MLP under AdamW: Noise sweep with fixed learning rate.

*Figure 11.* Hyperparameter transfer for the upscaled MLP model. Panels (a,b) show results under SGD, and panels (c,d) show results under AdamW. Curves report the mean across five runs, with min–max ranges across seeds. In (d), more widths are evaluated than in (c) because of the slightly noisy behavior at $N = 1024$.

from the properly upscaled model. The gap between this flat region and the optimum is precisely the benefit conferred by upscaling: breaking the symmetry sufficiently to escape the lower-dimensional parameter subspace and exploit the additional capacity of the wider model. Upon closer inspection, the left portion of each curve does exhibit a smooth decrease toward the optimum, confirming that the choice of $\overline{\sigma}_\Delta$ matters even within the stable region.

## G. Conclusions and Limitations

We present the first rigorous theoretical framework for hyperparameter transfer in the model upscaling setting. Our contributions are threefold. First, we extend the notion of function-preserving expansion (static equivalence) to *dynamic equivalence*—proving that, without noise injection, the upscaled model's entire training trajectory is equivalent to that of the base model—and establish both properties for general architectures and optimizers. Second, and most critically, we establish *hyperparameter transfer for upscaled systems*: by extending the Tensor Programs framework to accommodate upscaling, we show that the learning rate and injected noise magnitude can be tuned on a small, inexpensive proxy system and transferred directly to the target scale, substantially reducing the cost of hyperparameter tuning compared to prior upscaling methods such as Net2Net. Third, our infinite-width limit analysis rigorously characterizes the training dynamics of upscaled models, opening the door to future theoretical investigations connecting model equivalence, optimization, and scaling. Importantly, our framework applies universally across standard architectures and optimizers, in contrast to prior upscaling methods that are architecture-specific or purely empirical. Experiments on MLPs, ResNets, and GPT-2 consistently validate the theory's predictions: hyperparameters transfer robustly across widths, and upscaled models converge faster while achieving lower terminal training loss than models trained from scratch.

Several limitations merit discussion. First, our framework inherits the assumptions underlying $\mu$P: while $\mu$P is rigorously proven to yield optimal training dynamics in the infinite-width limit (Yang & Hu, 2021; Yang & Littwin, 2021), formal proofs of hyperparameter transfer exist only in simple settings (Hayou, 2026). Our experiments confirm that transfer is effective in practice, but it may not be uniformly robust across all tasks and architectures. Moreover, analogous to $\mu$Transfer (Yang et al., 2021), our framework transfers only *certain* hyperparameters—the learning rate and injected noise magnitude—and does not extend to others such as the growth factor $k$ or the base-model scale $n$, whose optimal values are likely task- and data-specific and may require complementary approaches such as scaling-law analyses (Du et al., 2024). Second, our scope is restricted to width upscaling; extending the framework to joint width-and-depth upscaling, as required by many practical deployments, is a promising direction for future work. Third, effective transfer empirically requires the tuning width $n_0$ to be sufficiently large (often exceeding 100 units) so that $\mu$P's asymptotic behavior is a good approximation, while the cost savings grow with the target-to-tuning ratio $n/n_0$. This makes the method most advantageous at large target widths—precisely the regime where hyperparameter tuning at scale is most expensive—but empirically validating these gains requires computational resources beyond our academic setting. Finally, our analysis addresses training dynamics rather than generalization, optimization landscape, or implicit bias. Performance may therefore degrade in settings prone to overfitting (as observed in the ResNet experiment on CIFAR-100), and understanding generalization under upscaling remains an open question.

