# OpenReview forum: "$\mu$pscaling small models: Principled warm starts and hyperparameter transfer"
_ICML.cc/2026/Conference — ICML 2026 regular_

### Official Review · Reviewer_8z1w · 2026-03-01

**Soundness:** 3
**Presentation:** 2
**Significance:** 2
**Originality:** 3
**Overall Recommendation:** 5
**Confidence:** 4

**Summary:**

Motivated by the high cost of large-scale training and the common need to train models at multiple sizes, this paper proposes a principled approach to warm-start large models using smaller ones. The authors extend $\mu$P-related theoretical results to analyze training after upscaling and provide empirical evaluations to support their method.

**Compliance With Llm Reviewing Policy:**

Affirmed.

**Final Justification:**

The paper studies an important and timing problem: enabling efficient training and HP transfer across model sizes. The work is grounded in a principled theoretical framework based on Tensor Programs and $\mu P$, and I find the analysis technically sound and well motivated. Providing a rigorous foundation for scaling and transfer is a meaningful contribution beyond heuristic approaches.

Overall, while clarity can be further improved, the strengths in soundness, motivation, and theoretical contribution outweigh the weaknesses. I support Accept.

**Key Questions For Authors:**

- How were the HP-transfer experiments conducted? For each width and LR, did you select the best checkpoint using validation loss, or compare at fixed training steps? Small or large LRs may underfit or diverge, so the selection protocol matters.
- After upscaling, $W_{\text{large}} = W_{\text{lifted}} + \Delta\in \mathbb{R}^{N\times N}$. So the large model is effectively initialized from a Gaussian with mean $W_{\text{lifted}}$ and variance $1/N$, while scratch training has mean zero. Is this the intended interpretation, and does it justify applying $\mu$P training-dynamics analysis after upscaling?
- For different target widths in the HP-transfer experiments, did you always start from the same pretrained small-model weights?
- ResNet shows worse validation accuracy than training from scratch. Is this consistent across optimizers and seeds?
- In GPT-2 paper, they use depth-$\mu$P scaling, e.g., $L^{-\frac{1}{2}}$ for residual connection. Did you also use it?
- What does Fig. 2(b) show regarding $\bar\sigma_{\Delta}$ How sensitive is performance to it, and how should it be chosen in practice?

**Limitations:**

See Strengths And Weaknesses.

**Strengths And Weaknesses:**

The paper studies an important and timely problem: how to train large-scale networks efficiently, especially when training models at multiple sizes is required. The theoretical analysis is solid and technically sound. However, the notation is not always easy to follow, largely due to the Tensor Program and $\mu$P framework. The experiments provide some support for the method, but they also raise several questions (see below).

The presentation could be improved. Sections 2.2 and 2.3 are quite technical and may be shortened or moved to the appendix, with more informal explanations and intuition kept in the main text. The introduction, particularly the statement of contributions, could also be clearer. To my understanding, the main contribution is a principled upscaling method together with an extension of the Master Theorem in Tensor Programs to justify the training dynamics and HP transfer under $\mu$P. If this is the core contribution, it would help to state it more directly. For example, the current framing around "function-preserving model expansions" is related but is not mentioned again in the main paper, which makes the overall narrative less focused.

In Section 3.2, the definition of the injected noise $\Delta^{(\ell)}$ is not clearly specified in the main text, which is somewhat confusing. Since correct and aligned width scaling is essential for ensuring that the Master Theorem and HP transfer apply under $\mu$P, the mean and variance of $\Delta^{(\ell)}$ should be stated explicitly. The paper only mentions in Meta-Algorithm 1 that the injected noise follows the same scaling as $\mu$P initialization of a fresh model. This implies that $\Delta^{(\ell)}$ is i.i.d. Gaussian with mean zero and variance $1/N$, which is later confirmed in Appendix B. Given that Section 3.2 already focuses on the MLP case, it would be clearer to directly state these details there instead of relying on indirect references.

Finally, since the authors mention extending the work to depth scaling in future directions, it would strengthen the related work section to discuss recent papers on depth scaling and HP transfer, including:
- https://arxiv.org/abs/2405.15712: Transformer limits under different depth scalings
- https://arxiv.org/abs/2512.21075: depth scaling in ResNet and restoration of GIA under depth-MuP
- https://arxiv.org/abs/2505.01618: empirical study of alternative depth scaling $L^{-1}$ and HP transfer
- https://arxiv.org/abs/2512.22382: empirical study of HP transfer across width, depth, and optimizers.

---

> ### Author Rebuttal · Authors · 2026-03-31
>
> We thank the reviewer for their thoughtful questions and constructive feedback. Below we address each point.
>
> **Presentation and framing of contributions**
>
> We will improve Sections 2.2 and 2.3 with more intuitive explanations and clarify the contributions statement.
>
> We stress that our primary contribution is **theoretical** rather than methodological: to our knowledge, this is the first rigorous framework for HP transfer of model upscaling. Our *first contribution*, the rigorous theory establishing static and dynamic equivalence across widths and its connection to $\mu$P, is a main contribution in its own right, opening the door to follow-up studies connecting model equivalence to infinite-width analysis and optimization theory.
>
> While all prior work on upscaling is methodological or empirical, building on Net2Net's "function-preserving expansion", our work aims to propose a new theoretical framework, introducing dynamic equivalence and connecting upscaling to the infinite-width limit literature. We will revise the contributions section to clarify this narrative.
>
> **Injected noise definition in Section 3.2**
>
> We agree that this lacks clarity and will add an explicit statement in Section 3.2 specifying that $\\Delta^{(\\ell)}$ denotes the noise injected in Step 2 of Meta-Algorithm 1, directing readers to Algorithm 2 in Appendix D where its distribution is given explicitly.
>
> **Related work on depth scaling**
>
> We will add a paragraph discussing the infinite-depth limit, HP transfer across depths, and depth upscaling, citing the listed references. We will also discuss how similar ideas might apply in the depth-scaling setting and the challenges involved.
>
> **Q1:** The best hyperparameters are selected based on *training loss* at the end of $T$ fixed steps on the small system ($n_0 \to kn_0$), then applied to the target ($n \to kn$) for the same $T$ steps. This follows $\mu$Transfer: since the theory governs training dynamics rather than generalization, training loss is the appropriate criterion. We train for the full $T$ steps to maintain a controlled setup, although prior work has empirically validated that one could potentially use fewer steps in the sweep for HP transfer. We will include these details in the revision.
>
> **Q2:** The interpretation is partially correct, with two caveats. First, under $\mu$P the noise variance differs for matrix-like vs. vector-like weights: $1/N$ for hidden weights but $1$ for input/output weights. Second, we introduce a tunable constant $\\overline{\\sigma}\_\\Delta$, so the actual variance is $\overline{\sigma}_\Delta^2/N$ for hidden weights.
>
> However, this alone does *not* justify applying $\mu$P analysis after upscaling. The standard Tensor Program framework does not handle non-zero-mean initialization. Our analysis must track two distinct sources of randomness: the base model's initialization and the injected noise. At any fixed training step, the weights decompose as the sum of the base initialization (lifted), accumulated base updates (lifted), injected noise, and accumulated upscaled-model updates, where the first and third terms are mean-zero random matrices, and the second and fourth are deterministic functions of them. This is formalized through our modified tensor program (Appendix E), which extends the framework to handle both sources of randomness.
>
> **Q3:** Our method involves four widths $n_0 < kn_0 < n < kn$ (Fig 1b). Since upscaling ($n \to kn$) and HP tuning ($n_0 \to kn_0$) target different widths, they necessarily start from different pretrained weights.
>
> **Q4:** Yes, this is consistent across seeds and optimization hyperparameters (momentum, weight decay, LR scheduling). This suggests the problem is specific to the CNN architecture and this task/dataset combination, which exhibits a large generalization gap.
>
> **Q5:** Yes, we applied the depth scaling.
>
> **Q6:** We believe the reviewer means Figure 3(b). The plot shows an asymmetric profile: values slightly above the optimum cause rapid degradation, while the low-noise region is relatively flat. This flatness reflects the fact that, as we theoretically establish, setting zero noise causes the upscaled model to evolve equivalently to the base model as if no upscaling had occurred. For this particular task, continuing to train the base model (with an adjusted lr) achieves a terminal loss not far from that of the properly upscaled model. But it is precisely this gap that constitutes the benefit of upscaling. Upon closer inspection, the left portion of each curve does exhibit a smooth decrease toward the optimum, confirming that the choice of $\overline{\sigma}_\Delta$ matters.
>
> Crucially, the optimal $\\overline{\\sigma}\_\\Delta$ is *stable across widths*: this is what HP transfer means. In practice, one tunes $\\overline{\\sigma}\_\\Delta$ on the small system ($n_0 \to kn_0$) and transfers directly to the target system ($n \to kn$), avoiding costly sweeps at scale.

---

> > ### Author Rebuttal · Reviewer_8z1w · 2026-04-03
> >
> > I appreciate the authors’ clarification.
> >
> > I believe the authors will make suitable revisions to improve the clarity and discussion regarding $\bar \sigma_{\Delta}$. I also feel that the theoretical contributions of this paper may have been underestimated by other reviewers. In my view, developing a principled framework based on rigorous theoretical analysis is just as important as proposing methodological contributions based on heuristics.
> >
> > Therefore, I am raising my score to Accept.

---

> > > ### Author Response · Authors · 2026-04-07
> > >
> > > We appreciate the reviewer's supportive feedback and the generous recognition of the theoretical contributions of our work. As you noted, establishing a rigorous, principled framework for model upscaling is the core objective of this study, and we are grateful that the reviewer finds merit in this direction.
> > >
> > > We thank the reviewer for the helpful suggestions regarding the noise parameter $\overline{\sigma}_{\Delta}$. In response, we have implemented the following revisions to the manuscript:
> > >
> > > * Clearly specified $\Delta^{(\ell)}$ in Section 3.2. We would like to note that the mean and variance of $\Delta^{(\ell)}$ were indeed stated explicitly in the original manuscript, but were buried in Algorithm 2 in Appendix D and not clearly presented when mentioned in Section 3.2. We have now added a pointer to clarify this, and we appreciate the reviewer pointing out this gap in readability.
> > > * Expanded the discussion of Figure 3(b) to: (1) clarify the implication of the relative stability below the optimal $\\overline{\\sigma}\_{\\Delta}$; and (2) highlight that the optimal $\\overline{\\sigma}\_{\\Delta}$ remains stable across widths, confirming the feasibility of HP transfer.
> > >
> > > We have also polished the overall presentation based on the points of confusion raised by the reviewers. We are grateful for this constructive feedback, which has helped us improve the clarity of our manuscript.

---

### Official Review · Reviewer_1Wn2 · 2026-03-11

**Soundness:** 2
**Presentation:** 2
**Significance:** 4
**Originality:** 4
**Overall Recommendation:** 5
**Confidence:** 4

**Summary:**

Summary
The authors extend previous work on upscaling (initialising larger models from pretrained smaller models) to demonstrate compatibility with tensor programs. As such, their upscaling recipe can inherit the hyperparameter transfer properties of muP, opening a pathway for upscaling to become common practise in practical large model training.

The authors first demonstrate equivalence between base and upscaled MLPs over a single training step with SGD and AdamW (Sections 2.1 - 2.2). They then illustrate equivalence for any model describable as a tensor program (Section 2.3), and how upscaling hyperparameter corrections can be reinterpreted under muP (Tables 1 and 2).

Since functional equivalence of larger models is actually undesirable (want better functions than possible at small scale), two new hyperparameters are required: a noise scale for perturbing the upscaled weights, and a new learning rate for the upscaled model. The authors propose a two-stage procedure to tune these new hyperparameters.
1. Upscale a tiny base model, and sweep over noise and learning rates independently to find optimum
2. Use hyperparameters from step 1 for upscaling real base model.

The authors find that training of upscaled models improves training loss across 3 different setups involving MLPs, Resnets, and Transformers (Figure 2). They also find noise and learning rates to be stable across 3 model sizes (Figure 3).

**Compliance With Llm Reviewing Policy:**

Affirmed.

**Final Justification:**

The authors resolved my main concern about FLOP-matched training of upscaled models. They show clear efficiency benefits to upscaling. I am raising my score as a result.

**Key Questions For Authors:**

Q1. How does upscaling compare to pretraining from scratch in a FLOP matched setting?
Q2. Figure 3b suggests noise std is fairly stable across model sizes. Can the value reported in the paper be taken as a practical recommendation that users no longer need to sweep?
Q3. If Q2 is yes, can you show more evidence of stability with larger sizes?

**Limitations:**

yes

**Strengths And Weaknesses:**

**Strengths**
* Elegantly combines upscaling with tensor programs
* Practical algorithm is simple and easy to implement from their description
* Diversity of experiments showing the technique working in different settings, as predicted by tensor programs
* Many members of ML community will be interested and likely build on this work

**Weaknesses**
* Major: If I understand correctly, training of an upscaled model uses roughly 1.5-1.75x the FLOPs of a wide model trained from scratch when accounting for pretraining smaller models from scratch and upscaling to intermediate width for upscaling hyperparameter transfer. 1.5x is the best case if you have access to optimal hyperparameters ahead of time. Eyeballing the GPT-2 training loss results it appears that the wide model trained from scratch reaches the initial upscaled model loss roughly 1/3 of the way through training. As a result, I suspect for the same FLOP count, the upscaling recipe is no better than training a wide model from scratch.
* Minor: A lot of details are buried in the Appendix. For example, the optimizer checkpoint upscaling is an important detail that I feel is warranted in the main text. More detailed information on experiments was also missing, making it harder to understand the true scale of experiments and how realistic it was to compare pretraining from scratch and upscaling the way the authors did.
* Minor: Meta-Algorithm 1 Upscaling procedure should include hyperparameter search too.
* Minor: It would help a lot of practically minded readers to have a footnote that says what a kronecker product with a matrix of ones looks like in pytorch code i.e., something like W_upscale = W.repeat_interleave(input_dim, dim=0).repeat_interleave(output_dim, dim=1), or to give a toy example. Appendix A gives this, however something small in the main text would go a long way.

---

> ### Author Rebuttal · Authors · 2026-03-31
>
> We thank the reviewer for their careful reading and constructive feedback. Below we address each point raised.
>
> > Major Concern: FLOP-matched comparison of upscaling vs. training from scratch; Q1
>
> We thank the reviewer for raising this point and would like to clarify our intended setup; we will **add this clarification to the paper**. Our work focuses on the scenario where one *already has access* to a trained base model of moderate size and wishes to train a larger model (e.g. to produce an LLM suite offering models at different sizes to cater to diverse user needs). In this setting, one can either train the larger model from scratch or apply upscaling to initialize it from the existing base model. The training cost of the base model is considered a sunk cost and is not factored into the comparison.
>
> This justifies the comparison between the wide model trained from scratch and the wide upscaled model in Fig 2: since both models have identical architectures and per-step compute costs, it suffices to compare their training and validation curves over epochs or steps, without the need for FLOPs-normalized curves.
>
> To be clear, we are *not* advocating for the alternative scenario in which one should always trains a large model by first training a smaller base model from scratch and then upscaling: in that case, the total compute including base model training would indeed need to be accounted for, and the reviewer's comparison would be appropriate.
>
> > Minor: Presentation improvements
>
> We thank the reviewer for these helpful suggestions and **will incorporate all of them in the revision**:
>
> 1. Add a concise description of the optimizer checkpoint transfer procedure in Section 3.1.
> 2. Move key experimental details (model widths, upscaling multipliers, parameter counts, hyperparameter sweep criteria) from the appendix into Section 4.
> 3. Extend Meta-Algorithm 1 to include the HP transfer step.
> 4. Add a footnote with a PyTorch code snippet illustrating the Kronecker product with a matrix of ones to aid practical understanding.
>
> > Q2–Q3: Stability of noise std across model sizes
>
> We are somewhat confused by the reviewer's question and we have two answers from two possible interpretations to this question.
>
> 1. The reason why the optimal noise std is stable *across model sizes* is a consequence of the "hyperparameter transfer" phenomenon. The observation in the plot shows that our theory allows us to correctly select a noise level that works for all model width $N$ for a given family of models and experimental setup. In practice, one tunes $\overline{\sigma}_\Delta$ on the small upscaling system ($n_0 \to kn_0$) and transfers directly to the target system ($n \to kn$), avoiding costly tuning on large-size.
>
> 2. We suspect the reviewer may have intended to ask whether Fig 3b suggests "training loss is stable across noise standard deviations", i.e., whether the relative flatness of the curves suggests that sweeping $\\overline{\\sigma}\_\\Delta$ is unnecessary. In response: even though the curves in Fig 3(b) appear relatively flat, this should not be interpreted as evidence that sweeping is unnecessary. First, the plot clearly shows that choosing a noise standard deviation slightly larger than the optimum leads to rapid performance degradation. Second, one might consider simply using a very small $\\overline{\\sigma}\_\\Delta$ to remain in the "safe" region, but as we theoretically establish in the paper, setting $\\overline{\\sigma}\_\\Delta$ causes the upscaled model to evolve equivalently to the base model, as if no upscaling had occurred. The relative flatness in the low-noise region reflects the fact that, for this particular task, continuing to train the base model (with an adjusted learning rate) achieves a terminal loss not far from that of the properly upscaled model—but it is precisely this gap that constitutes the benefit of upscaling. We also note that, upon closer inspection, the left portion of each curve exhibits a smooth decrease toward the optimum, confirming that the choice of $\\overline{\\sigma}\_\\Delta$ does matter.

---

> > ### Author Rebuttal · Reviewer_1Wn2 · 2026-04-03
> >
> > > Our work focuses on the scenario where one already has access to a trained base model of moderate size and wishes to train a larger model
> >
> > I'm not convinced that this is a sufficiently realistic scenario that you could completely discount the cost of training a smaller model. One approach (possibly the dominant approach) for generating a suite of models is to start large and distil. Another approach is to start small to derisk, then increase in size. I don't understand how you could not consider the cost of training the small model if sufficient resources were put behind it to make it good enough for release.
> >
> > > We are somewhat confused by the reviewer's question and we have two answers from two possible interpretations to this question.
> >
> > I'm very sorry for my badly specified question, but your answer provides sufficient clarification.

---

> > > ### Author Response · Authors · 2026-04-07
> > >
> > > We thank the reviewer for raising this concern. After further consideration, we agree that completely disregarding the cost of base model training can be unfair in certain application scenarios, and that a more comprehensive evaluation would strengthen our work. To this end, we will **add to the manuscript evaluations under both extremes: (1) fully excluding the base model training cost, as we do currently; and (2) fully including it, i.e., comparing the cost of (base model training + upscaled model training) against (training a wide model from scratch).** We note that the actual benefit of upscaling depends on the specific application scenario and likely lies somewhere between these two bounds. The results for MLP and GPT-2 are as follows; in both cases the upscaled model outperforms wide-from-scratch even under criterion (2). We acknowledge that this benefit is not observed in the ResNet experiment, where the wide-upscaled model reaches a similar terminal loss as wide-from-scratch, likely because ResNet's performance on CIFAR-100 already saturates at the base model's width—a limitation of upscaling specific to this model/task combination.
> > >
> > > * MLP (Fig. 2, first col):
> > >     * Base model training: 740 TFLOPs; upscaled model reaches terminal from-scratch loss at 2,854 TFLOPs; total cost = 3,594 TFLOPs.
> > >     * Wide model from scratch: 11,326 TFLOPs.
> > > * GPT-2 (Fig. 2, last col):
> > >     * Base model training: 787,191 TFLOPs; upscaled model reaches terminal from-scratch loss at 801,410 TFLOPs; total cost = 1,588,601 TFLOPs.
> > >     * Wide model from scratch: 2,242,335 TFLOPs.
> > >
> > > (Note: we are comparing the training cost in #FLOPs to reach the same loss)
> > >
> > > We would like to offer some more context for this choice of evaluation:
> > >
> > > * Model upscaling can be readily applied to open-source checkpoints from repositories such as HuggingFace, in which case the base model training cost is naturally a sunk cost. Accordingly, criterion (1) has been used in many prominent prior works on upscaling, including Net2Net [1], bert2BERT [2], LiGO [3], and Mango [4], while it is not used in some other works.
> > > * Criterion (2), which fully counts the base model training cost, can also be unfair, because the comparison is only valid if one trains the small model *solely* for upscaling and does so *exactly once*. In practice, the base model is often useful on its own and can be reused for multiple rounds of upscaling.
> > >
> > > Overall, it is difficult to determine the appropriate discount to apply to the base model training cost, which is why we have decided to present both extremes.
> > >
> > > We also apologize for the confusion introduced by the example scenario of "producing an LLM suite." We agree that if one proposes upscaling as an alternative approach for this purpose, one should evaluate the total cost of producing the entire suite—yet another distinct evaluation criterion. This has been investigated in recent work [5], and we believe that empirically evaluating our approach in this setting would be a valuable direction for future work.
> > >
> > > [1]Chen, Tianqi, Ian Goodfellow, and Jonathon Shlens. "Net2net: Accelerating learning via knowledge transfer." arXiv preprint arXiv:1511.05641 (2015).
> > >
> > > [2]Chen, Cheng, et al. "bert2bert: Towards reusable pretrained language models." Proceedings of the 60th Annual Meeting of the Association for Computational Linguistics (Volume 1: Long Papers). 2022.
> > >
> > > [3]Wang, Peihao, et al. "Learning to Grow Pretrained Models for Efficient Transformer Training." The Eleventh International Conference on Learning Representations.
> > >
> > > [4]Pan, Yu, et al. "Reusing pretrained models by multi-linear operators for efficient training." Advances in Neural Information Processing Systems 36 (2023): 3248-3262.
> > >
> > > [5]Yano, Kazuki, et al. "Efficient construction of model family through progressive training using model expansion." arXiv preprint arXiv:2504.00623 (2025).

---

### Official Review · Reviewer_HLgD · 2026-03-13

**Soundness:** 2
**Presentation:** 2
**Significance:** 3
**Originality:** 3
**Overall Recommendation:** 4
**Confidence:** 3

**Summary:**

This paper introduces a principled method to perform model upscaling, initializing a larger neural network from trained smaller ones. The key idea is to combine function-preserving weight transformations with injecting a small amount of noise to break symmetry and allow the wider model to exploit its additional capacity, based on the ideas of the µTransfer framework. Experimental evaluations on MLPs and GPT-2  models show the upscaled models converging faster and matching or outperforming wider models trained from scratch in terms of loss, though on ResNet architectures the upscaled model generalizes worse.

**Compliance With Llm Reviewing Policy:**

Affirmed.

**Final Justification:**

The authors engaged actively in the rebuttal process and ran new experiments, addressing several of my concerns. I have therefore increased my score, now leaning towards acceptance of the paper. I still believe some more empirical comparisons to other methods from the literature (see comments from reviewer 9RPp), even if not as theoretically grounded as the one proposed in this paper are missing to make this a clear accept.

**Key Questions For Authors:**

Could you extend your empirical evaluations to warmstart at much larger scales (e.g., upscaling width by 10×), which is the regime where µTransfer-based hyperparameter transfer is most relevant?

**Limitations:**

Though I appreciate the authors being open about the limitations of their method on the ResNet on CIFAR case, it is such a basic benchmark that a shortcoming here raises concerns about potential fundamental issues of the proposed method in practice.

**Strengths And Weaknesses:**

Strengths

-  Hyperparameter transfer and warmstarting methods, especially µTransfer-like are important for the training efficiency of large models
- The theoretical derivations are well-structured, progressively generalizing from simple MLPs with SGD to advanced architectures and adaptive optimizers

Weaknesses

- Empirical evaluations are very limited in scope and results are not conclusive. µTransfer is meant for jumping at least one order of magnitude in width, yet the paper only tests 2x width multipliers. The ResNet results on CIFAR-100 (a very basic benchmark) show upscaling actually hurts generalization
- No experimental comparison at all to other warmstarting methods in the literature
- Additional hyperparameter tuning costs (std of injected noise, learning rate) are not accounted for in the analysis in Section 4. Without measuring total compute including these sweeps, the practical savings of the method remain unquantified.

---

> ### Author Rebuttal · Authors · 2026-03-31
>
> We thank the reviewer for their thoughtful feedback. We address each concern below.
>
> > Empirical evaluations are limited in scope
>
> We stress that our primary contribution is **theoretical**: to our knowledge, this is the first rigorous framework for HP transfer of model upscaling. Our experiments validate the theory's predictions rather than empirically studying upscaling at scale.
>
> Our theory applies universally across a broad class of architectures and optimizers; to validate this, we experiment with MLPs, ResNets, and GPT-2 trained with SGD variants and AdamW. Since our theory does not make predictions about certain practical factors (e.g., base scale, growth factor, or generalization), and such factors tend to be task- and dataset-specific, meaning empirical investigation may not yield universal insights about the upscaling method itself, we focus on validating the specific phenomena the theory does predict.
>
> > Results are not conclusive
>
> We respectfully note that our experiments consistently confirm the theory's predictions across both aspects of our framework.
>
> 1. HP transfer (Figures 3 and 7): These figures validate our core contribution: theory-grounded HP transfer across upscaling systems of different widths (different from the original $\mu$Transfer).
>
> 2. Upscaling experiments (Figure 2): We will **add two clarifications**: (1) *Compute savings:* upscaled models reach low training loss significantly faster than training from scratch. (2) *Final performance:* upscaled models consistently achieve lower or comparable final training loss across all experiments and seeds.
>
> > No experimental comparison to other warmstarting methods
>
> In response, we will **add an empirical comparison with Net2Net (see attachment).** The key advantage is tuning cost: Net2Net requires tuning noise and target LR at the large-model scale (their Fig 4, App A), whereas our width-scaling rules enable zero-shot HP transfer from small systems. The tuning cost savings are explicitly computed. Final performance should be comparable, as Fig 3 validates HP transfer across sizes.
>
> We do not compare with other prior methods, as they are tailored for specific architectures and heuristically designed. Our objective is a mathematical framework that applies universally across architectures, rather than claiming empirical superiority over model-specific pipelines.
>
> > ResNet results on CIFAR-100 show upscaling hurts generalization
>
> We appreciate this observation. Our theory addresses training dynamics, not generalization, so this does not contradict our theory. In terms of *training*, the upscaled model consistently outperforms training from scratch in convergence speed and attain similar terminal loss.
>
> Generalization is a separate issue: large generalization gaps are a known property of CNN-based image classification. We include this experiment to show that practitioners may be confident in upscaling when validation and training performance largely align, but should exercise caution otherwise. This is unlikely specific to our method, but a consideration for upscaling in general. We will **expand on this in the revision.**
>
> > Additional HP tuning costs are not accounted for
>
> We thank the reviewer for raising this. We address it from two angles and will **add these clarifications**.
>
> 1. Practical savings can be compared against prior methods. Methods like Net2Net lack principled tuning guidance, requiring HPs to be tuned at the upscaled model size. Our method enables tuning on much smaller models, substantially reducing cost, which we compute explicitly (attachment Tab 1).
>
> 2. Tuning is performed at width $kn_0$ while the target trains at width $kn$. The relative tuning cost decreases as $n/n_0$ grows, so savings are most pronounced at large scale $n\gg n_0$. Fully demonstrating this in academic settings is difficult, as $n_0$ must be ~100 for reliable HP transfer. Our experiments are designed as proof-of-concept validations of the theory rather than demonstrations of compute savings at practical scale (attachment Fig 2)
>
> > Warmstart at much larger scale
>
> Our method involves four widths $n_0 < kn_0 < n < kn$ (see Fig 1b). Our answer depends on which width ratio the reviewer refers to.
>
> 1. Upscaling multiplier $k$: Our choices ($2\times$–$4\times$) are consistent with prior works e.g. Net2Net. Even modest multipliers can increase parameter count by an order of magnitude, making upscaling practically useful.
>
> 2. Target-to-tuning ratio $n/n_0$: This is the analogue of the $\mu$Transfer width multiplier. Prior works have theoretically and empirically show that, once $n_0$ is sufficiently large (close to infinite-width limit), HP transfer works robustly for $n\gg n_0$. We expect the same in our setting. But as mentioned earlier, empirically validating on large $n/n_0$ is prohibitive: original $\mu$Transfer was also evaluted on similar $n/n_0$.
>
> **Attachment**: https://anonymous.4open.science/r/icml_rebuttal26-44ED/rebuttal-attachment.pdf

---

> > ### Author Rebuttal · Reviewer_HLgD · 2026-04-03
> >
> > I thank the authors for responding to my concerns.
> >
> > **On the scope of empirical evaluation:** While the authors frame their contributions as primarily theoretical, the practical relevance of their findings lies in large-scale training runs. An empirical validation at meaningful scale is therefore crucial for any method targeting efficient training strategies. As the authors themselves acknowledge in the rebuttal, "savings are most pronounced at large scale." This reinforces my point that at least one larger-scale experiment should be included to confirm the theoretical derivations hold in the regime where the method is most useful.
> >
> > **On ResNet generalization on CIFAR-100:** To my understanding, µTransfer also focuses only on training dynamics, yet models trained with transferred hyperparameters (including ResNets) show comparable relative generalization performance at the hyperparameter optimum (see Figure 16 of [1]).
> >
> > **On additional HP tuning costs:** My concern was specifically about the results in Figure 2. To produce the "wide model (upscaled)" curve, one must perform tuning sweeps over the learning rate and noise standard deviation. These costs do not arise when training the "wide model (trained from scratch)" baseline. These additional costs should be accounted in some form.
> >
> > **On related work:** I appreciate the added Net2Net baseline and also await the authors' response to the related work raised by reviewer 9RPp.
> >
> > **On the HP transfer width range:** I was referring to the “target-to-tuning ratio”, analogous to the µTransfer width multiplier. The original µTransfer paper demonstrates transfer from widths 128 up to 8192 (a 64× range). In contrast, the transfer ranges in this paper are considerably smaller: The HP transfer verification in Figure 3 spans widths 256 to 1024, only a factor of 4×.
> >
> >
> > [1] Yang et al., "Tensor Programs V: Tuning Large Neural Networks via Zero-Shot Hyperparameter Transfer" https://arxiv.org/pdf/2203.03466

---

> > > ### Author Response · Authors · 2026-04-07
> > >
> > > **Scope of empirical validation and large "target-to-tuning ratio"**
> > >
> > > We thank the reviewer for clarifying that their concern pertains to the "target-to-tuning ratio" $n/n_0$ rather than the "upscaling multiplier" $k$ (our choice of $k$ is consistent with prior work). While we believe, based on theoretical and empirical evidence from prior work on $\mu$Transfer, that our method should continue working for $n \gg n_0$, we appreciate the constructive feedback and **have performed the GPT-2 experiment at $n/n_0 = 10$, as requested**. We apologize for the delayed reply, as this experiment took several days to run.
> > >
> > > Please refer to the attachment for the result. The setup follows Fig. 2 exactly, with HP tuning performed from width $n_0 = 192$ to $kn_0 = 384$. The base model is now trained at width $n = 1920$, and both wide models at width $kn = 3840$ (contrasting with the previous $n = 384$, $kn = 768$). The result exhibits the expected behavior: the wide-upscaled model converges much faster and reaches a lower terminal loss than wide-from-scratch, for both training and validation. We will include this in the revised manuscript, and we hope this resolves the reviewer's concern.
> > >
> > > **On ResNet generalization**
> > >
> > > We reiterate that our experiments are not designed to demonstrate practical scenarios in which our methods are useful. Rather, they are designed to verify our theoretical framework, which concerns training dynamics, not generalization. That said, we address the comparison with [1] directly.
> > >
> > > We agree that the generalization gap does not arise as a problem in all of [1]'s experiments. However, [1] itself states that "we primarily focus on hyperparameter transfer with respect to training loss. In settings where regularization is not the bottleneck to test performance, as in all of our experiments here, this also translates to efficacy in terms of test loss." Notably, [1] does not claim that generalization works universally.
> > >
> > > Our ResNet experiment is precisely a setting where regularization *is* the bottleneck: the dataset is small relative to model capacity, placing models deep in the overparameterized regime where the difference in test performance is dominated by the generalization gap rather than by any difference in training performance. Moreover, [1] compares same-width models with smoothly varied HPs, so both dynamics and generalization vary smoothly. In contrast, we compare a model trained from scratch against one warmstarted via upscaling, where both training dynamics and generalization differ substantially. Even with HPs tuned at the upscaled size, it is not certain that upscaling yields better generalization — this gap may be inherent to the upscaling procedure itself, rather than to HP transfer.
> > >
> > > The behavior we observe is fully consistent with [1]'s own caveat. Investigating the generalization behavior of upscaling is an interesting future direction, but orthogonal to the HP transfer guarantees our paper provides.
> > >
> > > **On additional HP tuning costs**
> > >
> > > We appreciate the clarification of this concern. First, we note that the most appropriate baseline is Net2Net, which itself does not account for additional HP tuning cost — and any such tuning for Net2Net must be performed at the upscaled size, making it substantially more costly than in our method. Second, we **provide an explicit cost breakdown for the new experiment with $n/n_0 = 10$**:
> > >
> > > - Training cost of wide-from-scratch: 37,929,809 TFLOPs
> > > - Training cost of wide-upscaled (to reach the same terminal loss): 6,349,450 TFLOPs; HP tuning cost (suppose performed on $5\times 5$ grid for lr and noise at width $kn_0 = 384$): 19,679,775 TFLOPs. Total cost: 26,029,225 TFLOPs.
> > >
> > > Even after accounting for the HP tuning cost, the total cost of upscaling to match the terminal loss of wide-from-scratch remains much smaller. We also note that this benefit should be more pronounced at larger scale, or when coupled with more efficient tuning methods like random search.
> > >
> > > **On related work** Please refer to our response to Reviewer 9RPp.
> > >
> > > **On the HP transfer width range**
> > >
> > > In the final revision, we **will add $N=128$ and $N=2048$ to Fig. 3**, extending the width range to $16\times$. Due to limited computational resources, we are unable to provide those results during the rebuttal period. Validating over even larger ranges would require resources comparable to the original $\mu$Transfer paper (conducted at Microsoft and OpenAI), which are beyond our current means. That said, our theory directly extends the Tensor Program framework, so the underlying HP transfer mechanism is the same. Since HP transfer based on $\mu$P has been empirically validated to be robust at large scale, we expect similar behavior in our setting. We believe the merit of our theoretical contribution is to provide confidence that the method will work at scale before committing large compute budgets.
> > >
> > > **Attachment:** https://anonymous.4open.science/r/icml_rebuttal26-44ED/rebuttal-attachment2.pdf

---

### Official Review · Reviewer_9RPp · 2026-03-16

**Soundness:** 3
**Presentation:** 2
**Significance:** 2
**Originality:** 2
**Overall Recommendation:** 3
**Confidence:** 4

**Summary:**

This paper proposes a principled framework for upscaling pretrained neural networks to larger widths, combining function-preserving weight transformations with μP-compatible hyperparameter rescaling. The central contributions are: (i) a theoretical treatment of static and dynamic equivalence between networks of different widths under general optimizers, (ii) an upscaling algorithm grounded in this theory with noise injection to break symmetry, and (iii) a hyperparameter transfer procedure that allows tuning on small proxy upscaling systems and transferring to the target scale.

**Compliance With Llm Reviewing Policy:**

Affirmed.

**Final Justification:**

The paper has merits, but due to limited experiments, I recommend rejection at this stage.

The authors should empirically compare against the baselines mentioned during the rebuttal on a fair experimental protocol.

**Key Questions For Authors:**

1. Why does the initialization loss of the upscaled model not match the terminal loss of the base
model in Figure 2? Is function preservation being achieved in practice?
2. What is the empirical benefit of μP-aware optimizer state transfer versus a simpler copy-and-
perturb without it?
3. Can FLOPs-normalized training curves be provided?
4. How sensitive are results to the growth factor k, and what motivates the choice of k=2?
5. How does this method compare empirically to [1] and [2], particularly on final performance and
compute efficiency?
6. What is the recommended practical procedure for a large-scale deployment?

**Limitations:**

The empirical contribution is weak: baselines are absent, function preservation is not convincingly demonstrated, comparisons are not FLOPs-normalized, and the experimental scope is narrow. The novelty of the hyperparameter transfer contribution is substantially diminished by concurrent and prior work [1]. The paper would be considerably stronger with rigorous empirical comparisons, FLOPs-normalized results, and a systematic analysis across growth factors and model scales.

**Strengths And Weaknesses:**

Strengths:

The problem is well-motivated. Warmstarting larger models from smaller pretrained ones is practically relevant, and the observation that hyperparameter tuning for upscaled models is underexplored is fair. The connection to μP is elegant, and the extension of dynamic equivalence to general entrywise optimizers (Proposition 2.4) and to the NExORT framework (Theorem 2.5) is technically clean. The upscaling algorithm is clearly specified, and the implementation details in the appendix are thorough.

Weaknesses:

Originality: The paper’s core contribution — copy-and-perturb with μP-compatible hyperparameter rescaling — has substantial overlap with prior work. Proposition 2.1 (static equivalence) is essentially Net2Net’s Net2WiderNet [2], a connection the authors acknowledge but whose implications for novelty are undersold. More critically, recent work [1] covers much of the same empirical territory, hyperparameter transfer for upscaling across multiple scales, and does so more carefully, comparing against optimal training baselines (tokens-per-parameter regime) and sweeping growth factors. The claim that hyperparameter transfer for upscaling is novel is difficult to sustain in light of [1]. The dynamic equivalence result is the primary theoretical novelty, but its practical value is questionable: The entire point of upscaling is to break that equivalence via noise injection, which makes the extensive theoretical development feel disconnected from the empirical contribution.

Missing baselines: This is the most significant empirical gap. There is no comparison to Net2Net [2], bert2BERT [3], or the approach of [1]. Without such comparisons, it is impossible to assess whether the μP-aware hyperparameter rescaling and optimizer state transfer provide any practical benefit over simpler copy-and-perturb schemes. This is a critical omission for a paper claiming to provide a principled, superior method.

Function preservation not demonstrated empirically: In Figure 2, the initialization loss of the upscaled model visibly does not match the terminal loss of the base model, which directly contradicts what function preservation should guarantee in the zero-noise limit. This discrepancy is never addressed or explained.

FLOPs-normalized comparisons are absent: Plotting performance against epochs or steps is misleading because the base and upscaled models have substantially different per-step compute costs. Without FLOPs-normalized training curves, it is impossible to assess whether upscaling actually reduces total compute relative to training the larger model from scratch, which is the paper’s central practical motivation.

Narrow experimental scope: Experiments cover a single model family for language modeling (GPT-2), a single base scale, and a single growth factor (k=2). By analogy with depth upscaling [4], the growth factor is known to matter, and there is an effective range beyond which benefits diminish. No analysis is provided for why k=2 is chosen or how results depend on it. The models are presumably small, and it is unclear whether conclusions transfer to the scales of practical interest.

Marginal final performance gains: Across the reported experiments, upscaled models converge faster initially but reach roughly the same final performance as training from scratch. For the ResNet, the upscaled model is strictly worse on validation. The paper does not engage seriously with this: if terminal performance is equivalent, the benefit is limited to a faster starting point, and its value depends on the compute regime.

1. Mallik, N. et al.: Warmstarting for Scaling Language Models. (2024)
2. Chen, T. et al.: Net2Net: Accelerating Learning via Knowledge Transfer. (2015)
3. Chen, C. et al. bert2BERT: Towards Reusable Pretrained Language Models. (2022)
4. Du, W. et al. Stacking Your Transformers: A Closer Look at Model Growth for Efficient LLM Pre-Training. (2024)
5. Samragh, M. et al. Scaling Smart: Accelerating Large Language Model Pre-Training with Small Model Initialization. (2024)

---

> ### Author Rebuttal · Authors · 2026-03-31
>
> We sincerely thank the reviewer for their constructive feedback. We were unable to identify references [1,4] from context alone and would be grateful if the reviewer could share them so we can engage with these works properly.
>
> We would like to clarify what may be a misunderstanding regarding the nature of our work. Our main contribution is **theoretical**: to our knowledge, we provide the first rigorous framework for hyperparameter (HP) transfer of model upscaling. Our experiments validate the theory's predictions rather than empirically studying upscaling at scale.
>
> **Theoretical overlap with Net2Net** is limited to Prop 2.1, a warm-up for intuition rather than a core contribution. Our theory goes substantially beyond Net2Net (which proves only static equivalence for MLPs) in three ways:
> 1. Extend static to dynamic equivalence, and rigorously prove both for general architectures and optimizers (Sec 2, App C).
> 2. (Core) Establish *HP transfer for upscaled systems* via infinite-width limit theory (distinct from typical $\mu$Transfer), greatly improving tuning efficiency compared to Net2Net (Sec 3, App E.2).
> 3. Enable rigorous characterization of infinite-width training dynamics with upscaling, opening the door to future analysis (Sec 3.2, App E).
>
> We plan to: **note in the abstract that our upscaling procedure is mechanically akin to Net2Net; expand the Net2Net discussion in App B; and demonstrate empirically how our HP transfer reduces tuning cost (see attachment).**
>
> > [1] covers the same empirical territory
>
> We cannot identify [1], but the reviewer notes it is empirical, making it fundamentally different from our mathematically grounded HP transfer claim. To our knowledge, no prior work establishes such a theoretical foundation for this setting.
>
> **Practical value of dynamic equivalence.** It provides an important baseline: zero noise is provably equivalent to continuing base-model training. If a nonzero noise level is selected in the sweep, practitioners can be confident that upscaling provides genuine benefits over simply continuing training.
>
> **Our theory is not disconnected from noise injection, but directly treats it.** Sec 3 and App E prove that appropriate scaling of noise and HPs keeps training in the optimal $\mu$P regime, ensuring noise injection is both safe (preserving the base-model signal) and useful (neither exploding nor vanishing) in the infinite-width limit.
>
> In revision, we will also **clarify practical benefits**: (i) compute savings via efficient HP tuning and accelerated convergence, and (ii) better final performance: upscaled models *do* consistently achieve lower final training loss across all experiments and seeds, except in the ResNet experiment where final loss is roughly the same due to model saturation on that particular dataset, not a flaw of our method.
>
> **Q1.** Nonzero injected noise expectedly raises initialization loss above the base model's terminal loss; this is not a failure of function preservation. We will **add a clarifying comment.** Also, we did numerically verified static and dynamic equivalences via pytest across architectures and optimizers (App C.4).
>
> **Q2.** The $\mu$P-scaled optimizers are critical for HP transfer, and transferring optimizer state promotes early-stage stability by preserving dynamic equivalence (App D). A simple copy-and-perturb scheme without state transfer disrupts momentum and variance estimates (e.g., in AdamW) and may cause post-upscaling instability.
>
> **Q3.** The relevant comparison in Fig 2 is wide-from-scratch (yellow) vs. wide-upscaled (pink). They share the same architecture and per-step cost, making FLOPs normalization unnecessary. The base model (blue) is included only for reference.
>
> **Q4.** The optimal k is expected to be task- and dataset-specific, so our theory cannot predict it and empirical studies may not yield universal insights. Our choice of k=2 (k=4 for MLPs) suffices for validating the theory and is consistent with prior work.
>
> **Q5.** We will **add an empirical comparison with Net2Net (see attachment).** The key advantage is tuning cost: Net2Net requires tuning noise and target LR at the large-model scale (their Fig 4, App A), whereas our width-scaling rules enable zero-shot HP transfer from small systems. Final performance should be comparable, as Fig 3 validates HP transfer across sizes. Methods like bert2BERT are tailored for specific architectures, whereas our objective is a mathematical framework that applies universally across standard architectures and optimizers, rather than claiming empirical superiority over model-specific pipelines.
>
> **Q6.** While Net2Net may not scale to large deployments due to tuning cost, our theory suggests coupling Net2Net with HP transfer, and tuning-cost savings are expected to grow at scale (see attachment). The method can be applied in practice with minimal code change.
>
> **Attachment**: https://anonymous.4open.science/r/icml_rebuttal26-44ED/rebuttal-attachment.pdf

---

> > ### Author Rebuttal · Reviewer_9RPp · 2026-03-31
> >
> > Sorry for the missing references at the end of the review text:
> >
> > 1. Mallik, N. et al.: Warmstarting for Scaling Language Models. (2024)
> > 2. Chen, T. et al.: Net2Net: Accelerating Learning via Knowledge Transfer. (2015)
> > 3. Chen, C. et al. bert2BERT: Towards Reusable Pretrained Language Models. (2022)
> > 4. Du, W. et al. Stacking Your Transformers: A Closer Look at Model Growth for Efficient LLM Pre-Training. (2024)
> > 5. Samragh, M. et al. Scaling Smart: Accelerating Large Language Model Pre-Training with Small Model Initialization. (2024)
> >
> > It would be nice if you could consider your answer in the light of the added information.

---

> > > ### Author Response · Authors · 2026-04-07
> > >
> > > We thank the reviewer for providing the references. We will add citation [1] (the other works have already been cited). Our previous comments regarding [1,2,3] remain valid. Moreover, we would like to clarify a possible misunderstanding: none of these works on model upscaling demonstrates how to *transfer* hyperparameters, which is the main contribution of our work.
> > >
> > > **Originality**
> > >
> > > Apart from the empirical–theoretical distinction, **[1]** proposes a fundamentally different approach from ours. The first difference lies in the upscaling procedure itself: their approach relies on zero-padding, whereas ours constructs a dynamically equivalent model by duplicating and rescaling both the weights and the optimizer state. Importantly, their method **does not guarantee robust HP transfer (despite using $\mu$P)**:
> > >
> > > * **Learning rate**: [1] uses the best lr of the base model to train the upscaled model, implicitly assuming that HP transfer holds. However, this assumption is neither justified empirically nor theoretically. In contrast, our method considers HP transfer from a small upscaling system (width $n_0 \to k n_0$) to a larger upscaling system (width $n$ (base model) $\to kn$ (upscaled model)), with both theoretical and empirical justification.
> > >     * Empirically, [1] does not validate whether the optimal lr matches between the base and upscaled models, whereas we numerically validate the lr transfer for our method in Fig. 3(a). Moreover, our experiments indicate that the optimal lr constant for the base and upscaled models does **not** match in all cases (e.g., in the MLP with AdamW experiment, they are $0.1$ and $0.013$ respectively; see Appendix F). This suggests that directly reusing the same lr is suboptimal for upscaling.
> > >     * Theoretically, HP transfer should not be expected in their setting, since the infinite-width limit theory underlying $\mu$Transfer does not apply to the training dynamics of [1]'s upscaled model, which involves non-zero-mean initialization. The training instability reported in [1]'s RQ3 for slightly larger $\lambda_{\text{shrink}}$ strongly supports this. In contrast, our method's HP transfer is theoretically justified: in Sec. 3 and Appendix E, we carefully extend the Tensor Programs framework to accommodate upscaling.
> > > * **Injected noise**: [1] does not prescribe a principled way to choose the injected noise, to which upscaling performance is highly sensitive according to our Fig. 3(b) and prior works on shrink-perturb methods. In contrast, our HP transfer framework enables this quantity to be tuned efficiently via HP transfer as well.
> > >
> > > **[3–5]** are also fundamentally different from our work. Not only are they all empirical studies focused on proposing upscaling methods for a specific LLM architecture—whereas our work provides a theoretical framework that applies universally to general models and optimizers—but more importantly, all of them require one to either re-tune hyperparameters at the computationally expensive target scale or resort to guessing or heuristics (leading to suboptimal performance). Our HP transfer framework was designed to resolve this inefficiency.
> > >
> > > **Growth factor**
> > >
> > > > [1] sweeps over growth factors
> > >
> > > We note that [1] performs this sweep at the original upscaled scale, which incurs substantial tuning cost and is unlikely to be practical in real-world upscaling scenarios. Moreover, as noted earlier, we believe the optimal growth factor $k$ is largely task- and dataset-dependent. Consequently, the empirical observations in [1] may not necessarily generalize to other models or tasks.
> > >
> > > > By analogy with depth upscaling [4], the growth factor is known to matter
> > >
> > > [4] provides an empirical, scaling-law analysis of the effect of the growth factor on model upscaling. While we agree that a similar empirical investigation for width upscaling would be valuable, it falls outside the scope of our work. Our contribution is to propose a theory-grounded HP transfer framework that effectively determines **certain** important HPs, including the learning rate and injected noise, and our experiments are designed to validate its effectiveness. Similar to $\mu$Transfer, our method does not extend to other HPs (e.g., the growth factor), making such investigation a separate question for future work.
> > >
> > > We hope we have adequately addressed all of the reviewer's concerns. In particular, we highlight the central contribution of our work: **theoretically grounded HP transfer for upscaled models**. Robust HP transfer in this setting has not been achieved (either empirically or theoretically) in prior works, and accomplishing this substantially reduces the tuning cost associated with model upscaling. We would greatly appreciate the reviewer reconsidering their evaluation.

---

### Decision · Program_Chairs · 2026-04-30

**Decision:**

Accept (regular)

**Comment:**

This paper presents a principled μP-based framework for model upscaling with theoretically grounded hyperparameter transfer. All reviewers acknowledged the solidity and novelty of the theoretical contributions. The primary concern centered on the limited scope of experiments and the lack of baselines. During the rebuttal, the authors actively addressed these issues by providing additional experiments at larger scale and detailed FLOPs analyses, which convinced two reviewers to raise their scores. While the experimental evaluation could still be strengthened with broader comparisons to existing methods, the theoretical foundation is sound and the practical algorithm is clearly specified. Overall, the merits outweigh the remaining weaknesses, and I recommend acceptance.